# REFINE: Inversion-Free Backdoor Defense via Model Reprogramming

**Yukun Chen**[1,2,*], **Shuo Shao**[1,2,*], **Enhao Huang**[1,2], **Yiming Li**[1,3,✉], **Pin-Yu Chen**[4], **Zhan Qin**[1,2], **Kui Ren**[1,2]

[1] State Key Laboratory of Blockchain and Data Security, Zhejiang University
[2] Hangzhou High-Tech Zone (Binjiang) Institute of Blockchain and Data Security
[3] Nanyang Technological University  [4] IBM Research

`{yukunchen, shaoshuo_ss, huangenhao, qinzhan, kuiren}@zju.edu.cn;`
`liyiming.tech@gmail.com; pin-yu.chen@ibm.com`

## Abstract

Backdoor attacks on deep neural networks (DNNs) have emerged as a significant security threat, allowing adversaries to implant hidden malicious behaviors during the model training phase. Pre-processing-based defense, which is one of the most important defense paradigms, typically focuses on input transformations or backdoor trigger inversion (BTI) to deactivate or eliminate embedded backdoor triggers during the inference process. However, these methods suffer from inherent limitations: transformation-based defenses often fail to balance model utility and defense performance, while BTI-based defenses struggle to accurately reconstruct trigger patterns without prior knowledge. In this paper, we propose REFINE, an inversion-free backdoor defense method based on model reprogramming. REFINE consists of two key components: **(1)** an input transformation module that disrupts both benign and backdoor patterns, generating new benign features; and **(2)** an output remapping module that redefines the model's output domain to guide the input transformations effectively. By further integrating supervised contrastive loss, REFINE enhances the defense capabilities while maintaining model utility. Extensive experiments on various benchmark datasets demonstrate the effectiveness of our REFINE and its resistance to potential adaptive attacks.

## 1 Introduction

Deep neural networks (DNNs) have been widely deployed across various domains (He et al., 2023; Liu et al., 2024; He et al., 2024; Zhang et al., 2024). To develop a high-performance DNN, developers necessitate not only high-quality data samples but also substantial computational resources. Consequently, developers frequently and directly rely on third-party models for follow-up development. However, the utilization of third-party DNNs can introduce security threats, particularly with regard to backdoor attacks (Gu et al., 2019; Li et al., 2022b; Dong et al., 2023; Gao et al., 2024).

Backdoor attacks aim to implant hidden backdoors into the model during training (Gu et al., 2019). After the attack, the backdoored model functions normally on benign inputs. However, when a specific trigger is present, the model will produce intentionally incorrect outputs. Backdoor attacks pose a severe threat to critical applications where model reliability is essential, highlighting the urgent need for effective backdoor defense strategies to safeguard AI systems (Li et al., 2024c).

Currently, several backdoor defenses (Huang et al., 2022; Li et al., 2024a;b; Hou et al., 2024) have been developed to tackle the threat of backdoor attacks. Among these, pre-processing-based defenses (Villarreal-Vasquez & Bhargava, 2020; Qiu et al., 2021) are particularly notable because they only apply certain modifications to input samples before model inference, without altering the original model structure and weights. Currently, there are two main types of pre-processing-based defenses. The first type of defenses relies on input transformations (Li et al., 2021c; Sun et al.,

---

*The first two authors contributed equally to this work. ✉ Corresponding author: Yiming Li. Our code is available at `https://github.com/WhitolfChen/REFINE` and `BackdoorBox`.

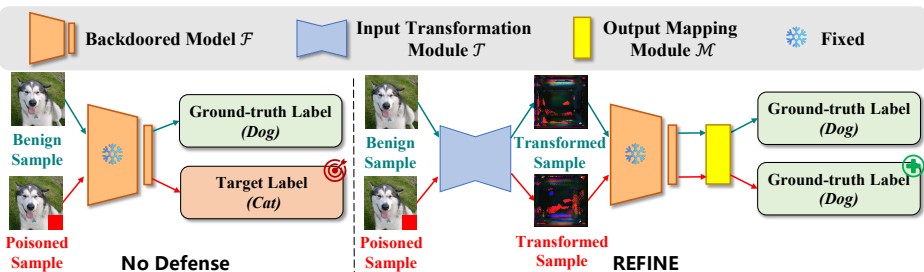

Figure 1: The defense process of our REFINE. The label remapping in the model's output domain significantly enhances the flexibility of input transformations while maintaining consistent sample predictions, effectively mitigating the trade-off often encountered in transformation-based pre-processing defenses. During prediction, the input sequentially passes through the well-trained input transformation module, the fixed backdoored model, and the pre-defined output mapping module, ultimately yielding the expected ground-truth (instead of the malicious target) label.

2023; Shi et al., 2023). These defenses aim to mismatch or eliminate potential trigger patterns by performing certain transformations to the input samples. The second type is based on backdoor trigger inversion (BTI) (Wang et al., 2019; 2023; Xu et al., 2024), which attempts to reconstruct the attacker's trigger patterns and remove them before the data is processed by the model.

In this paper, we revisit the aforementioned pre-processing-based backdoor defenses. We reveal that they both have intrinsic limitations. Specifically, transformation-based defenses face a trade-off between model utility and defense performance: more extensive transformations can achieve lower attack success rates but may negatively impact the model's benign accuracy. This occurs because these defenses lack information about backdoor-related features, forcing them to modify all features indiscriminately, including those critical for benign accuracy. The close coupling of benign and backdoor features makes it difficult to apply stronger transformations without significantly compromising model utility. On the other hand, BTI-based defenses can 'break' the trade-off by first obtaining the information of backdoor triggers via trigger inversion. However, due to the inherent difficulties of BTI (*e.g.*, lack of prior knowledge about the implanted backdoor and poisoned samples), existing BTI methods struggle to accurately invert trigger patterns. This limitation makes it difficult to purify the backdoored input from the poisoned domain to the benign domain, leading to limited effectiveness of BTI-based defenses. Accordingly, an intriguing and important question arises: *Could we break the curse of this trade-off without relying on backdoor trigger inversion?*

To tackle the above challenge, we first provide a theoretical analysis showing that the effectiveness of backdoor defenses is bounded by the distance between output features before and after pre-processing. Accordingly, the ineffectiveness of existing defenses is mostly due to their underlying assumption of having a fixed output domain. Based on the above understandings, inspired by model reprogramming (Chen, 2024), we propose REFINE, a REprogramming-based Inversion-Free backdoor defeNse mEthod, as shown in Figure 1. By allowing changes to the output domain, REFINE can significantly alter the input domain while largely maintaining model accuracy. Specifically, our REFINE involves an input transformation module and an output mapping module to reprogram the backdoored model and eliminate backdoor triggers. We utilize a trainable autoencoder as the input transformation module and redefine the model's output domain through a hard-coded remapping function. This adjustment to the output domain enables more extensive and effective input transformations. Besides, we enhance our method by applying supervised contrastive loss (Khosla et al., 2020), ensuring that transformed samples of the same class remain closely aligned.

Our contributions are three-fold. **(1)** We revisit existing pre-processing-based backdoor defenses and reveal their limitations. **(2)** Based on the empirical and theoretical analysis, we propose a simple yet effective defense (*i.e.*, REFINE). Our REFINE introduces trainable input transformation and output mapping modules for reprogramming and incorporates cross-entropy and supervised contrastive losses to enhance defense performance. **(3)** Extensive experiments on diverse benchmark datasets demonstrate the effectiveness of REFINE and its resistance to potential adaptive attacks.

## 2 BACKGROUND

### 2.1 BACKDOOR ATTACKS

Backdoor attacks (Gao et al., 2020; Li et al., 2024c) involve embedding hidden malicious behaviors into a model, typically by manipulating the training process with a small subset of poisoned data

containing adversary-specified trigger patterns. Whenever the trigger appears in the input during inference, the model executes the attacker's intended behavior, such as misclassifying the input to a target label. In the absence of the trigger, the model functions normally, rendering the backdoor hard to detect. Backdoor attacks pose serious threats in AI-enpowered systems.

The formulation of backdoor attacks is typically presented as follows. Given a training dataset $\mathcal{D} = \{(\boldsymbol{x}_i, \boldsymbol{y}_i)\}_{i=1}^N$, the attacker manipulates the training process of the model $\mathcal{F}$ by introducing a poisoned subset $\tilde{\mathcal{D}} = \{(\tilde{\boldsymbol{x}}_i, \boldsymbol{y}_t)\}_{i=1}^M$, where $\tilde{\boldsymbol{x}}_i = G(\boldsymbol{x}_i)$ with $G(\cdot)$ as a certain trigger injection function and $\boldsymbol{y}_t$ being the chosen target label, or by altering the training loss directly. During inference, the model behaves normally on benign samples, where $\boldsymbol{y}_j = \mathcal{F}(\boldsymbol{x}_j)$, while exhibiting backdoor behavior on poisoned samples, such as misclassifying to the target label $\boldsymbol{y}_t = \mathcal{F}(G(\boldsymbol{x}_j))$.

Generally, existing attacks can be classified into two types: **(1)** *Visible backdoor attacks*, which typically employ trigger patterns that are visible to humans, such as specific white-black squares (Gu et al., 2019), physical attacks (Li et al., 2021c), or adaptive attacks (Qi et al., 2023). **(2)** *Invisible backdoor attacks*, which introduce imperceptible triggers to enhance the stealth and evasiveness of attacks (Chen et al., 2017), including sample-specific attacks (Li et al., 2021d), trainable noise attacks (Doan et al., 2021), and sample rotation attacks (Xu et al., 2023)).

## 2.2 BACKDOOR DEFENSES

Currently, there are various backdoor defense methods designed to mitigate backdoor threats. These methods can generally be divided into three main paradigms (Li et al., 2024c): **(1)** *pre-processing-based defenses* (Liu et al., 2017; Li et al., 2021c; Shi et al., 2023). **(2)** *backdoor elimination* (Li et al., 2021b; Huang et al., 2022; Xu et al., 2024), which involves adjusting model parameters through fine-tuning, pruning or reconstruction to remove the backdoor. **(3)** *trigger elimination*, also known as testing sample filtering (Gao et al., 2019; Javaheripi et al., 2020; Li et al., 2023b). In this paper, we focus on pre-processing-based defenses since we consider scenarios where only fixed third-party models are accessible and defenders require to obtain the correct final results of all samples.

**Pre-processing-based Defenses.** Generally, pre-processing-based defenses can be categorized into two types: (1) *Transformation-based defenses*. Classical methods (Liu et al., 2017; Li et al., 2021c; Qiu et al., 2021) typically involve applying simple transformations to input, aiming to disrupt trigger patterns and prevent the model from exhibiting backdoor behavior. More Recently, many methods have leveraged the powerful reconstruction capabilities of generative models, such as diffusion models (Shi et al., 2023; May et al., 2023) and masked autoencoders (Sun et al., 2023), intending to retain the original benign features while minimizing the presence of backdoor-related features. However, there is a trade-off between removing backdoor patterns and restoring benign patterns, which remains a pressing issue to address. (2) *BTI-based defenses* (Wang et al., 2019; Xu et al., 2024; Wang et al., 2023), which focus on inverting the pre-injected triggers and utilizing them to purify the input samples. However, these methods may face issues with inaccuracies in the inverted triggers, which may lead to suboptimal purification of the input. How to design an effective pre-processing-based defense is still an important open question.

## 2.3 MODEL REPROGRAMMING

Model reprogramming (Kloberdanz et al., 2021; Neekhara et al., 2022; Jing et al., 2023) is a technique that extends the application of a pre-trained model from a source domain to a target domain. This technique involves adapting the input from the target domain to match that of the source domain. Specifically, model reprogramming introduces an input transformation module $\mathcal{T}(\boldsymbol{x}|\boldsymbol{\theta})$ and an output mapping module $\mathcal{M}(\boldsymbol{y}|\boldsymbol{\beta})$, where $\boldsymbol{\theta}$ and $\boldsymbol{\beta}$ are the trainable parameters of these two modules, respectively (Chen, 2024). Given a pre-trained model $\mathcal{F}(\cdot)$ and an input sample $\boldsymbol{x}$, model reprogramming first transforms $\boldsymbol{x}$ to $\tilde{\boldsymbol{x}}$ leveraging the input transformation module. Then input $\tilde{\boldsymbol{x}}$ into the pre-trained model $\mathcal{F}(\cdot)$ and get the output $\tilde{\boldsymbol{y}} = \mathcal{F}(\tilde{\boldsymbol{x}})$. Finally, the output mapping module is used to map $\tilde{\boldsymbol{y}}$ into the final output $\boldsymbol{y}$. Through fine-tuning the input transformation module and the output mapping module (*i.e.*, optimizing $\boldsymbol{\theta}$ and $\boldsymbol{\beta}$), model reprogramming can efficiently turn the pre-trained model from the source domain to a target domain. Compared to transfer learning, model reprogramming does not necessitate modifying the parameters of the pre-trained model. As such, it is more efficient and flexible. More details about related work are in Appendix H.

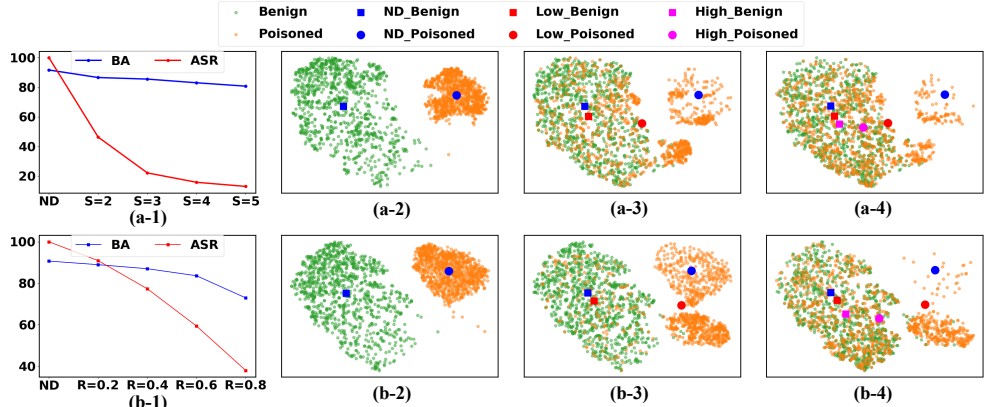

Figure 2: (a-1)&(b-1): The ASR and BA for ShrinkPad (the first row) and BDMAE (the second row) with different transformation intensities. (a-2)~(a-4)&(b-2)~(b-4): The t-SNE plots of the features of benign and backdoor samples under no defense (dubbed "ND"), low transformation intensity (dubbed "Low"), and high transformation intensity (dubbed "High"). Squares and solid circles represent the centroids of benign sample distributions and backdoor sample distributions. As the transformation intensity increases, the features of benign samples deviate from the origin. The results demonstrate the tradeoff faced by the transformation-based backdoor defense methods.

# 3 REVISITING EXISTING PRE-PROCESSING-BASED BACKDOOR DEFENSES

## 3.1 THREAT MODEL

This paper focuses on tackling the issue of pre-trained backdoored models via pre-processing-based backdoor defense. The defender may buy or acquire a pre-trained model from third-party platforms. However, there exists a threat that the pre-trained model is backdoored. Due to the limitations of computational resources, the defender seeks to mitigate the backdoor in an efficient and low-cost way (*e.g.*, without altering the parameters of the pre-trained model). Following prior works (Liu et al., 2017; Li et al., 2021c), we make the following assumptions. For adversaries, they can implant the backdoor into the pre-trained model in any way (*e.g.*, by poisoning the training data or intervening in the training process). For defenders, we assume that they have access to an *unlabeled* dataset that is independent and identically distributed to the training dataset of the pre-trained model.

## 3.2 THE LIMITATIONS OF TRANSFORMATION-BASED DEFENSES

Transformation-based defenses aim to mismatch or eliminate triggers by applying specific transformations to test samples. This type of defense method can be categorized into two types: random perturbations and generator reconstruction. Specifically, random perturbations involve the defender mismatching the trigger pattern through techniques such as scaling or rotation, while generator reconstruction leverages a pre-trained generative model to erase the trigger pattern. However, *the transformation-based backdoor defense methods face a trade-off between the utility of the model and the effectiveness of the backdoor elimination*, making them ineffective in practice.

In this section, we present the empirical results to support the above claim. We implement two representative transformation-based methods, ShrinkPad (Li et al., 2021c) (dubbed "SP") and BD-MAE (Sun et al., 2023) (dubbed "BD"), to defend the BadNets attack (Gu et al., 2019) on CIFAR-10. Specifically, ShrinkPad applies simple spatial transformations to the input, while BDMAE employs a trained masked autoencoder for data cleansing. We use "Pad Size" (dubbed "S"), which refers to the padding size applied around shrunk images, and "Mask Ratio" (dubbed "R"), which represents the masking rate applied to images before reconstruction, to control the transformation intensity for ShrinkPad and BDMAE, respectively. We aim to analyze how these transformations impact the model's benign accuracy (BA) and attack success rate (ASR) of the backdoor. Additionally, we treat the original model as a feature extractor. We then visualize how transformation intensity affects the differences in feature distribution between benign and poisoned samples of the same class.

As shown in Figure 2 (a-1) and (b-1), increasing the intensity of input transformation, which enlarges the feature distance between the original and transformed samples, reduces the backdoor ASR. However, it also leads to a decline in the model's BA. As depicted in Figure 2 (a-2)~(a-4) and (b-2)~(b-4), higher transformation intensity causes greater changes in the feature distribution

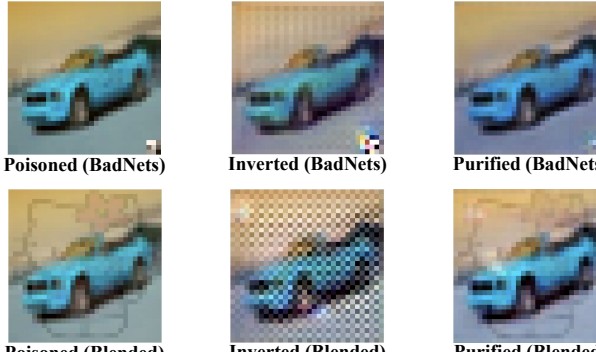

**Figure 3:** The visualization of BTI-DBF in inverting backdoor triggers under both BadNets and Blended attacks. We display the poisoned, inverted, and purified samples, respectively.

of backdoored samples within the same class, indicating that higher transformation levels effectively mismatch or remove trigger patterns. Nevertheless, the difficulty of decoupling benign patterns from backdoor patterns in the input domain results in that such transformations inevitably affect the benign features. It causes a shift in the centroid of the benign sample feature distribution (visualized as solid circles in Figure 2). The primary cause of this trade-off is the consistent output domain of the DNN before and after defenses, which forces the input transformation module to achieve two conflicting goals: **(1)** removing trigger patterns effectively and **(2)** maintaining benign patterns of samples while ensuring their correct classification. This conflict inspires us to consider that adjusting the model's output domain may help mitigate this issue.

### 3.3 The Limitations of BTI-based Defenses

BTI-based defenses can 'break' the trade-off between model utility and defense performance by incorporating the information of backdoor attacks via trigger inversion. In the pre-processing-based defense paradigm, BTI-based defenses typically involve two steps: trigger inversion and input purification. Specifically, the defender first exploits several data to invert the pre-injected trigger, then trains a generator to purify the input samples using the inverted trigger. The effectiveness of BTI-based defenses highly relies on the quality of the inverted trigger. However, we argue that *the inherent challenge of achieving high-quality trigger inversion, due to the lack of prior knowledge, hinders effective input purification*, ultimately limiting the performance of BTI-based defenses.

We implement the state-of-the-art BTI-based defense, BTI-DBF (Xu et al., 2024), to invert the backdoor triggers of BadNets (Gu et al., 2019) and Blended (Chen et al., 2017) on CIFAR-10. As shown in the Figure 3, BTI-DBF effectively reverses the trigger pattern of the BadNets attack and purifies the poisoned sample. However, for the Blended attack, the trigger pattern reversed by BTI-DBF significantly differs from the pre-injected one, leading to poor purification of the poisoned sample. This illustrates that the effectiveness of BTI-based defenses largely depends on the quality of trigger inversion, which is the inherent challenge of such defenses. Moreover, BTI-based defenses often identify "pseudo-triggers" inherent in DNNs (Ya et al., 2023), which usually arise from the model's vulnerability to adversarial perturbations. When defenders attempt to use such triggers to train purification generators, they may disrupt the benign features of the samples, while leaving the backdoor patterns largely unaffected. If the quality and authenticity of the inverted trigger patterns cannot be guaranteed, BTI-based defenses may potentially yield adverse outcomes.

In conclusion, achieving BTI is a challenging endeavor due to the lack of prior knowledge about the implanted backdoor and poisoned samples, highlighting the need for an inversion-free backdoor defense to resolve this trade-off.

## 4 Methodology

### 4.1 Motivation and Inspiration

In Section 3, we empirically evaluate existing pre-processing-based defenses and analyze why they are ineffective. In this section, we present a theoretical analysis and the inspiration to design an effective and efficient backdoor defense method. Given a pre-processing method $\mathcal{T}(\cdot)$ and a pre-trained model $\mathcal{F}(\cdot)$, we have the following theorem.

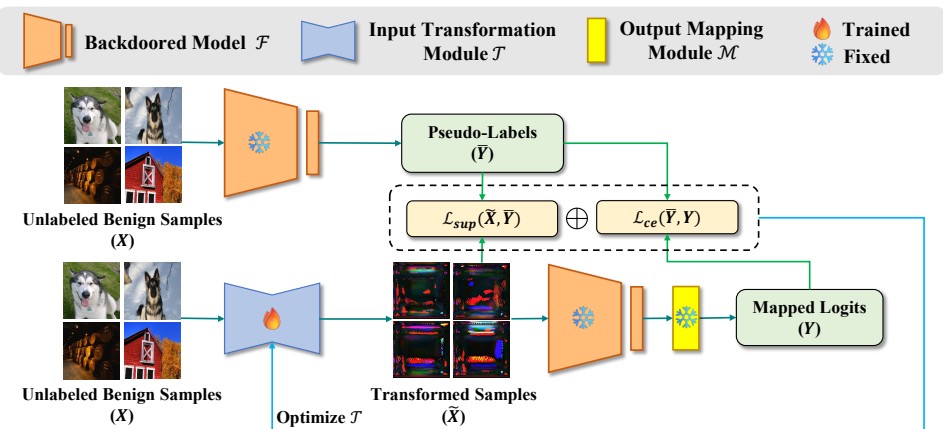

Figure 4: The main optimization pipeline of our REFINE. There are two main components: input transformation module $\mathcal{T}$ and output mapping module $\mathcal{M}$. Specifically, after obtaining the fixed pre-trained model, the defender first specifies a particular hard-coded mapping $\mathcal{M}$ and then optimizes $\mathcal{T}$ guided by the loss function $\mathcal{L}$, using the unlabeled benign dataset. The loss function $\mathcal{L}$ consists of the cross-entropy loss $\mathcal{L}_{ce}$ which aims to maintain the model's utility, and the supervised contrastive loss $\mathcal{L}_{sup}$ to enhance the defense capability via forcing orderly sample aggregation.

**Theorem 1.** *Given a K-class pre-trained deep learning model $\mathcal{F}(\cdot) = s(f(\cdot))$ where $s(\cdot)$ is the softmax function and $f(\cdot)$ is the feature extractor, and a pre-processing method $\mathcal{T}(\cdot)$, $\boldsymbol{x}$ is the data from a specific domain $\mathcal{D}$ (i.e., $\boldsymbol{x} \sim \mathcal{D}$) and $\tilde{\boldsymbol{x}} = \mathcal{T}(\boldsymbol{x}) \sim \tilde{\mathcal{D}}$. Let $\Phi_{\mathcal{D}}(\boldsymbol{x})$ and $\Phi_{\tilde{\mathcal{D}}}(\tilde{\boldsymbol{x}})$ denotes the probability density function of $\mathcal{D}$ and $\tilde{\mathcal{D}}$, we have*

$$\mathbb{E}_{\boldsymbol{x} \sim \mathcal{D}, \tilde{\boldsymbol{x}} \sim \tilde{\mathcal{D}}} \|\mathcal{F}(\boldsymbol{x}) - \mathcal{F}(\tilde{\boldsymbol{x}})\|_2 \leq 2\alpha\sqrt{K} \cdot \mathcal{W}_1(\mu, \tilde{\mu}), \tag{1}$$

*where $\mathcal{W}_1(\mu, \tilde{\mu})$ is the Wasserstein-1 distance between $\mu$ and $\tilde{\mu}$, $\mu$ and $\tilde{\mu}$ are the probability measures of the representations $f(\boldsymbol{x})$ and $f(\tilde{\boldsymbol{x}})$, and $\alpha = \max[\Phi_{\tilde{\mathcal{D}}}(\tilde{\boldsymbol{x}}|\boldsymbol{x})/\Phi_{\tilde{\mathcal{D}}}(\tilde{\boldsymbol{x}})]$.*

Theorem 1 indicates why existing defenses are ineffective. Assuming $\boldsymbol{x}$ is the poisoned sample, the left part of Eq. (1) means the distance between the prediction of the transformed poisoned sample and the original poisoned sample. Theorem 1 demonstrates that the distance is bounded by the Wasserstein-1 distance between the probability measures $\mu, \tilde{\mu}$ of the output representations. Thus, to maintain model utility, existing pre-processing-based defenses tend to retain the output representations, limiting their effectiveness against backdoors. Otherwise, they have to compromise the model utility to achieve greater backdoor defense performance. The proof is in Appendix A.

Following the above theorem, we can enhance the upper bound by increasing the distance between $\mu, \tilde{\mu}$. Inspired by model reprogramming techniques (Chen, 2024), we propose REFINE, a reprogramming-based inversion-free backdoor defense method. Our REFINE can significantly transform the input domain to destroy trigger patterns while maintaining model utility for it also changes the output domain. Specifically, we introduce an input transformation module to modify inputs, and an output mapping module to remap original classes to new shuffled ones. We also employ a supervised contrastive loss to further enlarge the distances among different classes. The technical details of our REFINE are illustrated in the following parts.

## 4.2 REFINE: REPROGRAMMING-BASED INVERSION-FREE BACKDOOR DEFENSE METHOD

In general, REFINE consists of two essential components: **(1)** the input transformation module $\mathcal{T}$, which disrupts the benign and backdoor patterns of input samples through transformations and generates new benign features; **(2)** the label mapping module $\mathcal{M}$, which formulates the specified source-target hard-coded label remapping function and maps the original classes to new shuffled classes. Additionally, we integrate the cross-entropy loss $\mathcal{L}_{ce}$ and the supervised contrastive loss $\mathcal{L}_{sup}$ to steer the optimization of $\mathcal{T}$. The illustration of our REFINE is shown in Figure 4.

### 4.2.1 INPUT TRANSFORMATION MODULE

To effectively alter potential trigger patterns in the input samples, we need to modify the input domain of the original model. Traditional model reprogramming methods (Elsayed et al., 2019; Tsai et al., 2020) add the optimized universal adversarial perturbation around the input samples, while

trigger patterns still remain intact on backdoored images to some extent. In contrast, we utilize a trainable autoencoder (e.g., UNet) as the foundational structure for our input transformation module. Arguably, this module not only preserves the consistency of sample dimension before and after transformation, but also affords greater flexibility in sample manipulation compared to conventional reprogramming methods. Upon inputting a batch of data, the input transformation module will encode the pixel features from the images and then decode them to produce new samples. The transformed samples $\tilde{X}$ can be described as follows:

$$\tilde{X} = \mathcal{T}(X, \theta), \tag{2}$$

where $X$ is a batch of input samples, and $\mathcal{T}(\cdot, \theta)$ is the input transformation module with $\theta$ as its trainable parameters. During this transformation process, both benign and backdoor patterns are disarranged, effectively removing potential triggers and causing the generation of new benign features orderly clustered by their respective classes.

### 4.2.2 OUTPUT MAPPING MODULE

Once the input samples are transformed into new samples via the input transformation module, they are subsequently processed by the original backdoored model, which generates confidence scores for each class, as expressed below:

$$\tilde{Y} = \mathcal{F}(\tilde{X}), \tag{3}$$

where $\mathcal{F}(\cdot)$ is the original backdoored model. As demonstrated in Section 3.2, fixing the model's output domain leads to a trade-off between model utility and defense performance. To address this issue, we introduce an output mapping module at the model's output end, aiming to alter the output domain and mitigate the aforementioned challenges. Specifically, the output mapping module redefines the class order of the model's output layer, which hard-codes a one-to-one label remapping function $f_L : \tilde{l} \mapsto l$, where $\tilde{l}, l \in L, \tilde{l} \neq l$, $L$ is the set of labels. The confidence scores generated by the original model can be remapped into new scores through $\mathcal{M}$, as follows:

$$Y = \mathcal{M}(\tilde{Y}). \tag{4}$$

The final predictions for the samples can be derived from the confidence scores $Y$ outputted by $\mathcal{M}$.

### 4.2.3 OPTIMIZING REFINE MODULES

To maximize the flexibility of input transformations for removing trigger patterns while maintaining the original model's accuracy, we incorporate two crucial loss functions, the cross-entropy loss and the supervised contrastive loss, to guide the optimization of the input transformation module. The formulation of the combined loss function can be expressed as follows:

$$\min_{\theta} \mathcal{L}_{refine} = \mathcal{L}_{ce} + \lambda \mathcal{L}_{sup}. \tag{5}$$

In Eq. (5), $\mathcal{L}_{ce}$ and $\mathcal{L}_{sup}$ indicate the cross-entropy loss and the supervised contrastive loss, respectively. $\lambda$ is a scalar temperature parameter, and $\theta$ represents the set of parameters in the input transformation module to be optimized during training. Since Theorem 1 does not guarantee the model performance on clean samples, adding $\mathcal{L}_{ce}$ to maintain the utility of the model is necessary.

In our threat model, the dataset available to the defender is unlabeled. Therefore, before calculating these loss functions, it is necessary to obtain the pseudo-labels $\bar{Y}$ for the current batch of unlabeled samples $X$, predicted by the original model (without any additional modules), as follows:

$$\bar{Y} = \arg\max(\mathcal{F}(X)). \tag{6}$$

**Leveraging Cross-entropy Loss to Maintain the Utility.** Due to the substantial modification of the original model's output domain facilitated by the output mapping module, the input transformation module is no longer constrained by the requirement to preserve the original benign features of the samples. Nevertheless, the model must retain its original performance within the new output domain, which necessitates the employment of cross-entropy loss to effectively guide the sample transformation process. The cross-entropy loss is typically formalized as follows:

$$\mathcal{L}_{ce} = -\frac{1}{N} \sum_{i=1}^{N} \bar{y}_i \log(y_i), \tag{7}$$

where $N$ represents the number of samples in the current data batch $\boldsymbol{X}$. $\bar{\boldsymbol{y}}_i \in \bar{\boldsymbol{Y}}$ denotes the pseudo-label for sample $\boldsymbol{x}_i \in \boldsymbol{X}$ (typically a one-hot encoded vector), and $\boldsymbol{y}_i \in \boldsymbol{Y}$ indicates the predicted probability remapped by the output mapping module for sample $\boldsymbol{x}_i$.

**Utilizing Supervised Contrastive Loss to Enhance Backdoor Defense.** Arguably, relying solely on cross-entropy loss is insufficient to maintain the original model's benign accuracy and mitigate the backdoor. Therefore, we introduce supervised contrastive loss (Khosla et al., 2020), where "supervised" refers to the original model as the supervisor. Specifically, the supervised contrastive loss aims to ensure that features of transformed samples from the same class are more similar, while those from different classes are further apart. It can be defined as follows.

$$\mathcal{L}_{sup} = \sum_{i \in I} \frac{-1}{|P(i)|} \sum_{p \in P(i)} \log \frac{\exp\left(\tilde{\boldsymbol{x}}_i \cdot \tilde{\boldsymbol{x}}_p / \tau\right)}{\sum_{a \in A(i)} \exp\left(\tilde{\boldsymbol{x}}_i \cdot \tilde{\boldsymbol{x}}_a / \tau\right)}, \tag{8}$$

where $I \equiv \{1, 2, ..., N\}$ represents indices of all samples in current data batch, $\tilde{\boldsymbol{x}}_i = \mathcal{T}(\boldsymbol{x}_i, \theta) \in \tilde{\boldsymbol{X}}$, the $\cdot$ symbol denotes the inner (dot) product, $\tau$ is a scalar temperature parameter, and $A(i) \equiv I \setminus \{i\}$. The set $P(i) \equiv \{p \in A(i) : \bar{\boldsymbol{y}}_p = \bar{\boldsymbol{y}}_i\}$ contain indices of all positives in the batch distinct from $i$, and $|P(i)|$ is its cardinality. The pseudo-code for the optimization process can be found in Appendix B.

### 4.2.4 UTILIZING REFINE FOR MODEL INFERENCE

During the model inference phase, we can apply the aforementioned well-trained modules to achieve high-performance and secure predictions. The input samples are sequentially processed through the input transformation module $\mathcal{T}(\cdot, \theta)$, the original pre-trained model $\mathcal{F}(\cdot)$, and the output mapping module $\mathcal{M}(\cdot)$. This process ultimately yields the predicted confidence scores, with all parameters remaining constant. The inference process can be formally expressed as follows.

$$\boldsymbol{y} = \mathcal{M}(\mathcal{F}(\mathcal{T}(\boldsymbol{x}, \theta))), \tag{9}$$

where $\boldsymbol{x}$ represents the sample to be predicted. The detailed process is illustrated in Figure 1.

## 5 EXPERIMENTS

In this section, we evaluate the effectiveness of our REFINE compared with different existing backdoor defenses. We also conduct an ablation study and evaluate the resistance to potential adaptive attacks. The analysis of the overhead of REFINE is in Appendix F and the implementation of REFINE in the black-box scenario is in Appendix E.

### 5.1 EXPERIMENTAL SETTINGS

**Datasets and Models.** We conduct experiments on two classical benchmark datasets, including CIFAR-10 (Krizhevsky et al., 2009) and (a subset of) ImageNet (Deng et al., 2009) containing 50 classes. We evaluated our method with ResNet-18 (He et al., 2016) on both datasets. We also validate the effectiveness of REFINE on other models in Appendix D. Note that our goal is to evaluate the effectiveness of backdoor defense methods instead of training a SOTA model. Therefore, the benign accuracies of our models may be lower than the SOTA models. We exploit U-Net (Ronneberger et al., 2015) as the structure of the input transformation module.

**Attack Setup.** We utilize 7 representative advanced backdoor attacks, including **(1)** BadNets (Gu et al., 2019), **(2)** Blended (Chen et al., 2017), **(3)** WaNet (Nguyen & Tran, 2021), **(4)** PhysicalBA (dubbed 'Physical') (Li et al., 2021c), **(5)** BATT (Xu et al., 2023), **(6)** LabelConsistent (dubbed 'LC') (Turner et al., 2019), and **(7)** Adaptive-Patch (dubbed 'Adaptive') (Qi et al., 2023), to comprehensively evaluate the performance of different defenses.

**Defense Setup.** We compare the defense performance of REFINE with both types of pre-processing-based defenses. For transformation-based defenses, we utilize three advanced methods, including **(1)** ShrinkPad (Li et al., 2021c), **(2)** BDMAE (Sun et al., 2023), **(3)** ZIP (Shi et al., 2023). For BTI-based defenses, we employ three methods as baseline, including **(1)** Neural Cleanse (dubbed 'NC') (Wang et al., 2019), **(2)** UNICORN (Wang et al., 2023), **(3)** BTI-DBF(P) (Xu et al., 2024).

**Evaluation Metrics.** Consistent with the standard evaluation metrics in backdoor-related studies (Li et al., 2024c), we utilize benign accuracy (BA) and attack success rate (ASR) to assess all defense

Table 1: The performance (%) of REFINE and the transformation-based backdoor defenses. The best results are **boldfaced**, while all failed cases (BA drop or ASR > 10%) are marked in red.

| Dataset | Defense | No Defense | | ShrinkPad | | BDMAE | | ZIP | | **REFINE** | |
|---|---|---|---|---|---|---|---|---|---|---|---|
| | Attack | BA | ASR | BA↑ | ASR↓ | BA↑ | ASR↓ | BA↑ | ASR↓ | BA↑ | ASR↓ |
| CIFAR-10 | BadNets | 91.24 | 100 | 84.51 | 13.37 | 89.53 | 3.18 | 81.95 | 19.06 | **90.43** | 0.78 |
| | Blended | 91.04 | 100 | 83.95 | 5.94 | 89.08 | 84.80 | 81.54 | 3.72 | **89.85** | 1.73 |
| | WaNet | 91.15 | 99.97 | 84.45 | 34.79 | 87.45 | 99.92 | 81.79 | 7.58 | **90.33** | 0.97 |
| | Physical | 93.77 | 99.99 | 90.07 | 13.67 | **93.07** | 3.76 | 78.32 | 25.24 | 90.82 | 1.69 |
| | BATT | 92.48 | 100 | 86.13 | 100 | **91.71** | 100 | 82.27 | 98.89 | 90.54 | 1.21 |
| | LC | 92.12 | 95.95 | 85.87 | 9.61 | 90.18 | 4.62 | 81.93 | 90.08 | **90.97** | 0.80 |
| | Adaptive | 91.34 | 97.17 | 84.42 | 9.34 | 89.14 | 10.30 | 81.65 | 78.49 | **90.30** | 0.81 |
| ImageNet | BadNets | 65.42 | 99.55 | 61.44 | 2.65 | 53.64 | 3.51 | 59.12 | 11.43 | **66.27** | 1.81 |
| | Blended | 66.15 | 98.93 | 59.84 | 24.53 | 54.80 | 96.08 | 59.32 | 92.98 | **66.59** | 1.11 |
| | WaNet | 67.11 | 98.81 | 59.44 | 40.12 | 52.12 | 94.82 | 57.92 | 0.82 | 66.23 | 1.36 |
| | Physical | 71.64 | 99.80 | **71.80** | 56.73 | 58.40 | 8.98 | 64.00 | 17.63 | 67.11 | 1.97 |
| | BATT | 67.76 | 100 | **70.40** | 100 | 58.88 | 100 | 65.92 | 98.57 | 66.19 | 2.71 |
| | LC | 67.44 | 80.96 | 61.48 | 0.86 | 54.84 | 10.65 | 60.24 | 77.02 | 66.43 | **0** |
| | Adaptive | 66.76 | 93.20 | 62.44 | 6.37 | 56.16 | 69.40 | 61.28 | 94.57 | 66.99 | 1.48 |

Table 2: The performance (%) of REFINE and the BTI-based backdoor defenses. The best results are **boldfaced**, while all failed cases (BA drop or ASR > 10%) are marked in red.

| Dataset | Defense | No Defense | | NC | | UNICORN | | BTI-DBF(P) | | **REFINE** | |
|---|---|---|---|---|---|---|---|---|---|---|---|
| | Attack | BA | ASR | BA↑ | ASR↓ | BA↑ | ASR↓ | BA↑ | ASR↓ | BA↑ | ASR↓ |
| CIFAR-10 | BadNets | 91.24 | 100 | 76.44 | 37.40 | 86.08 | 22.96 | 89.14 | 5.60 | **90.43** | 0.78 |
| | Blended | 91.04 | 100 | 87.28 | 89.47 | 84.09 | 53.46 | 87.44 | 61.52 | **89.85** | 1.73 |
| | WaNet | 91.15 | 99.97 | 83.81 | 6.24 | 85.59 | 6.67 | 88.64 | 4.37 | **90.33** | 0.97 |
| | Physical | 93.77 | 99.99 | 89.00 | 52.36 | 90.04 | 41.04 | **91.74** | 9.53 | 90.82 | 1.69 |
| | BATT | 92.48 | 100 | 72.28 | 5.88 | 74.99 | **0.87** | 89.89 | 4.80 | **90.54** | 1.21 |
| | LC | 92.12 | 95.95 | 83.60 | 38.66 | 75.10 | 1.89 | 89.52 | 88.92 | **90.97** | 0.80 |
| | Adaptive | 91.34 | 97.17 | 85.45 | 31.49 | 68.79 | 9.59 | 88.94 | 45.99 | **90.30** | 0.81 |
| ImageNet | BadNets | 65.42 | 99.55 | 63.44 | 61.93 | 51.96 | 88.90 | 64.32 | 6.17 | **66.27** | 1.81 |
| | Blended | 66.15 | 98.93 | 62.68 | 96.90 | 60.64 | 98.00 | 65.44 | 97.67 | **66.59** | 1.11 |
| | WaNet | 67.11 | 98.81 | 62.20 | 91.80 | 61.64 | 94.86 | 65.56 | 92.17 | **66.23** | 1.36 |
| | Physical | 71.64 | 99.80 | 71.00 | 98.74 | 67.00 | 56.26 | **73.60** | 7.60 | 67.11 | 1.97 |
| | BATT | 67.76 | 100 | 62.92 | **0.65** | 68.56 | 41.86 | **72.00** | 6.94 | 66.19 | 2.71 |
| | LC | 67.44 | 80.96 | 62.88 | 66.99 | 58.92 | 28.52 | 65.96 | 73.81 | **66.43** | **0** |
| | Adaptive | 66.76 | 93.20 | 62.80 | 91.92 | 61.04 | 90.13 | **67.32** | 93.35 | 66.99 | **1.48** |

methods. BA and ASR are the accuracies of the benign samples and the poisoned samples, respectively. An effective defense is indicated by a higher BA and a lower ASR.

## 5.2 MAIN RESULTS

As shown in Tables 1-2, our REFINE successfully mitigates backdoor threats in all cases while preserving high benign accuracy. Specifically, the ASRs of our method are lower than 3% (< 2% in most cases). For the BA, the models under REFINE experience less than 3% drop on the CIFAR-10 dataset compared to the undefended models. On the ImageNet dataset, the BA even improves, due to the increased depth of the original models introduced by the input transformation module. In contrast, other baseline defenses may fail in certain cases, with BA drop or ASR > 10%.

## 5.3 ABLATION STUDY

There are three important components in our methods, including (**1**) input transformation method, (**2**) hard-coded remapping function (HRF for short) in the output mapping module, and (**3**) supervised contrastive loss (SCL for short) of transformed samples. In this section, we present an ablation study on the former two modules and verify their effectiveness. We also test different architectures of the input transformation module and conduct additional ablation studies in Appendix D.

As shown in Table 3, we evaluate the defense performance of REFINE without the hard-coded remapping function (w/o HRF) or without the supervised contrastive loss (w/o SCL). Experimental results indicate that without the hard-coded remapping function, REFINE successfully preserves the BA of the original model, but struggles to reduce the ASR of the backdoor. This is because, without the hard-coded remapping function, the output domain of the model remains unchanged. Subsequently, it encounters the same trade-off problem as other transformation-based defenses, and is difficult to find a balance between transformation intensity and defense performance. Also, in

Table 3: The performance (%) of REFINE with/without the hard-coded remapping function (HRF) or with/without the supervised contrastive loss (SCL).

| Defense | No Defense | | w/o HRF | | w/o SCL | | **REFINE** | |
|---|---|---|---|---|---|---|---|---|
| Attack | BA | ASR | BA↑ | ASR↓ | BA↑ | ASR↓ | BA↑ | ASR↓ |
| BadNets | 91.70 | 100 | 91.23 | 70.76 | 89.26 | 1.43 | 90.92 | 0.68 |
| Blended | 91.10 | 98.76 | 90.59 | 75.30 | 90.38 | 0.10 | 90.65 | 0.51 |
| WaNet | 91.09 | 99.98 | 91.03 | 99.53 | 89.08 | 1.45 | 90.45 | 0.88 |
| Physical | 93.59 | 100 | 92.86 | 1.60 | 88.63 | 1.97 | 90.92 | 1.36 |
| BATT | 92.43 | 99.91 | 91.67 | 72.46 | 88.82 | 5.87 | 90.89 | 1.97 |
| LC | 92.30 | 99.74 | 91.88 | 69.15 | 90.37 | 0.59 | 90.57 | 1.25 |
| Adaptive | 90.54 | 100 | 89.77 | 62.94 | 88.06 | 0.32 | 90.17 | 0.27 |

Table 4: The performance (%) of REFINE against potential adaptive attacks.

| Setting | Normal Attack | | | | Adaptive Attack | | | |
|---|---|---|---|---|---|---|---|---|
| Defense | No Defense | | **REFINE** | | No Defense | | **REFINE** | |
| Dataset | BA | ASR | BA↑ | ASR↓ | BA | ASR | BA↑ | ASR↓ |
| CIFAR-10 | 91.74 | 100 | 90.71 | 1.07 | 84.53 | 100 | 83.05 | 0.98 |
| ImageNet | 66.94 | 99.59 | 69.00 | 0.70 | 58.39 | 100 | 60.53 | 1.09 |

the absence of supervised contrastive loss, REFINE can effectively reduce ASR with the help of the hard-coded remapping function. However, it encounters difficulties in restoring the BA of the original model, which may adversely affect the model's inference capabilities.

## 5.4 RESISTANCE TO POTENTIAL ADAPTIVE ATTACKS

In this section, we examine whether the adversary can circumvent our defenses if they have full knowledge of the process of our REFINE. After training the original backdoored model, the adversary can fine-tune it utilizing an input transformation module, along with a randomly initialized hard-coded output mapping module, to simulate our REFINE. During fine-tuning, the loss function for model optimization can be expressed as follows:

$$\min_{\delta} \mathcal{L}_{adap} = \mathcal{L}_b + \gamma \mathcal{L}_{refine}, \tag{10}$$

where $\mathcal{L}_b$ indicates the cross-entropy loss function in the original training phase of the backdoored model, and $\mathcal{L}_{refine}$ represents the loss function of REFINE. $\gamma$ is a scalar temperature parameter, and $\delta$ denotes the trainable parameters of the backdoored model. Ideally, the adversary can achieve the backdoor target with a low value of $\mathcal{L}_{refine}$ by optimizing Eq. (10). Consequently, the REFINE may not work well since the $\mathcal{L}_{refine}$ is already low.

As shown in Table 4, REFINE is still highly effective with high BAs (BA drop $< 1.5\%$) and low ASRs ($< 1.5\%$). It is mostly because defenders can arbitrarily specify the output mapping function and train an input transformation module that may entirely differ from the attacker's. Besides, the original backdoored model experiences a decrease in BA after undergoing adaptive attack training, due to the inherent difficulty of optimizing multiple loss functions simultaneously. As such, these results demonstrate that our REFINE is resistant to adaptive attacks.

## 6 CONCLUSION

In this paper, we revisited existing pre-processing-based backdoor defense methods, including backdoor-trigger-inversion-based (BTI-based) defenses and transformation-based defenses. We revealed the limitations of the two defense methods. Subsequently, according to the empirical and theoretical analysis, we proposed REFINE, a reprogramming-based inversion-free backdoor defense method. This method was motivated by the insight that increasing the distances of the feature representations before and after the transformation may lead to a better effectiveness of backdoor defense. Specifically, we introduced an input transformation module and an output mapping module. We also utilized the supervised contrastive loss to enhance the defense performance. Results on benchmark datasets verified the effectiveness of our REFINE and the resistance to the adaptive attack. We hope our REFINE can provide a new angle to facilitate the design of more effective backdoor defenses.

## ACKNOWLEDGEMENTS

This research is supported in part by the National Key Research and Development Program of China under Grant 2021YFB3100300 and the National Natural Science Foundation of China under Grants (62441238, 62072395, and U20A20178). Pin-Yu Chen is not supported by any external fundings. This work was mostly done when Yiming Li was a Research Professor at the State Key Laboratory of Blockchain and Data Security, Zhejiang University, China. He is currently at College of Computing and Data Science, Nanyang Technological University, Singapore.

## ETHICS STATEMENT

This paper proposes an inversion-free backdoor defense method, REFINE. Our method can be utilized to mitigate the effect of the backdoor. Therefore, our REFINE is a defensive method and our work does not discover any new threat. Our research also does not include any human subjects. Accordingly, this paper does not raise ethical issues.

## REPRODUCIBILITY STATEMENT

The details of our implementations and experiments can be found in Appendix C. We provide the official implementation of REFINE at `https://github.com/WhitolfChen/REFINE`. Additionally, we also integrate REFINE into `BackdoorBox` for easy access and usage.

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

## APPENDIX

## A  THE PROOF OF THEOREM 1

**Theorem 1.** *Given a $K$-class pre-trained deep learning model $\mathcal{F}(\cdot) = s(f(\cdot))$ where $s(\cdot)$ is the softmax function and $f(\cdot)$ is the feature extractor, and a pre-processing method $\mathcal{T}(\cdot)$, $\boldsymbol{x}$ is the data from a specific domain $\mathcal{D}$ (i.e., $\boldsymbol{x} \sim \mathcal{D}$) and $\tilde{\boldsymbol{x}} = \mathcal{T}(\boldsymbol{x}) \sim \tilde{\mathcal{D}}$. Let $\Phi_{\mathcal{D}}(\boldsymbol{x})$ and $\Phi_{\tilde{\mathcal{D}}}(\tilde{\boldsymbol{x}})$ denotes the probability density function of $\mathcal{D}$ and $\tilde{\mathcal{D}}$, we have*

$$\mathbb{E}_{\boldsymbol{x} \sim \mathcal{D}, \tilde{\boldsymbol{x}} \sim \tilde{\mathcal{D}}} \|\mathcal{F}(\boldsymbol{x}) - \mathcal{F}(\tilde{\boldsymbol{x}})\|_2 \leq 2\alpha\sqrt{K} \cdot \mathcal{W}_1(\mu, \tilde{\mu}), \tag{1}$$

*where $\mathcal{W}_1(\mu, \tilde{\mu})$ is the Wasserstein-1 distance between $\mu$ and $\tilde{\mu}$, $\mu$ and $\tilde{\mu}$ are the probability measures of the representations $f(\boldsymbol{x})$ and $f(\tilde{\boldsymbol{x}})$, and $\alpha = \max[\Phi_{\tilde{\mathcal{D}}}(\tilde{\boldsymbol{x}}|\boldsymbol{x})/\Phi_{\tilde{\mathcal{D}}}(\tilde{\boldsymbol{x}})]$.*

Following similar approaches in (Yang et al., 2021), the proof of Theorem 1 is as follows.

*Proof.* Let $[K]$ represents the set of the first $K$ positive integers, *i.e.*, $[K] = \{1, 2, 3, ..., K\}$. According to the definition of mathematical expectation, we have

$$
\begin{aligned}
&\mathbb{E}_{\boldsymbol{x} \sim \mathcal{D}, \tilde{\boldsymbol{x}} \sim \tilde{\mathcal{D}}} \|\mathcal{F}(\boldsymbol{x}) - \mathcal{F}(\tilde{\boldsymbol{x}})\|_2 \\
&= \int_{\boldsymbol{x} \sim \mathcal{D}, \tilde{\boldsymbol{x}} \sim \tilde{\mathcal{D}}} \|\mathcal{F}(\boldsymbol{x}) - \mathcal{F}(\tilde{\boldsymbol{x}})\|_2 \Phi_{\mathcal{D}, \tilde{\mathcal{D}}}(\boldsymbol{x}, \tilde{\boldsymbol{x}}) d\boldsymbol{x} d\tilde{\boldsymbol{x}} \\
&= \int_{\boldsymbol{x} \sim \mathcal{D}, \tilde{\boldsymbol{x}} \sim \tilde{\mathcal{D}}} \|\mathcal{F}(\boldsymbol{x}) - \mathcal{F}(\tilde{\boldsymbol{x}})\|_2 \Phi_{\mathcal{D}}(\boldsymbol{x}) \Phi_{\tilde{\mathcal{D}}}(\tilde{\boldsymbol{x}}|\boldsymbol{x}) d\boldsymbol{x} d\tilde{\boldsymbol{x}} \\
&\leq \alpha \int_{\boldsymbol{x} \sim \mathcal{D}, \tilde{\boldsymbol{x}} \sim \tilde{\mathcal{D}}} \|\mathcal{F}(\boldsymbol{x}) - \mathcal{F}(\tilde{\boldsymbol{x}})\|_2 \Phi_{\mathcal{D}}(\boldsymbol{x}) \Phi_{\tilde{\mathcal{D}}}(\tilde{\boldsymbol{x}}) d\boldsymbol{x} d\tilde{\boldsymbol{x}},
\end{aligned} \tag{2}
$$

where $\alpha = \max[\Phi_{\tilde{\mathcal{D}}}(\tilde{\boldsymbol{x}}|\boldsymbol{x})/\Phi_{\tilde{\mathcal{D}}}(\tilde{\boldsymbol{x}})]$. Assuming $\boldsymbol{x} \in \mathbb{R}^d$ is a $d$-dimension vector and $\boldsymbol{x}_i$ denotes the $i$-th element of $\boldsymbol{x}$, we have

$$\|\boldsymbol{x}\| = \sqrt{\sum_{i=1}^{d} \boldsymbol{x}_i^2} \leq \sqrt{d \cdot \max_{i \in [d]}[\boldsymbol{x}_i^2]} = \sqrt{d} \cdot \max_{i \in [d]}[|\boldsymbol{x}_i|]. \tag{3}$$

Since $\mathcal{F}(\cdot)$ is a $K$-class pre-trained model, we have

$$
\begin{aligned}
&\alpha \int_{\boldsymbol{x} \sim \mathcal{D}, \tilde{\boldsymbol{x}} \sim \tilde{\mathcal{D}}} \|\mathcal{F}(\boldsymbol{x}) - \mathcal{F}(\tilde{\boldsymbol{x}})\|_2 \Phi_{\mathcal{D}}(x) \Phi_{\tilde{\mathcal{D}}}(\tilde{\boldsymbol{x}}) d\boldsymbol{x} d\tilde{\boldsymbol{x}} \\
&\leq \alpha\sqrt{K} \int_{\boldsymbol{x} \sim \mathcal{D}, \tilde{\boldsymbol{x}} \sim \tilde{\mathcal{D}}} \max_{k \in [K]} |[\mathcal{F}(\boldsymbol{x})]_k - [\mathcal{F}(\tilde{\boldsymbol{x}})]_k| \cdot \Phi_{\mathcal{D}}(\boldsymbol{x}) \Phi_{\tilde{\mathcal{D}}}(\tilde{\boldsymbol{x}}) d\boldsymbol{x} d\tilde{\boldsymbol{x}} \\
&= \alpha\sqrt{K} \int_{\boldsymbol{x} \sim \mathcal{D}, \tilde{\boldsymbol{x}} \sim \tilde{\mathcal{D}}} \max_{k \in [K]} |[s(f(\boldsymbol{x}))]_k - [s(f(\tilde{\boldsymbol{x}}))]_k| \cdot \Phi_{\mathcal{D}}(\boldsymbol{x}) \Phi_{\tilde{\mathcal{D}}}(\tilde{\boldsymbol{x}}) d\boldsymbol{x} d\tilde{\boldsymbol{x}}.
\end{aligned} \tag{4}
$$

After that, we define $k^+$ and $k^-$ as the following equations.

$$\begin{cases} k^+ = \arg\max_{k \in [K]} [s(f(\boldsymbol{x}))]_k - [s(f(\tilde{\boldsymbol{x}}))]_k \\ k^- = \arg\max_{k \in [K]} [s(f(\tilde{\boldsymbol{x}}))]_k - [s(f(\boldsymbol{x}))]_k \end{cases}. \tag{5}$$

Because the output of $s(\cdot)$ is a probability logit and the sum total is 1, there exist at least one $k_1$ such that $[s(f(\boldsymbol{x}))]_{k_1} - [s(f(\tilde{\boldsymbol{x}}))]_{k_1} \geq 0$ and also at least one $k_2$ leading to $[s(f(\tilde{\boldsymbol{x}}))]_{k_2} - [s(f(\boldsymbol{x}))]_{k_2} \geq 0$. Therefore,

$$
\begin{aligned}
&\max_{k \in [K]} |[s(f(\boldsymbol{x}))]_k - [s(f(\tilde{\boldsymbol{x}}))]_k| \\
&= \max_{k \in [K]} \{[s(f(\boldsymbol{x}))]_k - [s(f(\tilde{\boldsymbol{x}}))]_k, [s(f(\tilde{\boldsymbol{x}}))]_k - [s(f(\boldsymbol{x}))]_k \\
&\leq [s(f(\boldsymbol{x}))]_{k^+} - [s(f(\tilde{\boldsymbol{x}}))]_{k^+} + [s(f(\tilde{\boldsymbol{x}}))]_{k^-} - [s(f(\boldsymbol{x}))]_{k^-}.
\end{aligned} \tag{6}
$$

According to Eq. (6), we have

$$\int_{\boldsymbol{x}\sim\mathcal{D},\tilde{\boldsymbol{x}}\sim\tilde{\mathcal{D}}} \max_{k\in[K]} |[s(f(\boldsymbol{x}))]_k - [s(f(\tilde{\boldsymbol{x}}))]_k| \cdot \Phi_{\mathcal{D}}(\boldsymbol{x})\Phi_{\tilde{\mathcal{D}}}(\tilde{\boldsymbol{x}})d\boldsymbol{x}d\tilde{\boldsymbol{x}}$$

$$\leq \int_{\boldsymbol{x}\sim\mathcal{D},\tilde{\boldsymbol{x}}\sim\tilde{\mathcal{D}}}([s(f(\boldsymbol{x}))]_{k^+} - [s(f(\tilde{\boldsymbol{x}}))]_{k^+} + [s(f(\tilde{\boldsymbol{x}}))]_{k^-} - [s(f(\boldsymbol{x}))]_{k^-}) \cdot \Phi_{\mathcal{D}}(\boldsymbol{x})\Phi_{\tilde{\mathcal{D}}}(\tilde{\boldsymbol{x}})d\boldsymbol{x}d\tilde{\boldsymbol{x}}$$

$$= \mathbb{E}_{\boldsymbol{x}\sim\mathcal{D}}[[s(f(\boldsymbol{x}))]_{k^+} - [s(f(\boldsymbol{x}))]_{k^-}] - \mathbb{E}_{\tilde{\boldsymbol{x}}\sim\tilde{\mathcal{D}}}[[s(f(\tilde{\boldsymbol{x}}))]_{k^-} - [s(f(\tilde{\boldsymbol{x}}))]_{k^+}. \tag{7}$$

Based on the fact that $[s(\cdot)]_k$ is 1-Lipschitz continuous for any $k \in [K]$ (Gao & Pavel, 2017) and thus $[s(\cdot)]_{k^+} - [s(\cdot)]_{k^-}$ is 2-Lipschitz continuous, we have

$$\mathbb{E}_{\boldsymbol{x}\sim\mathcal{D}}[[s(f(\boldsymbol{x}))]_{k^+} - [s(f(\boldsymbol{x}))]_{k^-}] - \mathbb{E}_{\tilde{\boldsymbol{x}}\sim\tilde{\mathcal{D}}}[[s(f(\tilde{\boldsymbol{x}}))]_{k^-} - [s(f(\tilde{\boldsymbol{x}}))]_{k^+}$$

$$\leq 2 \cdot \sup_{g:\mathbb{R}^K \mapsto \mathbb{R}, \mathrm{Lip}(g)\leq 1} \mathbb{E}_{\boldsymbol{x}\sim\mathcal{D}}[g(f(\boldsymbol{x}))] - \mathbb{E}_{\tilde{\boldsymbol{x}}\sim\tilde{\mathcal{D}}}[g(f(\tilde{\boldsymbol{x}}))]. \tag{8}$$

Following the Kantorovich-Rubinstein theorem of the dual representation of the Wasserstein-1 distance, finally, we have

$$\mathbb{E}_{\boldsymbol{x}\sim\mathcal{D},\tilde{\boldsymbol{x}}\sim\tilde{\mathcal{D}}}\|\mathcal{F}(\boldsymbol{x}) - \mathcal{F}(\tilde{\boldsymbol{x}})\|_2$$

$$\leq 2\alpha\sqrt{K} \cdot \sup_{g:\mathbb{R}^K \mapsto \mathbb{R}, \mathrm{Lip}(g)\leq 1} \mathbb{E}_{\boldsymbol{x}\sim\mathcal{D}}[g(f(\boldsymbol{x}))] - \mathbb{E}_{\tilde{\boldsymbol{x}}\sim\tilde{\mathcal{D}}}[g(f(\tilde{\boldsymbol{x}}))]$$

$$= 2\alpha\sqrt{K} \cdot \mathcal{W}_1(\mu, \tilde{\mu}), \tag{9}$$

where $\mu$ and $\tilde{\mu}$ are the probability measures of the representations $f(\boldsymbol{x})$ and $f(\tilde{\boldsymbol{x}})$. $\qquad\square$

## B  THE PSEUDO-CODE OF REFINE

The pseudo-code of our REFINE optimization process is shown in Algorithm 1.

---
**Algorithm 1** REFINE Optimization Process

---
**Input:** The backdoored model $\mathcal{F}$, the unlabeled benign dataset $\boldsymbol{D} = \{\boldsymbol{x}_i\}_{i=1}^M$, the randomly initialized input transformation module $\mathcal{T}(\cdot, \theta)$, the specified output mapping module $\mathcal{M}(\cdot)$.
**Output:** The input transformation module parameters $\boldsymbol{\theta}$.

1: **for** data batches $\boldsymbol{X} = \{\boldsymbol{x}_i\}_{i=1}^N$ in $\boldsymbol{D}$ **do**
2:      Obtain the transformed input $\tilde{\boldsymbol{X}} = \mathcal{T}(\boldsymbol{X}, \theta)$.
3:      Obtain the original model output $\tilde{\boldsymbol{Y}} = \mathcal{F}(\tilde{\boldsymbol{X}})$.
4:      Obtain the mapped output $\boldsymbol{Y} = \mathcal{M}(\tilde{\boldsymbol{Y}})$.
5:      Obtain the predicted labels $\bar{\boldsymbol{Y}} = \arg\max(\mathcal{F}(\boldsymbol{X}))$.
6:      Compute the supervised contrastive loss $\mathcal{L}_{sup}(\tilde{\boldsymbol{X}}, \bar{\boldsymbol{Y}})$.
7:      Compute the cross-entropy loss $\mathcal{L}_{ce}(\bar{\boldsymbol{Y}}, \boldsymbol{Y})$.
8:      Optimize $\boldsymbol{\theta}$ with the composite loss: $\arg\min_{\boldsymbol{\theta}} \mathcal{L} = \mathcal{L}_{ce} + \lambda\mathcal{L}_{sup}$.

9: **return** $\boldsymbol{\theta}$

---

## C  IMPLEMENTATION DETAILS

### C.1  DETAILS OF THE EXPERIMENTAL SETTINGS

**Details of Datasets. (1)** *CIFAR-10.* The CIFAR-10 dataset (Krizhevsky et al., 2009) contains 50,000 training samples and 10,000 testing samples in total. The dataset has 10 classes and each class has 5,000 training samples and 1,000 testing samples. Tbe size of each image sample is $3\times32\times32$. **(2)** *ImageNet.* The ImageNet dataset (Deng et al., 2009) consists of 1,000 classes containing over 14 million manually annotated images. In this paper, we select a subset with 50 different classes and each class contains 500 training samples and 100 testing samples with size $3\times224\times224$.

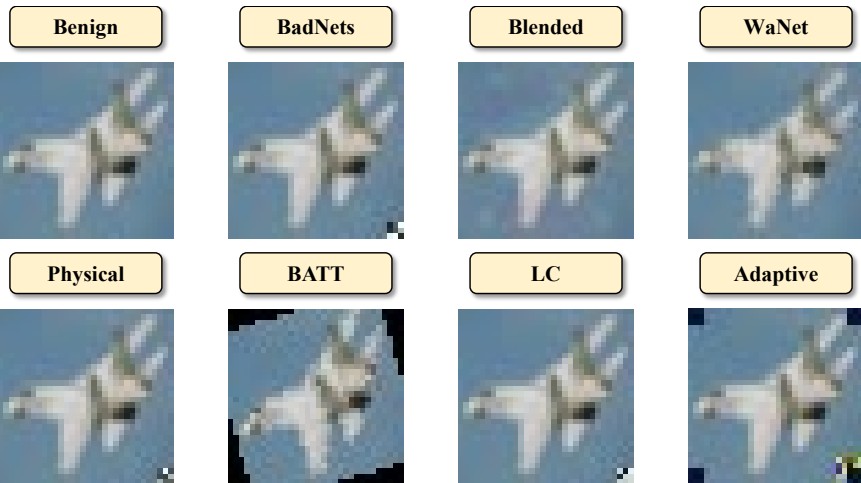

Figure 5: The illustration of the adopted backdoor attacks.

**Details of Training Backdoored Models.** We utilize the SGD with a momentum of 0.9 and a weight decay of $5 \times 10^{-4}$ as the optimizer for training all backdoored DNNs. The batch size is set to 128 on both of CIFAR-10 and ImageNet. We set the initial learning rate as 0.1 and train all models for 150 epochs, with the learning rate reduced by a factor of 0.1 at the 100-th and 130-th epoch.

**Details of Optimization.** For training the input transformation module, we employ SGD with a momentum of 0.9 and a weight decay of $5 \times 10^{-4}$ as the optimizer. The initial learning rate is set to 0.01, and the batch size is set to 256 for CIFAR-10 and 64 for ImageNet. The input transformation module is trained for 150 epochs, with the learning rate decayed by a factor of 0.8 at the 100-th and 130-th epochs. For the training loss function, we set the temperature parameter as 0.1. For the output mapping module, we randomly assign a hard-coded remapping function before each defense.

**Computational Resources.** In our implementations, we utilize PyTorch as the deep learning framework. All our experiments are implemented with RTX 3090 GPUs.

## C.2 DETAILS OF THE ADOPTED BACKDOOR ATTACKS

In our experiments, we adopt 7 representative backdoor attacks to evaluate the defense performance of our REFINE and other baseline backdoor defense methods. We implement all 7 backdoor attacks using BackdoorBox (Li et al., 2024c). We hereby provide a detailed introduction to these backdoor attacks, as follows.

- **BadNets**: Gu et al. (2019) introduced the earliest poisoning-based backdoor attack that aims to poison the training dataset using a visible, distinctive pixel pattern. In this paper, we utilize a $3 \times 3$ random square as the trigger pattern on the bottom right of samples in CIFAR-10 and a $20 \times 20$ square on ImageNet. The poisoning rate is set to 0.1.

- **Blended**: To evade human visual detection of poisoned samples, Chen et al. (2017) designed a covert data poisoning method known as Blended, which attempts to embed triggers implicitly within the samples. In this paper, we utilize an image of Hello Kitty as the trigger pattern and set the blending rate and poisoning rate to 0.1 across both datasets.

- **WaNet**: WaNet (Nguyen & Tran, 2021) is another type of invisible attacks that employs a warp-based trigger. We follow its default settings and set the poisoning rate to 0.1.

- **PhysicalBA**: Li et al. (2021c) demonstrated that DNNs applied in physical scenarios could also be vulnerable to backdoor threats and proposed backdoor attacks that simulate physical transformations. We follow its default settings and set the poisoning rate to 0.1.

- **BATT**: Xu et al. (2023) noted that simple transformations specific to samples can pose significant backdoor threats to models and introduced the Backdoor Attack with Transformation-based Triggers (BATT). We follow its default settings and set the poisoning rate to 0.1.

Table 5: The performance (%) of REFINE and two baseline defenses on different model architectures. The best results are **boldfaced**.

| Model | Defense
Attack | No Defense | | BDMAE | | BTI-DBF(P) | | REFINE | |
|---|---|---|---|---|---|---|---|---|---|
| | | BA | ASR | BA↑ | ASR↓ | BA↑ | ASR↓ | BA↑ | ASR↓ |
| VGG16 | BadNets | 86.47 | 99.53 | 84.87 | 2.83 | 84.95 | 4.44 | **87.62** | **1.47** |
| | Blended | 86.45 | 98.70 | 84.29 | 82.95 | 83.90 | 36.97 | **87.34** | **2.30** |
| | WaNet | 87.12 | 99.77 | 84.33 | 97.34 | 84.71 | 2.91 | **87.57** | **1.50** |
| | Physical | 87.85 | 99.98 | 88.79 | 6.15 | 88.09 | 9.15 | **89.62** | **1.69** |
| | BATT | 88.40 | 99.99 | 87.28 | 99.99 | 86.07 | 13.13 | **89.24** | **0.81** |
| | LC | 87.46 | 72.87 | 85.24 | 10.19 | 85.26 | 42.06 | **88.60** | **1.77** |
| | Adaptive | 86.49 | 87.57 | 83.76 | 27.17 | 83.94 | 33.05 | **87.70** | **2.30** |
| DenseNet121 | BadNets | 86.36 | 99.99 | 84.81 | 1.44 | 85.04 | 2.76 | **88.51** | **0.96** |
| | Blended | 86.90 | 99.90 | 84.92 | 51.03 | 84.74 | 74.51 | **88.58** | **1.09** |
| | WaNet | 86.34 | 99.34 | 83.92 | 96.93 | 84.29 | **0.86** | **88.04** | 1.72 |
| | Physical | 86.69 | 97.89 | 85.66 | 3.64 | 83.96 | 9.20 | **87.33** | **1.84** |
| | BATT | 86.45 | 100 | 85.63 | 100 | 84.04 | 3.63 | **88.45** | **0.11** |
| | LC | 86.79 | 57.72 | 85.42 | 12.01 | 84.88 | 21.98 | **88.82** | **1.23** |
| | Adaptive | 86.70 | 54.19 | 85.07 | 25.05 | 84.83 | 25.33 | **88.74** | **1.10** |
| ViT | BadNets | 66.98 | 99.97 | 64.16 | 6.13 | 64.18 | 6.84 | **75.97** | **1.89** |
| | Blended | 66.01 | 99.57 | 63.30 | 96.17 | 63.74 | 98.67 | **76.23** | **2.23** |
| | WaNet | 66.17 | 98.03 | 61.44 | 43.71 | 63.63 | 4.17 | **76.15** | **2.51** |
| | Physical | 67.09 | 99.92 | 63.91 | 6.25 | 62.66 | 8.00 | **76.28** | **2.40** |
| | BATT | 68.66 | 99.99 | 67.01 | 99.97 | 66.33 | 98.73 | **72.75** | **3.35** |
| | LC | 67.39 | 92.25 | 63.75 | 5.85 | 64.28 | 74.16 | **80.36** | **1.60** |
| | Adaptive | 66.38 | 83.01 | 63.85 | 14.67 | 64.48 | 45.72 | **77.16** | **2.51** |

- **Label-consistent Attack (LC)**: To address the issue of easily identifiable mislabeled poisoned data in poisoning datasets, Turner et al. (2019) proposed clean-label backdoor attacks, which aim to poison samples of specific classes to inject backdoors. We employ projected gradient descent (PGD) to generate adversarial samples, setting the maximum perturbation size to $\epsilon = 8$. The trigger patterns utilized are identical to those employed in BadNets. The poisoning rate is set to 0.25 on the CIFAR-10 dataset and 1.0 on the ImageNet dataset.

- **Adaptive-Patch**: Qi et al. (2023) observed that models trained on poisoned datasets often learn distinct latent representations for poisoned and clean samples, and they proposed adaptive backdoor attacks to mitigate this separation phenomenon. In this paper, we follow the default settings utilized in its original paper. For CIFAR-10, the poisoned rate and covered rate are set to 0.01 and 0.02, respectively; for ImageNet, they are set to 0.03 and 0.06, respectively.

The poisoned samples of these backdoor attacks are depicted in Figure 5.

### C.3 DETAILS OF THE ADOPTED BACKDOOR DEFENSES

In our experiments, we compare REFINE with two types of pre-processing-based defense methods, namely transformation-based defenses and BTI-based defenses. Each type of defense includes three specific baseline defenses. Specifically, for transformation-based defenses, we utilize three representative and advanced methods, including **(1)** ShrinkPad (Li et al., 2021c), **(2)** BDMAE (Sun et al., 2023), **(3)** ZIP (Shi et al., 2023). We implement this type of defenses using their open-source codes. For BTI-based defenses, we employ three methods as baseline, including **(1)** Neural Cleanse (dubbed 'NC') (Wang et al., 2019), **(2)** UNICORN (Wang et al., 2023), **(3)** BTI-DBF(P) (Xu et al., 2024). For NC and UNICORN, we utilize their open-source code to implement the first step of the BTI-based defense, which is trigger inversion. Then, following the method outlined in BTI-DBF, we utilize the trigger patterns inverted by NC or the backdoor image generator inverted by UNICORN to train a purification generator, thereby completing the second step of the defense, input purification. For BTI-DBF(P), we implement it using its open-source code.

Table 6: Performance (%) of REFINE under different sizes of the unlabeled benign dataset.

| Proportion | No Defense | | 100% | | 80% | | 60% | | 40% | | 20% | |
|---|---|---|---|---|---|---|---|---|---|---|---|---|
| Attack | BA | ASR | BA↑ | ASR↓ | BA↑ | ASR↓ | BA↑ | ASR↓ | BA↑ | ASR↓ | BA↑ | ASR↓ |
| BadNets | 92.31 | 100 | 91.20 | 0.86 | 90.22 | 1.05 | 89.53 | 1.21 | 87.81 | 1.11 | 83.93 | 2.21 |

Table 7: Performance (%) of REFINE under different values of temperature parameters $\lambda$.

| $\lambda$ | No Defense | | 1.0 | | 0.8 | | 0.6 | | 0.4 | | 0.2 | |
|---|---|---|---|---|---|---|---|---|---|---|---|---|
| Attack | BA | ASR | BA↑ | ASR↓ | BA↑ | ASR↓ | BA↑ | ASR↓ | BA↑ | ASR↓ | BA↑ | ASR↓ |
| BadNets | 91.74 | 100 | 90.83 | 0.92 | 90.87 | 0.76 | 90.60 | 0.60 | 91.03 | 0.51 | 90.69 | 1.27 |

## D ADDITIONAL ABLATION STUDY

### D.1 RESULTS ON ADDITIONAL MODEL ARCHITECTURES

In this section, we conduct experiments on three additional model architectures, including VGG-16 (Simonyan, 2014), DenseNet-121 (Huang et al., 2017), and ViT (Dosovitskiy et al., 2021). We conduct experiments on the CIFAR-10 dataset. We compare the defense performance of our RE-FINE with the most advanced transformation-based defense (*i.e.*, BDMAE) and the SOTA BTI-based defense (*i.e.*, BTI-DBF(P)).

As shown in Table 5, REFINE effectively defends against 7 representative attacks across 3 different model architectures, significantly outperforming baseline defenses. Specifically, under the REFINE defense, the benign accuracy shows a slight improvement, while the backdoor attack success rate is reduced to below 3.5%. The additional experimental results verify the effectiveness of REFINE.

### D.2 EFFECT OF THE UNLABELED BENIGN DATASET SIZE

In this section, we evaluate the defense performance of REFINE under different sizes of the unlabeled benign dataset. We train a backdoored classification model on CIFAR-10 using the BadNets attack on a ResNet-18 architecture. For defense, we use different proportions (100% to 20%) of the CIFAR-10 dataset as the unlabeled benign dataset. As shown in Table 6, the results indicate that as the number of unlabeled samples decreases, the BA of REFINE experiences a slight decline, while the ASR remains consistently low.

### D.3 EFFECT OF THE SCALAR TEMPERATURE PARAMETER $\lambda$

In this section, we evaluate the defense performance of REFINE under different values of temperature parameters $\lambda$. The attack setup is consistent with that in Section D.2. During the defense, we test various temperature parameters ranging from 1 to 0.2. As shown in Table 7, the results indicate that the value of temperature parameter has minimal impact on the defense performance of REFINE.

### D.4 EFFECT OF THE NUMBER OF CHANNALS IN UNET HIDDEN LAYERS

We hereby evaluate the performance of REFINE using UNet models with varying numbers of hidden layer channels. Specifically, the dimensionality of the encoded features can be adjusted by altering the number of output channels in the first layer of the UNet encoder. The attack setup is consistent with that in Section D.2. For the defense, we tested different channel numbers, including 32, 48, 64, and 80. As shown in Table 8, the number of channels in the UNet hidden layers has minimal impact on the defense performance of REFINE, with both BA and ASR remaining at an optimal level.

### D.5 EFFECT OF THE DATA DISTRIBUTION USED FOR DEFENSE

In our main experiments, we assume that the defender can acquire independent and identically distributed (i.i.d.) unlabeled datasets. In this section, we explore the defense performance under different data distributions. We train a ResNet-18 model on the CIFAR10 dataset using the BadNets

Table 8: Performance (%) of REFINE under different number of channels in UNet hidden layers.

| Channels | No Defense | | 32 | | 48 | | 64 | | 80 | |
|---|---|---|---|---|---|---|---|---|---|---|
| Attack | BA | ASR | BA↑ | ASR↓ | BA↑ | ASR↓ | BA↑ | ASR↓ | BA↑ | ASR↓ |
| BadNets | 91.74 | 100 | 89.49 | 1.09 | 90.61 | 0.64 | 90.18 | 1.43 | 91.07 | 0.78 |

Table 9: The performance (%) of REFINE in scenarios with different data distribution.

| Defense | No Defense | | REFINE | |
|---|---|---|---|---|
| Attack | BA | ASR | BA↑ | ASR↓ |
| BadNets | 91.18 | 100.00 | 88.39 | 1.40 |

Table 10: The performance (%) of REFINE and T-MR. The best results are **boldfaced**.

| Defense | No Defense | | T-MR | | REFINE | |
|---|---|---|---|---|---|---|
| Attack | BA | ASR | BA↑ | ASR↓ | BA↑ | ASR↓ |
| BadNets | 91.18 | 100.00 | 75.51 | 3.36 | **90.50** | **1.05** |
| WaNet | 91.29 | 99.91 | 74.49 | 25.76 | **90.64** | **1.93** |
| Adaptive | 92.54 | 99.93 | 75.49 | 5.87 | **90.87** | **1.76** |

attack. For defense, we trained the input transformation module of REFINE using CINIC10 (Darlow et al., 2018), a dataset with the same categories as CIFAR10 but a different data distribution.

As shown in Table 9, REFINE is still highly effective in reducing the attack success rate (ASR < 1.5%) while maintaining the model's benign accuracy (BA drop < 3%). This favorable result is due to the fact that REFINE first assigns pseudo-labels to the unlabeled benign samples using the original model, and then trains the input transformation module based on these pseudo-labels.

### D.6 EFFECT OF IMPROVED TRANSFORMATION MODULE

In this section, we conduct additional defense experiments using traditional model reprogramming methods (Elsayed et al., 2019) (dubbed 'T-MR'). We select three representative types of backdoor attacks, including BadNets, WaNet, and BATT. We train backdoor ResNet-18 models on the CIFAR-10 dataset. We compare the defense performance of REFINE with T-MR.

As shown in Table 10, the T-MR defense has a significant impact on the model's BA (BA drop > 15%) but fails to effectively reduce the ASR under the WaNet attack. This is because traditional model reprogramming methods only add a universal adversarial perturbation around the image, while the trigger pattern remains unchanged on the backdoor image to some extent.

## E    REFINE IN THE BLACK-BOX SCENARIO

In our main experiments, we assume that we can obtain white-box access to the pre-trained backdoored models. We hereby initially explore how to implement our REFINE in the black-box scenario where defenders can only get black-box access to the backdoored model. In this scenario, only the class confidence scores are accessible and it is hard to calculate the gradients to optimize the REFINE modules. To tackle the aforementioned challenge, we leverage the surrogate model technique. Specifically, we distill a surrogate model from the original black-box model using an unlabeled dataset $D$. We employ the mean squared error (MSE) loss to align the output confidence scores between the black-box model $\mathcal{F}(\cdot)$ and the surrogate model $\mathcal{F}_s(\cdot)$, as follows:

$$\mathcal{L}_{distill} = \frac{1}{|D|} \sum_{\boldsymbol{x} \in D} [\mathcal{F}(\boldsymbol{x}) - \mathcal{F}_s(\boldsymbol{x})]^2. \tag{10}$$

The surrogate model is then leveraged to replace the pre-trained model in our REFINE and optimize the input transformation module. Subsequently, the trained input transformation and output mapping modules are subsequently applied to the original black-box model.

Table 11: Performance (%) of REFINE in defending against attacks in black-box scenarios.

| Defense | No Defense | | | | REFINE | | | |
|---------|------------|--|--|--|--------|--|--|--|
| Model | Black-box | | Surrogate | | Surrogate | | Black-box | |
| Attack | BA | ASR | BA | ASR | BA↑ | ASR↓ | BA↑ | ASR↓ |
| BadNets | 90.60 | 100 | 91.20 | 1.24 | 88.21 | 0.92 | 88.17 | 0.36 |
| Blended | 91.08 | 96.94 | 90.69 | 2.23 | 88.34 | 0.62 | 87.75 | 0.18 |
| WaNet | 91.50 | 99.93 | 90.92 | 99.84 | 88.77 | 3.37 | 87.44 | 0.04 |
| Physical | 93.61 | 100 | 92.21 | 2.56 | 90.18 | 1.52 | 89.84 | 2.23 |
| BATT | 93.24 | 99.89 | 92.76 | 4.30 | 90.86 | 2.01 | 89.21 | 3.72 |
| LC | 91.95 | 93.06 | 91.53 | 1.11 | 89.04 | 0.87 | 88.69 | 1.05 |
| Adaptive | 90.15 | 100 | 90.36 | 1.57 | 88.41 | 0.32 | 87.91 | 0.44 |

Table 12: The overhead (minutes) of REFINE compared with BDMAE and BTI-DBF(P).

| Defense | BDMAE | BTI-DBF(P) | REFINE |
|---------|-------|------------|--------|
| Overhead | 39.67 | 15.49 | 36.70 |

To validate the feasibility of our REFINE in the black-box scenario, we employ the backdoored ResNet-50 pre-trained on the CIFAR-10 dataset as the black-box model and ResNet-18 as the surrogate model. As shown in Table 11, we evaluate both the black-box original model and the surrogate model in terms of BA and ASR before and after applying the REFINE defense. The ASRs of our REFINE are all below $4\%$. The results indicate that even though the input transformation module is trained using the surrogate model, our REFINE is still capable of achieving high performance of backdoor defense for the black-box original model.

## F  THE OVERHEAD OF OUR REFINE

In this section, we evaluate the overhead of our REFINE. Specifically, we measure the training time of the input transformation module and the model inference time on the CIFAR-10 using the ResNet-18 model. We employ a UNet with 32 hidden layer channels as the structure for the input transformation module. During training, we employ SGD with a momentum of 0.9 and a weight decay of $5 \times 10^{-4}$ as the optimizer. The initial learning rate is set to 0.01, with a batch size of 256. The input transformation module is trained for 150 epochs, with the learning rate decaying by a factor of 0.8 at the 100-th and 130-th epochs. For the training loss function, the temperature parameter is set to 0.1. We conduct all training using a single RTX 3090 GPU. For the output mapping module, a hard-coded remapping function is randomly assigned before each defense. Here, we compare REFINE's time consumption with that of BDMAE and BTI-DBF(P), which are the representative of SOTA transformation-based and BTI-based defenses, respectively.

As shown in Table 12, the overall time overhead of our REFINE is on par with SOTA baselines. Moreover, training the transformation module is a one-time process and can be done offline, although the pre-processing and model inference happen online. Once training of our REFINE is complete, inference on 10,000 images from CIFAR-10 takes 6.31 seconds, with the cost per image being nearly 0. Although REFINE introduces some additional overhead, we believe this cost is reasonable and acceptable.

## G  COMBINING REFINE WITH EXISTING DEFENSES

Arguably, our method can be used in conjunction with existing (model reconstruction-based) defenses to further enhance their effectiveness. To demonstrate this, we first applied model fine-tuning defense (dubbed 'FT') to a ResNet-18 model subjected to the BadNets attack on CIFAR-10, followed by the REFINE defense. As shown in Table 13, the FT+REFINE defense effectively reduces the backdoor ASR while maintaining the model's BA.

Table 13: The performance (%) of FT and FT+REFINE on ResNet-18.

| Defense | No Defense | | FT | | FT+REFINE | |
|---------|------|------|--------|---------|--------|---------|
| Attack | BA | ASR | BA($\uparrow$) | ASR($\downarrow$) | BA($\uparrow$) | ASR($\downarrow$) |
| BadNets | 91.18 | 100.00 | 91.89 | 91.67 | 90.42 | 0.87 |

## H RELATED WORK

### H.1 BACKDOOR ATTACK

**Visible Backdoor Attacks.** This type of attack typically employs patterns that are visible to humans as triggers. BadNets (Gu et al., 2019) is the first backdoor attack technique that injects samples with simple but visually noticeable patterns into the training data, such as white squares or specific marks. Li et al. (2021c) then proposed a transformation-based enhancement that strengthens the attack's resilience and establishes its applicability to physical scenarios. To address the issue of latent feature separation in backdoor attacks, Qi et al. (2023) employed asymmetric trigger planting strategies and developed adaptive backdoor poisoning attacks. Besides, Gao et al. (2023) revealed that clean-label attacks were difficult due to the conflicting effects of 'robust features' in poisoned samples and proposed a simple yet effective method to improve these attacks by targeting 'hard' samples instead of random ones.

**Invisible Backdoor Attacks.** To enhance the stealth of backdoor attacks, Chen et al. (2017) was the first to introduce the use of triggers that are imperceptible to humans, aiming to evade detection by basic data filtering techniques or human inspection. They proposed a blending strategy that generates poisoned images by subtly merging the backdoor trigger with benign images. After that, a series of studies focused on designing invisible backdoor attacks. WaNet (Nguyen & Tran, 2021) and ISSBA (Li et al., 2021d) employed warping-based triggers and perturbation-based triggers, respectively, introducing sample-specific trigger patterns during training; LIRA (Doan et al., 2021) formulated the learning of an optimal, stealthy trigger injection function as a non-convex constrained optimization problem, where the trigger generator function is trained to manipulate inputs using imperceptible noise; BATT (Xu et al., 2023) utilized images rotated to a specific angle as triggers, representing a new attack paradigm where triggers extend beyond basic pixel-wise manipulations.

A few existing literature also provided novel and comprehensive discussions on backdoor attacks from various domains and applications, such as diffusion models (Chou et al., 2024), 3D point clouds (Wei et al., 2024), ViTs (Yang et al., 2024a), code generation (Yang et al., 2024b), audio (Zhai et al., 2021; Cai et al., 2024), and federated learning (Shao et al., 2024). Moreover, some existing works also explore utilizing the backdoor attack for good purposes, such as copyright protection (Li et al., 2022a; 2023a; Guo et al., 2023; 2024; Li et al., 2025) and XAI evaluation (Ya et al., 2023).

### H.2 BACKDOOR DEFENSES

Currently, there are various backdoor defense methods (Li et al., 2024a;b) designed to mitigate backdoor threats. These methods can generally be divided into three main paradigms (Li et al., 2024c): (1) trigger-backdoor mismatch, which primarily refers to pre-processing-based defenses (Liu et al., 2017; Li et al., 2021c; Shi et al., 2023). (2) backdoor elimination (Li et al., 2021b; Zhao et al., 2020; Zeng et al., 2021; 2022; Huang et al., 2022; Xu et al., 2024), such as model reconstruction (Wang et al., 2020; Li et al., 2021b; Zeng et al., 2022), poison suppression (Huang et al., 2022; Tang et al., 2023), and training sample filtering (Hayase & Kong, 2020; Zeng et al., 2021). (3) trigger elimination, also known as testing sample filtering (Gao et al., 2019; Xie et al., 2024; Yi et al., 2025).

**Pre-processing-based Defenses.** These methods incorporate a pre-processing module prior to feeding samples into DNNs, altering the trigger patterns present in the samples. Consequently, the modified triggers no longer align with the hidden backdoor, thereby preventing the backdoor activation. AutoEncoderDefense (Liu et al., 2017) is the first pre-processing-based backdoor defense by employing a pre-trained autoencoder as the pre-processing module. Based on the idea that trigger regions have the most significant impact on predictions, Februus (Doan et al., 2020) effectively mitigates backdoor attacks by removing potential trigger artifacts and reconstructing inputs, all while preserving performance for both poisoned and benign samples. Li et al. (2021c) observed that

poisoning-based attacks with static trigger patterns degrade sharply with slight changes in trigger appearance or location and proposed spatial transformations (e.g., shrinking, flipping) as an efficient defense with minimal computational cost. Deepsweep (Qiu et al., 2021) proposes a unified defense that (1) fine-tunes the infected model using a data augmentation policy to remove backdoor effects and (2) pre-processes input samples with another augmentation policy to disable triggers during inference. Recently, many pre-processing-based defenses utilize the generative model, such as the diffusion model and the masked autoencoder, to purify suspicious samples. ZIP (Shi et al., 2023) applies linear transformations, such as blurring, to poisoned images to disrupt backdoor patterns and subsequently employs a pre-trained diffusion model to recover the semantic information lost during the transformation. BDMAE (Sun et al., 2023) detects potential triggers in the token space by evaluating image structural similarity and label consistency between test images and MAE restorations, refines these results based on trigger topology, and finally adaptively fuses the MAE restorations into a purified image for prediction.

**Backdoor Elimination Defenses.** In contrast to pre-processing-based defenses, backdoor elimination methods typically mitigate backdoor threats by directly modifying model parameters or prevent backdoor injection by controlling the model training process. Li et al. (2021a) identified two key weaknesses of backdoor attacks: (1) models learn backdoored data significantly faster than clean data, and (2) the backdoor task is associated with a specific target class. Consequently, they proposed Anti-Backdoor Learning (ABL), which introduces a two-stage gradient ascent mechanism: (1) isolating backdoor examples in the early training phase, and (2) breaking the correlation between backdoor examples and the target class in the later training phase. Inspired by the phenomenon where poisoned samples tend to cluster together in the feature space of the attacked DNN model, Huang et al. (2022) proposed a novel backdoor defense by decoupling the original end-to-end training process into three stages. Yang et al. (2023) removed backdoors by suppressing the skip connections in key layers identified by their method and fine-tuned these layers to restore high BA and further reduce the ASR. Neural Polarizer (Zhu et al., 2023) achieved effective defense by training an additional linear transformation, called neural polarizer, using only a small portion of clean data without modifying the model parameters. DataElixir (Zhou et al., 2024) detects target labels by quantifying distribution discrepancies, selects purified images based on pixel and feature distances, and determines their true labels by training a benign model. Xu et al. (2024) discovered that even in the feature space, the triggers generated by existing BTI methods differ significantly from those used by the adversary. Consequently, they proposed BTI-DBF, which decouples benign features instead of directly decoupling backdoor features. This method primarily involves two key steps: (1) decoupling benign features, and (2) triggering inversion by minimizing the differences between benign samples and their generated poisoned versions while maximizing the differences of the remaining backdoor features.

**Trigger Elimination Defenses.** These defenses filter out malicious samples during the inference process rather than during training. As a result, the deployed model exclusively predicts benign test samples or purified attack samples, thereby preventing backdoor activation by removing trigger patterns. STRIP (Gao et al., 2019) perturbs the input samples and observes the randomness in predicted classes from the deployed model for these perturbed inputs. If the entropy of the predicted classes is low, this violates the input-dependence characteristic of a benign model, indicating the presence of malicious features within the input. Du et al. (2020) demonstrated that applying differential privacy can enhance the utility of outlier detection and novelty detection, and further extended this approach for detecting poisoned samples in backdoor attacks. Besides, CleaNN (Javaheripi et al., 2020) leverages dictionary learning and sparse approximation to characterize the statistical behavior of benign data and identify triggers, representing the first end-to-end framework capable of online mitigation against backdoor attacks in embedded DNN applications.

## H.3 MODEL REPROGRAMMING

Elsayed et al. (2019) first proposed adversarial reprogramming, which aims to repurpose a classifier trained on ImageNet-1K for tasks such as classifying CIFAR-10 and MNIST images and counting the number of squares in an image. BAR (Tsai et al., 2020) extended model reprogramming to blackbox scenarios and applied it to the bio-medical domain. Driven by advancements in deep speech processing models and the fact that speech data is a univariate time signal, Voice2Series (Yang et al., 2021) learns to reprogram acoustic models for time series classification and output label mapping through input transformations. Neekhara et al. (2022) analyzed the feasibility of adversar-

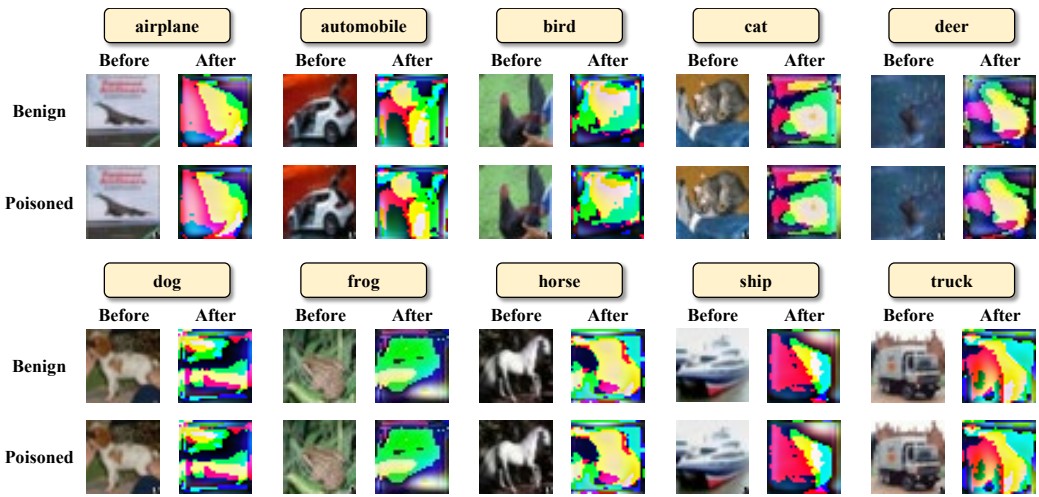

Figure 6: The visualization of transformed samples $\tilde{x}$. We display the benign and poisoned samples and transformed benign and poisoned samples for each class. For each class of small areas, the upper left corner represents the benign sample, the upper right corner represents the transformed benign sample, the bottom left corner represents the poisoned sample and the bottom right corner represents the poisoned sample after transformations.

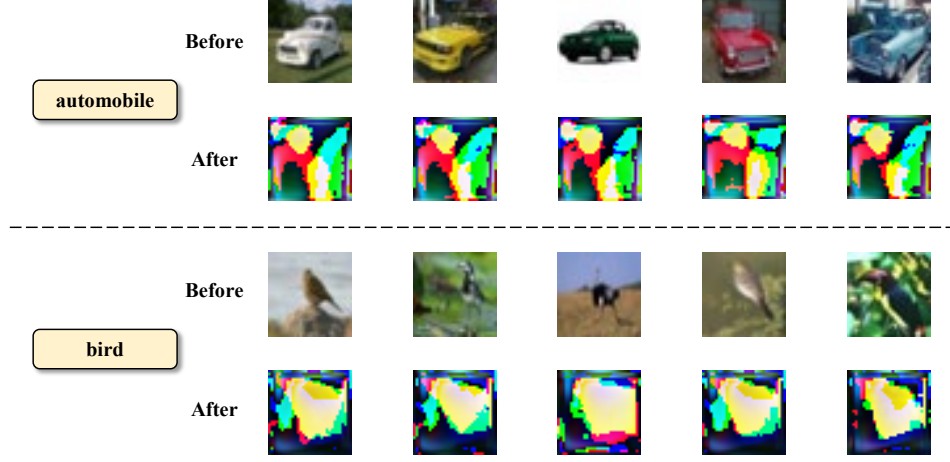

Figure 7: The visualization of transformed samples $\tilde{x}$ for classes 'automobile' and 'bird' of CIFAR-10. For each class, we display five input images and their transformed images.

ially repurposing image classification neural networks for natural language processing (NLP) and other sequence classification tasks. They developed an effective adversarial program that maps a series of discrete tokens onto an image, which can then be classified into the desired category by an image classification model. Li et al. (2023c) found that combining Visual Prompting (VP) with PATE—a state-of-the-art differential privacy training method that utilizes knowledge transfer from a team of teachers—achieves a cutting-edge balance between privacy and practicality with minimal expenditure on privacy budget. More Recently, a novel application (Dey & Nair, 2024) of model reprogramming repurposed models originally designed for able-bodied individuals to predict joint movements in amputees, significantly enhancing assistive technologies and improving mobility for amputees. Currently, model reprogramming has been shown to outperform transfer learning and training from scratch in many applications (Tsai et al., 2020; Yang et al., 2021; Vinod et al., 2023), without altering the original model's parameters.

# I  THE VISUALIZATION OF THE TRANSFORMED SAMPLES $\tilde{x}$

In this section, we visualize the transformed benign and poisoned samples $\tilde{x}$ generated by the UNet of our REFINE. We train a backdoored ResNet-18 model on CIFAR-10 using the BadNets attack

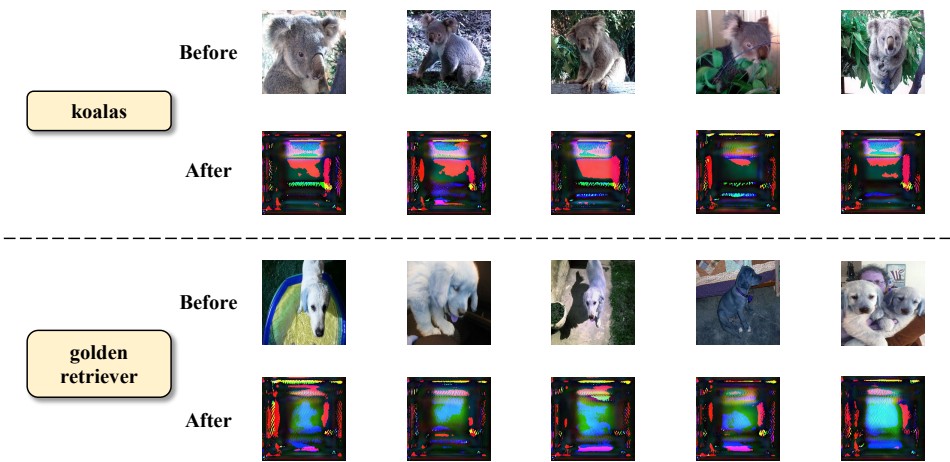

Figure 8: The visualization of transformed samples $\tilde{x}$ for classes 'koalas' and 'golden retriever' of ImageNet. For each class, we display five input images and their transformed images.

with a specified $3 \times 3$ trigger patterns at the bottom right corner of images, and the hard-coded remapping function $f_L$ of the output mapping module $\mathcal{M}$ is defined as follows:

$$f_L = \tilde{l} \mapsto l, \tag{11}$$

where

$$\tilde{l} = \{airplane, automobile, bird, cat, deer, dog, frog, horse, ship, truck\}, \tag{12}$$

and

$$l = \{cat, deer, automobile, ship, frog, bird, horse, truck, airplane, dog\}. \tag{13}$$

As shown in Figure 6, for both benign and poisoned samples, the transformed sample patterns are very similar, and the transformed pattern of the poisoned sample effectively removes the trigger. This further illustrates the effectiveness of our REFINE in mitigating backdoor threats.

As shown in Figure 7 and 8, samples from the same class exhibit visual similarities after transformation. However, the transformed samples do not contain any human-recognizable information. This phenomenon occurs because the input transformation module maps the samples to a new benign feature space, and the constraint imposed by the supervised contrastive loss ensures that transformed samples from the same class exhibit more similar benign features.

## J  THE VISUALIZATION OF THE FEATURE DISTRIBUTION BEFORE AND AFTER REFINE

We hereby visualize the changes in the feature distribution of the input samples before and after REFINE. Specifically, we trained a backdoor ResNet-18 model on CIFAR-10 using the BadNets attack and extracted the features from the input of the model's fully connected (FC) layer as the feature values of the input samples.

As shown in Figure 9, before applying REFINE, the feature distributions of benign and poisoned samples are clustered in two distinct locations. After applying REFINE, the feature distributions of benign and poisoned samples are interwoven and clustered in the same new location. This indicates that REFINE effectively removes the trigger patterns from the poisoned samples and maps samples of the same class to a new benign distribution.

As shown in Figure 10, before applying REFINE, the benign samples of each class form distinct clusters in the feature space. After applying REFINE, the benign samples, adjusted by the input transformation module and output mapping module, form new clusters in different positions. This empirically demonstrates that REFINE is able to maintain the model's benign accuracy.

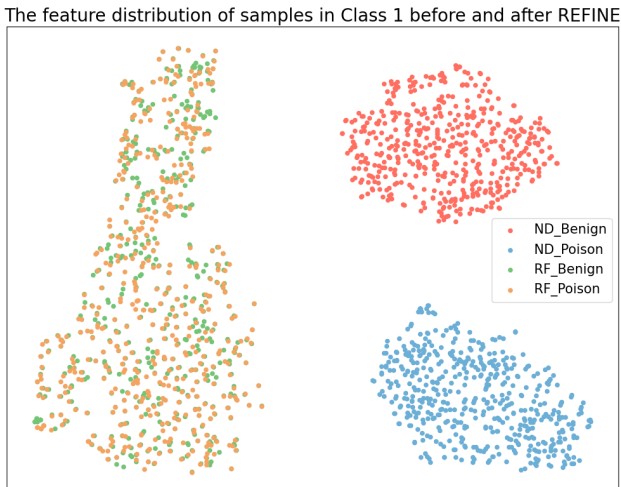

Figure 9: The t-SNE plots of the feature distribution of samples in Class 1 before and after REFINE. ND_Benign and ND_Poison represent the features of benign and poisoned samples under the No Defense (ND) scenario, respectively. RF_Benign and RF_Poison represent the features of benign and poisoned samples after applying REFINE, respectively.

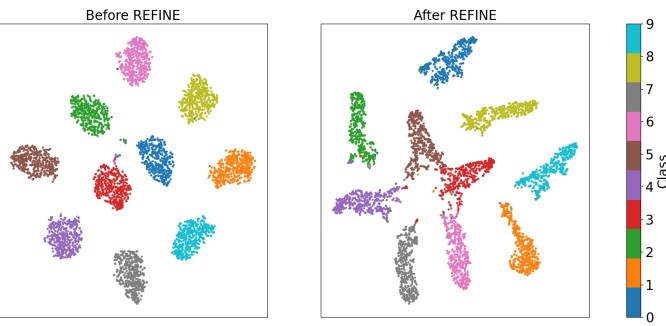

Figure 10: The t-SNE plots of the feature distribution of benign samples from different classes, both before and after REFINE.

## K  SOCEITAL IMPACT

This paper aims to design an effective and efficient backdoor defense method and have a positive societal impact. Specifically, we propose a novel pre-processing-based backdoor defense method, REFINE, based on model reprogramming. REFINE can mitigate the backdoor behaviors injected into the third-party pre-trained models. Therefore, our REFINE can assist in ensuring the stable and reliable operation of the AI models, mitigating the potential threat of backdoors, and facilitating the reuse and deployment of the models. Moreover, the application of our REFINE may also facilitate the emergence of new business models such as model trading.

On the other hand, in this paper, we propose to leverage the model reprogramming techniques to build the input transformation and output mapping modules to mitigate the backdoors. The insight of our method can also be applied to the use of the pre-trained model in an unauthorized way. For instance, an adversary might use the model for an unauthorized task via model reprogramming, leading to copyright infringement (Shao et al., 2025; Wang et al., 2022). However, we argue that the negative societal impact is negligible. The model developer can employ several existing protection methods, such as non-transfer learning (Wang et al., 2022), to prevent such misbehaviors. Moreover, although we do not find effective adaptive attacks against our REFINE, an adversary may design a more advanced adaptive attack to circumvent our proposed method since its effectiveness lacks of

Table 14: The performance (%) of REFINE in the 10% unlabeled data scenario on ResNet-18.

| Defense | No Defense | | REFINE | |
|---------|------------|------|--------|------|
| Attack | BA | ASR | BA↑ | ASR↓ |
| BadNets | 91.18 | 100.00 | 78.02 | 2.90 |
| Blended | 90.64 | 98.18 | 77.89 | 2.59 |
| WaNet | 91.29 | 99.91 | 78.79 | 1.83 |
| PhysicalBA | 93.67 | 100.00 | 79.87 | 2.34 |

theoretical guarantees. Even so, the model users and developers can still prevent the backdoor threat from the source by only using trusted pre-trained models.

## L  POTENTIAL LIMITATIONS AND FUTURE DIRECTIONS

Firstly, as outlined in our threat model, the goal of our defense is to protect against pre-trained models from third-party platforms. Specifically, similar to other baseline methods, we assume that the defender possesses a certain amount of unlabeled sample datasets. To explore the effectiveness of REFINE in few-shot scenarios, we conduct additional experiments using 10% unlabeled clean data. We apply the REFINE defense to a ResNet-18 model trained on the CIFAR-10 dataset, which is subjected to the BadNets attack. In this case, the unlabeled training set for REFINE used only 10% of the CIFAR-10 training set.

As shown in Table 14, even with only 10% unlabeled data, REFINE is still effective to some extent. REFINE effectively reduces the ASR, although it does have some impact on the model's BA. Therefore, in cases where the defender lacks the number of unlabeled samples, it becomes impossible to train the input transformation module, thereby hindering the execution of the intended defense. Currently, with the widespread application of generative models, obtaining a sufficient amount of unlabeled samples is no longer a challenging task. In the future, we will continue to explore how to maintain the effectiveness of our REFINE in few-shot scenarios.

Secondly, we need to train a local input transformation module, which requires certain computational resources and time. While this overhead is somewhat higher than that of pre-processing defenses based on random transformations, it is significantly lower than the overhead associated with pre-processing defenses based on generative models and BTI-based methods, as presented in Appendix F. This overhead is considered acceptable compared to retraining a DNN from scratch.

Finally, our method primarily focuses on backdoor defense for image classification models. Fortunately, existing researchs (Yang et al., 2021; Neekhara et al., 2022) have demonstrated that model reprogramming techniques can yield favorable results in fields such as text and audio. We will explore the reprogramming-based backdoor defense in other modalities and tasks in our future work.

## M  DISCUSSION ON ADOPTED DATA

In our experiments, we only use open-source dataset (*i.e.*, CIFAR-10 (Krizhevsky et al., 2009) and ImageNet (Deng et al., 2009)) for evaluation. Our research strictly obeys the open-source licenses of these datasets and does not lead to any privacy issues. The ImageNet dataset may include some personal elements. For instance, data about human faces is available in the ImageNet dataset. Nevertheless, our work treats all objects equally and does not intentionally exploit or manipulate these elements. As such, our work complies with the requirements of these datasets and should not be construed as a violation of personal privacy.

