# OpenReview forum: "REFINE: Inversion-Free Backdoor Defense via Model Reprogramming"
_ICLR.cc/2025/Conference — ICLR 2025 Poster_

### Official Review · Reviewer_GGnQ · 2024-10-23

**Soundness:** 2
**Presentation:** 4
**Contribution:** 3
**Rating:** 5
**Confidence:** 4

**Summary:**

This paper addresses the challenge of backdoor attacks in deep neural networks, where current defenses struggle to balance accuracy and trigger removal. The proposed REFINE method combines input transformation and output remapping to effectively neutralize backdoors without relying on trigger inversion. Experiments show REFINE's strong defense capabilities while maintaining model performance.

**Strengths:**

1. The idea may seem bold and imaginative, but it is worth exploring in depth. I believe both the threat model setup and the proposed solution are good.
2. Overall, the paper is well-written, making it relatively easy for someone unfamiliar with model reprogramming, like myself, to grasp the design concepts.
3. The method operates effectively with a small amount of unlabeled data, which not only cleverly avoids direct comparison with fine-tuning-based defenses but also demonstrates its practicality through ablation experiments that confirm its minimal data requirements.

**Weaknesses:**

1.Regarding the theoretical explanation in the paper (Formula 1), I feel it’s missing a key component and has only achieved about 50% of its purpose. It aims to explain how amplifying the distributional differences in output features changes the variation in prediction results. As the authors emphasize, model reprogramming significantly amplifies these distributional differences, and trigger patterns can be randomized. I agree with this point, but at this stage, the performance guarantee for clean samples doesn’t tie back to this theory; I see it as being empirically supported by the two losses. In fact, even with model reprogramming, Formula 1 still holds, so it may not fundamentally support the authors’ viewpoint (since Formula 1 is a rule that must be met across the overall distribution; I believe focusing on local distributions could be a potential solution).

2.The whole experiments are only conducted on a ResNet-18 model. Furthermore, the discussion of the black-box part in the experiments seems counterproductive to me; I don’t understand the rationale behind setting up absurdly similar black-box and substitute models for exploration.

3.The adaptive attack setup is unreasonable. Based on the general consensus in the community, in adaptive attacks, the attacker is assumed to know the defender’s hard-coded choices. Disallowing the defender from making real-time adjustments (which is a strategy directly aimed at adaptive attacks) is therefore a logical error. Additionally, the authors haven’t clearly explained why a conflict exists between adaptive loss and backdoor loss, resulting in a decrease in BA, and I don’t clearly see a trade-off here.

4.Why is BTI-DBF unable to reverse even the most basic BadNet trigger? I seriously question the validity of this experiment. In the original BTI-DBF paper, it’s evident that it can reconstruct the triggers of certain fixed samples quite effectively.

5.Additionally, I would like to see more examples of transformed samples. Do transformed samples from the same class still share some visual similarities, or do they contain almost no human-recognizable information? What is the underlying significance of these transformations? I am curious if the authors could offer some more unique, high-level insights.

**Questions:**

See weakness part.

---

> ### Author Response · Authors · 2024-11-20
> **Author Response (Part 1)**
>
> Dear Reviewer GGnQ, thank you very much for your careful review of our paper and thoughtful comments. We are encouraged by your positive comments on our **bold and imaginative idea**, **well-writing of the paper**, **effectiveness with a small amount of unlabeled data**, **excellent presentation**, and **good contribution**. We hope the following responses can help clarify potential misunderstandings and alleviate your concerns.
>
> ---
>
> **Q1**: Regarding the theoretical explanation in the paper (Formula 1), I feel it’s missing a key component and has only achieved about 50% of its purpose. It aims to explain how amplifying the distributional differences in output features changes the variation in prediction results. As the authors emphasize, model reprogramming significantly amplifies these distributional differences, and trigger patterns can be randomized. I agree with this point, but at this stage, the performance guarantee for clean samples doesn’t tie back to this theory; I see it as being empirically supported by the two losses. In fact, even with model reprogramming, Formula 1 still holds, so it may not fundamentally support the authors’ viewpoint (since Formula 1 is a rule that must be met across the overall distribution; I believe focusing on local distributions could be a potential solution).
>
> **R1**: Thank you for these insightful comments! We are deeply sorry that our submission may lead to some misunderstandings that we want to clarify.
>
> - We admit that our Theorem 1 does not propose a guarantee for the performance of the clean samples. Instead, we attempt to **maximize the performance of the model by adding a term into the loss function**.
> - We agree that Theorem 1 holds with or without the model reprogramming. However, **it does not necessarily mean that Theorem 1 cannot guide the design of our method**.
>     - Theorem 1 demonstrates that the distance between the prediction of the transformed poisoned sample and the original poisoned prediction is bounded by the Wasserstein-1 distance between the probability measures of the output representations.
>     - As such, to design an effective pre-processing-based backdoor defense method, **we can enhance the upper bound by increasing the distance between the probability measures of the output representations**.
>     - Motivated by the above understanding, we **utilize model reprogramming to significantly transform the output domain and destroy potential trigger patterns**.
> - We do not fully understand the meaning of 'focusing on local distributions' since there is limited information. If you could provide more details, we are also very willing to provide further discussions :)

---

> ### Author Response · Authors · 2024-11-20
> **Author Response (Part 2)**
>
> ---
>
> **Q2**: The whole experiments are only conducted on a ResNet-18 model. Furthermore, the discussion of the black-box part in the experiments seems counterproductive to me; I don’t understand the rationale behind setting up absurdly similar black-box and substitute models for exploration.
>
> **R2**: Thank you for these insightful comments! We are deeply sorry for not demonstrating the effectiveness of our method across different network structures in our previous submission. We do agree with you that it is beneficial to verify the universality of our defense across different network structures.
> - To address your concerns, following your suggestions, **we conduct additional experiments on five different network structures**, including ResNet18, ResNet50, VGG16, DenseNet121 and InceptionV3. We select three representative types of backdoor attacks, including BadNets, WaNet, and Adaptive-Patch (dubbed ‘Adaptive’). We conduct experiments on the CIFAR-10 dataset. We compare the defense performance of our REFINE with the most advanced transformation-based defense (i.e., BDMAE).
> - As shown in Tables 1 to 5, REFINE effectively defends against three representative attacks across five different network architectures, significantly outperforming BDMAE. Specifically, under the REFINE defense, the benign accuracy (BA) drop is less than 1.5%, with some cases showing an increase in BA. Meanwhile, the backdoor attack success rate (ASR) reduces to below 3%. **The additional experimental results verify the effectiveness of REFINE.**
>
> We provide more details in *Appendix E.1* of our revision.
>
> **About Black-box Experiments**:
> - The black-box section represents only our initial exploration of REFINE's defense performance in black-box scenarios. **We aimed to demonstrate that REFINE can be adapted for black-box defense through certain methods, such as using distilled surrogate models.**
> - Considering that the logits distribution output by distillation can be challenging to align across different model architectures, we selected a structurally similar smaller model, ResNet-18, as the surrogate for the black-box model ResNet-50. Using a surrogate model with a significantly different architecture from the black-box model would make it difficult to achieve satisfactory defense performance.
> - We will explore how to comprehensively design REFINE in a black-box manner in future work.
>
> **Table 1**. The performance (\%) of REFINE and BDMAE on **ResNet18**. **ND** means that no defense is applied. The best results are **boldfaced**.
> |          | BA (ND) | ASR (ND) | BA (BDMAE) | ASR (BDMAE) | BA (REFINE) | ASR (REFINE) |
> | -------- | ------- | -------- | ---------- | ----------- | ----------- | ------------ |
> | BadNets  | 91.18   | 100      | 90.48      | 14.81       | **90.50**   | **1.05**     |
> | WaNet    | 91.29   | 99.91    | 89.87      | 99.93       | **90.64**   | **1.93**     |
> | Adaptive | 89.62   | 100      | 89.40      | 49.18       | **90.54**   | **1.23**     |
>
> **Table 2**. The performance (\%) of REFINE and BDMAE on **ResNet50**. **ND** means that no defense is applied. The best results are **boldfaced**.
> |          | BA (ND) | ASR (ND) | BA (BDMAE) | ASR (BDMAE) | BA (REFINE) | ASR (REFINE) |
> | -------- | ------- | -------- | ---------- | ----------- | ----------- | ------------ |
> | BadNets  | 91.91   | 100      | **91.04**  | 10.99       | 90.71       | **1.53**     |
> | WaNet    | 91.70   | 99.98    | 89.83      | 99.89       | **91.09**   | **0.35**     |
> | Adaptive | 89.59   | 85.11    | 89.06      | 35.91       | **90.05**   | **2.19**     |
>
> **Table 3**. The performance (\%) of REFINE and BDMAE on **VGG16**. **ND** means that no defense is applied. The best results are **boldfaced**.
> |          | BA (ND) | ASR (ND) | BA (BDMAE) | ASR (BDMAE) | BA (REFINE) | ASR (REFINE) |
> | -------- | ------- | -------- | ---------- | ----------- | ----------- | ------------ |
> | BadNets  | 84.44   | 99.36    | 84.25      | 18.32       | **86.86**   | **1.62**     |
> | WaNet    | 84.75   | 99.15    | 83.36      | 99.25       | **86.41**   | **2.39**     |
> | Adaptive | 84.98   | 99.99    | 84.69      | 40.09       | **86.63**   | **2.04**     |
>
> **Table 4**. The performance (\%) of REFINE and BDMAE on **DenseNet121**. **ND** means that no defense is applied. The best results are **boldfaced**.
> |          | BA (ND) | ASR (ND) | BA (BDMAE) | ASR (BDMAE) | BA (REFINE) | ASR (REFINE) |
> | -------- | ------- | -------- | ---------- | ----------- | ----------- | ------------ |
> | BadNets  | 86.40   | 99.99    | 86.05      | 11.85       | **89.44**   | **0.96**     |
> | WaNet    | 86.31   | 98.77    | 85.42      | 98.91       | **88.74**   | **0.88**     |
> | Adaptive | 85.16   | 100      | 84.45      | 45.36       | **88.74**   | **0.35**     |

---

> ### Author Response · Authors · 2024-11-20
> **Author Response (Part 3)**
>
> **Table 5**. The performance (\%) of REFINE and BDMAE on **InceptionV3**. **ND** means that no defense is applied. The best results are **boldfaced**.
> |          | BA (ND) | ASR (ND) | BA (BDMAE) | ASR (BDMAE) | BA (REFINE) | ASR (REFINE) |
> | -------- | ------- | -------- | ---------- | ----------- | ----------- | ------------ |
> | BadNets  | 90.46   | 99.97    | 90.61      | 80.51       | **91.03**   | **0.75**     |
> | WaNet    | 90.09   | 99.73    | 89.64      | 99.76       | **91.01**   | **0.54**     |
> | Adaptive | 88.58   | 13.53    | 88.54      | 13.52       | **90.36**   | **0.54**     |
>
>
>
> ---
>
> **Q3**: The adaptive attack setup is unreasonable. Based on the general consensus in the community, in adaptive attacks, the attacker is assumed to know the defender’s hard-coded choices. Disallowing the defender from making real-time adjustments (which is a strategy directly aimed at adaptive attacks) is therefore a logical error. Additionally, the authors haven’t clearly explained why a conflict exists between adaptive loss and backdoor loss, resulting in a decrease in BA, and I don’t clearly see a trade-off here.
>
> **R3**: Thank you for this insightful comment! We sincerely apologize for not clearly explaining the setup of the adaptive attack. Here, we would like to provide further clarification to address potential misunderstandings and alleviate your concerns.
>
> - There may have been a misunderstanding. In general, for adaptive attacks, the community usually assumes that **the attacker is aware of the defender’s method but does not know the specific parameters used by defenders**. Specifically, in the case of our method, REFINE, the attacker is aware of the input transformation module, output mapping module, and the loss function. However, the attacker is unaware of the initialization parameters of the input transformation module and the hard-coded choices of the output mapping module. Arguably, your suggested scenarios usually do not exist in practice.
> - In our threat model, the defender obtains a pre-trained model from a third-party platform. During defense, the output mapping used by REFINE is randomly specified and is not known to the attacker in advance. As such, **we respectfully argue that this is not the same as the "real-time adjustments" you referred**.
> - **Why there is a trade-off in the adaptive attack**:
>     - We can view the adaptive loss as a constraint used to train a 'new' model, which consists of the input transformation module, the original model, and the output mapping module. **This model is different from the original model**.
>     - As the temperature parameter $\gamma$ increases, **the adaptive attack becomes more inclined to optimize the performance of the 'new' model instead of the original model**, leading to a degradation in the performance of the original model.
>
> ---
>
> **Q4**: Why is BTI-DBF unable to reverse even the most basic BadNet trigger? I seriously question the validity of this experiment. In the original BTI-DBF paper, it’s evident that it can reconstruct the triggers of certain fixed samples quite effectively.
>
> **R4**: Thank you for your insightful comments! We sincerely apologize for not providing sufficient details about the experimental procedures in Section 3.3, which may have caused confusion and concern.
>
> - The difference between the experimental results presented in Figure 3 and the original BTI-DBF paper [1]:
>     - In general, **this is caused by different experimental settings**.
>         - In the original BTI-DBF's BadNets attack setup, only normalization was used as the data pre-processing function.
>         - In our BadNets attack setup, we add two data augmentation techniques, including random cropping and random flipping.
>         - Compared to the BTI-DBF's setup, our configuration is more reasonable. This is because attackers can manipulate the model’s training process, including data augmentation, to enhance the robustness of the backdoor attack.
>         - **Applying data augmentation to the attack method makes it more difficult to invert the backdoor trigger**. For example, in the case of BadNets attack, operations such as random cropping and padding increase the number of triggers that can activate the backdoor, making it harder to identify the ground-truth trigger used by the attacker.
>     - Nevertheless, we have adjusted the experimental results shown in Figure 3 to make them more clear and more intuitive.
>
> **References**
>
> 1. Towards reliable and efficient backdoor trigger inversion via decoupling benign features.

---

> ### Author Response · Authors · 2024-11-20
> **Author Response (Part 4)**
>
> ---
>
> **Q5**: Additionally, I would like to see more examples of transformed samples. Do transformed samples from the same class still share some visual similarities, or do they contain almost no human-recognizable information? What is the underlying significance of these transformations? I am curious if the authors could offer some more unique, high-level insights.
>
> **R5**: Thank you for these constructive suggestions! We are deeply sorry that we did not provide sufficient visualizations of the experimental results for REFINE. We would like to add supplementary content in the appendix to further analyze the reasons behind REFINE’s effectiveness.
>
> - As you expected, **samples from the same class exhibit visual similarities after transformation**. However, **the transformed samples do not contain any human-recognizable information**.
>     - It is probably because the input transformation module maps the samples to a new benign feature space, and the constraint imposed by the supervised contrastive loss ensures that transformed samples from the same class exhibit more similar benign features.
>     - Please kindly find more examples of transformed samples in *Appendix J*.
> - We believe the underlying significance of these transformations is that **they map the input from the original benign or poisoned image domains to a new benign image domain**, ensuring that both benign and backdoor samples are distributed similarly in the feature space.
>     - To visually reflect the changes of input sample distribution in the feature space before and after REFINE, **we used t-SNE plots for visualization**. We add two figures and a detailed explanation in *Appendix K* to clarify those details. The illustrations of added figures are as follows:
> ```!
> As shown in Figure 8, before applying REFINE, the feature distributions of benign and poisoned samples are clustered in two distinct locations. After applying REFINE, the feature distributions of benign and poisoned samples are interwoven and clustered in the same new location. This indicates that REFINE effectively removes the trigger patterns from the poisoned samples and maps samples of the same class to a new benign feature distribution.
> ```
> ```!
> As shown in Figure 9, before applying REFINE, the benign samples of each class form distinct clusters in the feature space. After applying REFINE, the benign samples, adjusted by the input transformation module and output mapping module, form new clusters in different positions. This empirically demonstrates that REFINE is able to maintain the model's benign accuracy.
> ```

---

> ### Author Response · Authors · 2024-11-23
> **Thanks to Reviewer GGnQ**
>
> Please allow us to thank you again for reviewing our paper and the valuable feedback, and in particular for recognizing the strengths of our paper in terms of the bold and imaginative idea, well-writing of the paper, effectiveness with a small amount of unlabeled data, excellent presentation, and good contribution.
>
> Please let us know if our response and the new experiments have properly addressed your concerns. We are more than happy to answer any additional questions during the post-rebuttal period. Your feedback will be greatly appreciated.

---

> ### Author Response · Authors · 2024-11-24
> **A Gentle Reminder of the Final Feedback**
>
> Thank you very much again for your initial comments. They are extremely valuable for improving our work. We shall be grateful if you can have a look at our response and modifications, and please let us know if anything else that can be added to our next version.

---

> ### Author Response · Authors · 2024-11-25
> **A Second Reminder of the Post-rebuttal Feedback**
>
> Dear Reviewer GGnQ,
>
> We greatly appreciate your initial comments. We totally understand that you may be extremely busy at this time. But we still hope that you could have a quick look at our responses to your concerns. We appreciate any feedback you could give to us. We also hope that you could kindly update the rating if your questions have been addressed. We are also happy to answer any additional questions before the rebuttal ends.
>
> Best Regards,
>
> Paper4208 Authors

---

> > ### Comment · Reviewer_GGnQ · 2024-11-25
> >
> > #### Q1
> > I hope the authors acknowledge this fact in the manuscript. From my previous reading, I believe the authors are concealing the issue that *Theorem 1 does not propose a guarantee for the performance of clean samples*.
> >
> > #### Q2
> > Thank you for providing additional experiments. However, there are too many supplementary experimental models. I am curious why these critical experiments were not included prior to submission. That said, I do acknowledge and appreciate these additional results.
> > Furthermore, could you clarify the reasoning behind *"Considering that the logits distribution output by distillation can be challenging to align across different model architectures"*? There are existing methods that attempt to steal models even under hard-label scenarios. If soft labels are ineffective, how can this issue be addressed?
> >
> > #### Q3
> > I partially agree with the response, but I would only fully accept it if supported by references to relevant literature.
> >
> > #### Q4
> > I want to emphasize that the authors themselves admitted *"this is caused by different experimental settings."* This suggests an intentional effort to degrade the performance of other methods. Even though this is in the motivation section, I question whether the comparisons in the experimental section are fair.
> > Additionally, do the updated images ensure optimal reproducibility? From what I observed, the reverse-engineered images largely retain the characteristics of the trigger. This undermines the claim that trigger reverse-engineering is ineffective. I believe the motivation section needs to be rewritten to ensure it highlights the key or universal issues with these approaches, rather than adjusting experimental settings to worsen their performance, or worse, omitting such details and misleading readers.
> >
> > #### Q5
> > The results for low-pixel datasets seem inconsistent with the results presented in the pipeline figure. Are the results for the 224×224 datasets still of the same nature? Where exactly does the pipeline figure's reference come from?
> >
> > ---
> >
> > In summary, I am dissatisfied with the authors’ rebuttal. While I initially liked the idea and leaned toward giving it a high score, the progressively exposed issues have made me reconsider my stance.

---

> ### Comment · Reviewer_GGnQ · 2024-11-25
>
> I was previously occupied and only now have time to address the response. Additionally, regarding the loss conflict mentioned in Q3, I didn’t fully grasp what the authors intended to convey (I feel somewhat lost regarding the detailed content of the manuscript, and I believe a more straightforward clarification would be helpful). According to optimization theory, optimization can proceed regardless of how low the loss is. The claim in the paper that failure is caused by the refine loss being sufficiently low doesn’t seem valid to me.

---

> > ### Author Response · Authors · 2024-11-26
> > **Author Response Round 2 (Part 1)**
> >
> > Dear Reviewer GGnQ, we sincerely thank you for taking the time to respond to our reply despite your busy schedule. We are also greatly encouraged by your recognition of our idea. We also found that there may still be some potential misunderstandings. We highly value your efforts and respect your opinions and therefore we want to further clarify them.
> >
> > ---
> > **Q1**: I hope the authors acknowledge this fact in the manuscript. From my previous reading, I believe the authors are concealing the issue that *Theorem 1 does not propose a guarantee for the performance of clean samples*.
> >
> > **R1**: Thank you for this insightful comment!
> > - In our previous rebuttal, we simply want to clarify why Theorem 1 can guide the design of our method (for allowing altering the output domain), instead of trying to hide that Theorem 1 cannot propose a guarantee for the performance of benign samples.
> > - However, we do fully understand your concern. To avoid potential misunderstanding, we have revised our manuscript to clearly mention this fact in *Section 4.2.3*, as follows:
> >
> > ```!
> > Since Theorem 1 does not guarantee the model performance on clean samples, adding $\mathcal{L}_{ce}$ to maintain the utility of the model is necessary.
> > ```
> >
> > ---
> >
> > **Q2-1**: Thank you for providing additional experiments. However, there are too many supplementary experimental models. I am curious why these critical experiments were not included prior to submission. That said, I do acknowledge and appreciate these additional results.
> >
> > **R2-1**: Thank you for recognizing our additional experiments! We're glad that our new experiments alleviate your concerns.
> > - To be honest, we did not include this part of the experiment purely due to time and space constraints. As you can see in our upload history, we fought to deliver this paper almost until the last minute of submission DDL.
> > - Of course, we fully acknowledge that this is our problem and we should have started relevant experiments earlier. However, many times plans really cannot keep up with changes. I believe that as a senior expert (for being invited as the reviewer), you can fully understand this.
> >
> >
> > Thank you again for your constructive suggestions for adding these new experiments. We are willing to provide more details to further alleviate your concerns if necessary :)
> >
> > ---
> > **Q2-2**: Could you clarify the reasoning behind "*Considering that the logits distribution output by distillation can be challenging to align across different model architectures*"? There are existing methods that attempt to steal models even under hard-label scenarios. If soft labels are ineffective, how can this issue be addressed?
> >
> >
> > **R2-2**: Thank you for this insightful question! We are deeply sorry that our previous rebuttal may lead to your some potential misunderstandings that we want to clarify here.
> >
> > - In our paper, **we only want to initially explore how to extend our REFINE method to the black-box setting**. We included this discussion simply to make our paper more solid and comprehensive, as well as to provide a future research direction for follow-up works.
> >     - That is why we put this part in the appendix instead of mentioning it in the main content.
> >     - We did not even mention this contribution in our main content.
> > - In general, we aimed to show that we can probably extend our method to the black-box version by conducting (black-box) model-stealing at first.
> >     - We want to first clarify that **existing stealing methods [1, 2, 3] still suffered from relatively poor performance** (e.g., BA drop > 25% on CIFAR-100) under the black-box setting, even using the soft labels. The stealing would be much more difficult using only hard labels. PS: Model stealing cannot take a lot of time (or many epochs) like training from scratch, otherwise the stealing would be meaningless.
> >     - Besides, arguably, **the reprogramming process generally requires that the output distribution of the distilled surrogate model is similar to that of the (black-box) backdoored model**, instead of only having a similar general prediction performance.
> >     - Based on the above considerations, we used a similar model structure for our experiments in the appendix.
> > - However, we also fully understand your concern that this model selection may not be valid in practice, even though the defender can first use the side-channel attack to infer the structure of deployed models. As we mentioned before, we do admit that we did not completely address the black-box problem. We will further comprehensively discuss it in our future work.
> >
> > **Reference**
> > 1. Dual Student Networks for Data-Free Model Stealing.
> > 2. Towards Data-Free Model Stealing in a Hard Label Setting.
> > 3. Fully Exploiting Every Real Sample: SuperPixel Sample Gradient Model Stealing

---

> > > ### Author Response · Authors · 2024-11-26
> > > **Author Response Round 2 (Part 2)**
> > >
> > > ---
> > >
> > > **Q3**: I partially agree with the response, but I would only fully accept it if supported by references to relevant literature.
> > >
> > > **R3**: Thank you for the comment. The setting of adaptive attacks is widely recognized in the field of backdoor attacks [1, 2]. A NeurIPS competition [3] also utilized the same setting which the adversary knows the defense methods (Neural Cleanse, MNTD, and ABS) but does not know the implementation parameters.
> > >
> > > This setting is practical since **the backdoor defense is a post-hoc defense conducted after the adversary injecting backdoors into the model**. Therefore, it is unlikely for the adversary to have full knowledge of the implementation details of the defense.
> > >
> > > **References**
> > > 1. Towards Reliable and Efficient Backdoor Trigger Inversion via Decoupling Benign Features.
> > > 2. On Model Outsourcing Adaptive Attacks to Deep Learning Backdoor Defenses.
> > > 3. NeurIPS 2022 Competition: The Trojan Detection Challenge.
> > >
> > > ---
> > >
> > > **Q4**: I want to emphasize that the authors themselves admitted "*this is caused by different experimental settings.*" This suggests an intentional effort to degrade the performance of other methods. Even though this is in the motivation section, I question whether the comparisons in the experimental section are fair.
> > > Additionally, do the updated images ensure optimal reproducibility? From what I observed, the reverse-engineered images largely retain the characteristics of the trigger. This undermines the claim that trigger reverse-engineering is ineffective. I believe the motivation section needs to be rewritten to ensure it highlights the key or universal issues with these approaches, rather than adjusting experimental settings to worsen their performance, or worse, omitting such details and misleading readers.
> > >
> > > **R4**: Thank you for this insightful comment! We are deeply sorry that our previous rebuttal may lead to your some potential misunderstandings that we want to clarify here.
> > > - We would like to clarify that **we did not intentionally adjust the experimental settings to lower our baseline results**.We respectfully note that this action was unnecessary because the performance of our method is still better than that of BTI-DBF even using results in its original paper.
> > >     - In the experimental section, we applied the same, more reasonable attack settings to each defense tested. **Specifically, we applied sufficient data augmentation to the attacks.** This method is better suited to the model's real training scenario, as explained in [1, 2].
> > >     - **The attacks tested in the original baseline paper did not include sufficient data augmentation**, which we consider unreasonable. It should not be claimed that we intentionally lowered the baseline results simply because we adopted a more reasonable attack experiment settings.
> > > - For the results obtained by BTI-DBF in Figure 3:
> > >     - **For the BadNets attack**, while BTI-DBF is able to roughly reverse-engineer the trigger pattern, its purification process introduces significant distortion to the original image. **This result is consistent with the figures in the original BTI-DBF paper**, validating the reliability of our experimental results. Moreover, this does not conflict with our claim that **BTI-based defenses struggle to reverse the ground-truth trigger**.
> > >     - **For the Blended attack**, BTI-DBF fails to reverse-engineer the trigger pattern. This result demonstrates the limitations of BTI-based defenses.
> > >     - Thus, as shown in Figure 3, BTI-DBF can **partially reverse** the trigger of BadNets (consistent with their original paper) but **fails to reverse** the trigger of Blended.
> > >
> > > We respectfully argue that we have presented the most accurate experimental results without intentionally lowering its performance, nor have we misled readers in any way. We believe that the length of this article and our discussion of potential pitfalls adequately demonstrate our integrity. However, we are very willing to provide more details and discussions if you still have any concerns. Again, we do highly value your efforts and respect your opinions :)
> > >
> > > **References**
> > > 1. A survey on Image Data Augmentation for Deep Learning.
> > > 2. Image data augmentation approaches: A comprehensive survey and future directions.
> > >
> > > ---

---

> > > > ### Author Response · Authors · 2024-11-26
> > > > **Author Response Round 2 (Part 3)**
> > > >
> > > > **Q5**: The results for low-pixel datasets seem inconsistent with the results presented in the pipeline figure. Are the results for the 224×224 datasets still of the same nature? Where exactly does the pipeline figure's reference come from?
> > > >
> > > > **R5**: Thank you for this insightful comment!
> > > > - The samples shown in our pipeline figure are from ImageNet (due to the resolution consideration), so the results differ from those of CIFAR-10.
> > > > - To further alleviate your concerns, in the revised paper, on *Page 26, Figure 8*, we show the feature maps of ImageNet images after passing through the input transformation module. **Similar to the phenomenon observed in CIFAR-10, samples from the same class exhibit visual similarities after transformation to some extent.**
> > > >
> > > > ---
> > > >
> > > > **Q6**: Additionally, regarding the loss conflict mentioned in Q3, I didn’t fully grasp what the authors intended to convey (I feel somewhat lost regarding the detailed content of the manuscript, and I believe a more straightforward clarification would be helpful). According to optimization theory, optimization can proceed regardless of how low the loss is. The claim in the paper that failure is caused by the refine loss being sufficiently low doesn’t seem valid to me.
> > > >
> > > > **R6**: Thank you for this constructive suggestion! We sincerely apologize for not clearly explaining the conflict between *backdoor loss* $L_{b}$ and *REFINE loss* $L_{refine}$. Here, we provide further clarification.
> > > >
> > > > - In general, **optimizing two different modules** (i.e., the input transformation module and the backdoor model) **simultaneously will generally yield relatively worse results on one of the module** (i.e., the backdoor model) **than optimizing just that one**. This is mostly because the directions of the gradients of the two modules are not always consistent.
> > > > - Specifically, the process of our adaptive attack is as follows:
> > > >     1. The attacker trains a standard backdoor model (with $L_{b}$).
> > > >     2. The attacker assigns a random hard-coded mapping function to the output mapping module, and randomly initializes an input transformation module. Here, the attacker is unaware of the hard-coded function that the defender will use, as this function can be considered as a randomly initialized parameter.
> > > >     3. Without fixing the standard backdoor model, the attacker simultaneously optimizes both the backdoor model (with $L_{b} + \gamma L_{refine}$ ) and the input transformation module (with $L_{refine}$ ).
> > > > - In Step 1, the attacker optimizes to achieve a very low value for $L_{b}$. However, during the optimization process in Step 3, the initially large value of $L_{refine}$ (resulting in a higher weight) causes simultaneous adjustments to parameters of the backdoor model and the input transformation module, leading to an inevitable increase in $L_{b}$.
> > > >
> > > > We are very willing to provide more details and discussions if you still have any concerns. Again, we do highly value your efforts and respect your opinions :)
> > > >
> > > > ---

---

> > > > > ### Comment · Reviewer_GGnQ · 2024-11-26
> > > > > **Final Decision**
> > > > >
> > > > > Thank you for your response. I have carefully reviewed your response, and I believe that my concerns regarding the fairness of comparisons and the black-box experiments have been addressed. As a result, I no longer intend to lower my score and have decided to maintain it. But I am unable to find compelling evidence that would justify a higher score.

---

> ### Author Response · Authors · 2024-11-26
> **Thank you and a small request for more information**
>
> Dear Reviewer GGnQ,
>
> We sincerely thank you again for your constructive and insightful comments. Your timely feedback greatly helps us in improving the quality of this work. We are also glad to see that our rebuttal addressed all your concerns, although our submission/previous rebuttal may have led you to some potential misunderstandings that make you upset or even angry.
>
> We notice that you have decided to maintain the original score, although we have addressed all your previous concerns. Although we feel very frustrated about this, we do fully respect your decision since we understand it is due to your own judgments and is ultimately your right as a reviewer.
>
> Nevertheless, we would be very grateful if you could kindly share your main criteria and the decision process behind your assessment (i.e., weak reject) of our paper. We would also greatly appreciate a more detailed explanation of both the strengths and remaining weaknesses of our work, as well as the key areas that you think we need to improve. Your feedback would be invaluable to us, as it will guide us in refining this work and in determining how to proceed with follow-up research.
>
> In any case, your previous comments have provided us with very valuable guidances, for which we are very grateful. We wish you all the best :)
>
> Best Regards,
>
> Paper4208 Authors

---

### Official Review · Reviewer_Q8vQ · 2024-11-02

**Soundness:** 2
**Presentation:** 3
**Contribution:** 2
**Rating:** 5
**Confidence:** 4

**Summary:**

This paper  proposes REFINE, an inversion-free backdoor defense method based on model reprogramming. Based on the research, the authors revisit existing pre-processing-based backdoor defenses and reveal their limitations. The proposed defense, REFINE, introduces trainable input transformation and output mapping modules for reprogramming and incorporates cross-entropy and supervised contrastive losses to enhance defense performance. Extensive experiments are conducted to validate the effectiveness of REFINE.

**Strengths:**

1. Revisit the pre-processing defenses against backdoor attacks and reveal their limitations. The pre-processing-based defense is important to protect model security while not changing model structure or weights.
2. Propose a pre-processing defense against backdoor attacks, which seems to be simple but effective.
3. Conduct extensive experiments to demonstrate the effectiveness of the proposed defense.

**Weaknesses:**

1. The claim of the limitations of prior works are subjective and confusing. For the first limitation, the authors think "transformation-based backdoor defenses methods face a trade-off between utility and effectiveness". So, can the proposed defense overcome this limitation? From the design and experimental results, REFINE also suffer from the same problem. Otherwise, the BA with REFINE should be same with  original model. Moreover, the authors try to utilize experiments to validate their claims. However, the experimental setting is not fair enough. For example, the authors mention ShrinkPad as the baseline. However, in that paper, ShrinkPad is not the best defense method. Also, for the second limitation, the authors do not analyse the SOTA work, and the shown experimental results are different from the original paper, e.g., the experiments with BADNET shown in Fig.3.
2. The categories of the mentioned defenses are not sound enough. According to the manuscript, the BTI-based defense (e.g., NC) reverses the backdoor trigger to eliminate the trigger in the inputs. However, such a kind of defense does not only work in this way. Thus, the comparison seems to be not fair enough.
3. The experiments do not involve some SOTA works. e.g, [1, 2]. Moreover, the experimental results differ from the original paper a lot, e.g., BTI-DBF. Please explain why.
[1] Neural Polarizer: A Lightweight and Effective Backdoor Defense via Purifying Poisoned Features
[2] Black-box Backdoor Defense via Zero-shot Image Purification

**Questions:**

1. Please improve the soundness of the limitation study.
2. Please discuss the reasonability of the categories of the mentioned defenses.
3. Please explain the different experimental results.

---

> ### Author Response · Authors · 2024-11-20
> **Author Response (Part 1)**
>
> Dear Reviewer Q8vQ, thank you very much for your careful review of our paper and thoughtful comments. We are encouraged by your positive comments on our **efforts to revisit the limitations of the pre-processing defenses**, **simple but effective method**, **extensive experiments**, and **good presentation**. We hope the following responses can help clarify potential misunderstandings and alleviate your concerns.
>
> ---
>
> **Q1-1**: The claim of the limitations of prior works are subjective and confusing. For the first limitation, the authors think "transformation-based backdoor defenses methods face a trade-off between utility and effectiveness". So, can the proposed defense overcome this limitation? From the design and experimental results, REFINE also suffer from the same problem. Otherwise, the BA with REFINE should be same with original model.
>
> **R1-1**: Thank you for this insight comment! We sincerely apologize for any misunderstandings and confusion caused by our insufficient explanation. We hereby provide further clarification on this issue.
>
> - We admit that we did not fully address this problem. In our paper, we use the words 'break' to indicate that **we alleviate this problem instead of completely addressing this problem**.
> - We argue that **we have already made a very significant improvement in alleviating this problem compared to the baseline method**. Specifically, The experimental results in our paper indicate that REFINE results in a benign accuracy drop of no more than 3%. Additional experiments further demonstrate that, in the VGG16, DenseNet121, and InceptionV3 architectures, REFINE actually improves benign accuracy.
> - To avoid potential misunderstandings, we modify our statement in the introduction by adding 'to a large extent', as follows:
> ```!
> REFINE can significantly alter the input domain while preserving the model's accuracy to a large extent for it allows changing the output domain.
> ```
> - To further justify the effectiveness of our method, **we conduct additional experiments on five different network structures**, including ResNet18, ResNet50, VGG16, DenseNet121 and InceptionV3. We select three representative types of backdoor attacks, including BadNets, WaNet, and Adaptive-Patch (dubbed ‘Adaptive’). We conduct experiments on the CIFAR-10 dataset. We compare the defense performance of our REFINE with the most advanced transformation-based defense (i.e., BDMAE).
> - As shown in Tables 1 to 5, REFINE effectively defends against three representative attacks across five different network architectures, significantly outperforming BDMAE. Specifically, under the REFINE defense, the benign accuracy (BA) drop is less than 1.5%, while the backdoor attack success rate (ASR) reduces to below 3%. Meanwhile, on the VGG16, DenseNet121, and InceptionV3 model architectures, REFINE not only reduces the ASR but also even improves the original model's BA. This directly proves that REFINE is not limited by the trade-off faced by transformation-based defenses. **The additional experimental results verify the effectiveness of REFINE.**
>
> We provide more details in *Appendix E.1* of our revision.
>
> **Table 1**. The performance (\%) of REFINE and BDMAE on **ResNet18**. **ND** means that no defense is applied. The best results are **boldfaced**.
> |          | BA (ND) | ASR (ND) | BA (BDMAE) | ASR (BDMAE) | BA (REFINE) | ASR (REFINE) |
> | -------- | ------- | -------- | ---------- | ----------- | ----------- | ------------ |
> | BadNets  | 91.18   | 100      | 90.48      | 14.81       | **90.50**   | **1.05**     |
> | WaNet    | 91.29   | 99.91    | 89.87      | 99.93       | **90.64**   | **1.93**     |
> | Adaptive | 89.62   | 100      | 89.40      | 49.18       | **90.54**   | **1.23**     |
>
> **Table 2**. The performance (\%) of REFINE and BDMAE on **ResNet50**. **ND** means that no defense is applied. The best results are **boldfaced**.
> |          | BA (ND) | ASR (ND) | BA (BDMAE) | ASR (BDMAE) | BA (REFINE) | ASR (REFINE) |
> | -------- | ------- | -------- | ---------- | ----------- | ----------- | ------------ |
> | BadNets  | 91.91   | 100      | **91.04**  | 10.99       | 90.71       | **1.53**     |
> | WaNet    | 91.70   | 99.98    | 89.83      | 99.89       | **91.09**   | **0.35**     |
> | Adaptive | 89.59   | 85.11    | 89.06      | 35.91       | **90.05**   | **2.19**     |
>
> **Table 3**. The performance (\%) of REFINE and BDMAE on **VGG16**. **ND** means that no defense is applied. The best results are **boldfaced**.
> |          | BA (ND) | ASR (ND) | BA (BDMAE) | ASR (BDMAE) | BA (REFINE) | ASR (REFINE) |
> | -------- | ------- | -------- | ---------- | ----------- | ----------- | ------------ |
> | BadNets  | 84.44   | 99.36    | 84.25      | 18.32       | **86.86**   | **1.62**     |
> | WaNet    | 84.75   | 99.15    | 83.36      | 99.25       | **86.41**   | **2.39**     |
> | Adaptive | 84.98   | 99.99    | 84.69      | 40.09       | **86.63**   | **2.04**     |

---

> ### Author Response · Authors · 2024-11-20
> **Author Response (Part 2)**
>
> **Table 4**. The performance (\%) of REFINE and BDMAE on **DenseNet121**. **ND** means that no defense is applied. The best results are **boldfaced**.
> |          | BA (ND) | ASR (ND) | BA (BDMAE) | ASR (BDMAE) | BA (REFINE) | ASR (REFINE) |
> | -------- | ------- | -------- | ---------- | ----------- | ----------- | ------------ |
> | BadNets  | 86.40   | 99.99    | 86.05      | 11.85       | **89.44**   | **0.96**     |
> | WaNet    | 86.31   | 98.77    | 85.42      | 98.91       | **88.74**   | **0.88**     |
> | Adaptive | 85.16   | 100      | 84.45      | 45.36       | **88.74**   | **0.35**     |
>
> **Table 5**. The performance (\%) of REFINE and BDMAE on **InceptionV3**. **ND** means that no defense is applied. The best results are **boldfaced**.
> |          | BA (ND) | ASR (ND) | BA (BDMAE) | ASR (BDMAE) | BA (REFINE) | ASR (REFINE) |
> | -------- | ------- | -------- | ---------- | ----------- | ----------- | ------------ |
> | BadNets  | 90.46   | 99.97    | 90.61      | 80.51       | **91.03**   | **0.75**     |
> | WaNet    | 90.09   | 99.73    | 89.64      | 99.76       | **91.01**   | **0.54**     |
> | Adaptive | 88.58   | 13.53    | 88.54      | 13.52       | **90.36**   | **0.54**     |
>
> ---
>
> **Q1-2**: Moreover, the authors try to utilize experiments to validate their claims. However, the experimental setting is not fair enough. For example, the authors mention ShrinkPad as the baseline. However, in that paper, ShrinkPad is not the best defense method. Also, for the second limitation, the authors do not analyse the SOTA work, and the shown experimental results are different from the original paper, e.g., the experiments with BADNET shown in Fig.3.
>
> **R1-2**: Thanks for your insightful comment! We are deeply sorry for potential misunderstandings and concerns caused by our limitation analysis sections. We would like to provide some additional clarifications here.
> - As mentioned in our paper, pre-processing-based defenses can be divided into two categories: transformation-based defenses and BTI-based defenses.
>     - In Section 3.2, we discussed and experimentally demonstrated the limitation of transformation-based defenses, where we chose ShrinkPad and BDMAE as examples.
>         - ShrinkPad is not the most effective transformation-based defense, but **it is one of the most classical and representative transformation-based methods**.
>         - **We have included the state-of-the-art method (i.e., BDMAE) in our analysis**. Its superiority is supported by Table 3 in the referenced paper [1]
>     - Additionally, we are not sure of what you meant by "in that paper". We assume you are referring to our paper. If we misunderstood, please kindly correct us, and we are very willing to alleviate your concerns.
>     - In Section 3.3, we discussed and visually demonstrated the limitations of BTI-based defenses. **We believe that BTI-DBF (ICLR 2024 Spotlight) represents the current SOTA work on backdoor trigger inversion, which is validated in a Spotlight paper [2] at NeurIPS 2024**, as follows:
> ```!
> We evaluate several mainstreamed purification methods, including pruning-based defense ANP [49]; robust fine-tuning defense I-BAU [53] (referred to as BAU for short), FT-SAM [55] (referred to as SAM for short), FST [32], as well as the state-of-the-art trigger reversing defense BTI-DBF [50] (referred to as BTI for short).
> ```
> - The difference between the experimental results presented in Figure 3 and the original BTI-DBF paper [3]:
>     - In general, **this is caused by different experimental settings**.
>         - In the original BTI-DBF's BadNets attack setup, only normalization was used as the data pre-processing function.
>         - In our BadNets attack setup, we add two data augmentation techniques, including random cropping and random flipping.
>         - Compared to the BTI-DBF's setup, our configuration is more reasonable. This is because attackers can manipulate the model’s training process, including data augmentation, to enhance the robustness of the backdoor attack.
>         - **Applying data augmentation to the attack method makes it more difficult to invert the backdoor trigger**. For example, in the case of BadNets attack, operations such as random cropping and padding increase the number of triggers that can activate the backdoor, making it harder to identify the ground-truth trigger used by the attacker.
>     - Nevertheless, we have adjusted the experimental results shown in Figure 3 to make them more clear and more intuitive.
>
>
> **References**
>
> 1. Black-box backdoor defense via zero-shot image purification.
> 2. Uncovering, Explaining, and Mitigating the Superficial Safety of Backdoor Defense.
> 3. Towards reliable and efficient backdoor trigger inversion via decoupling benign features.

---

> ### Author Response · Authors · 2024-11-20
> **Author Response (Part 3)**
>
> ---
>
> **Q2**: The categories of the mentioned defenses are not sound enough. According to the manuscript, the BTI-based defense (e.g., NC) reverses the backdoor trigger to eliminate the trigger in the inputs. However, such a kind of defense does not only work in this way. Thus, the comparison seems to be not fair enough.
>
> **R2**: Thank you for this insightful comment! We apologize for not accurately explaining the complete paradigm of BTI-based defenses.
> - We admit that backdoor trigger inversion can also be used for other types of defense methods, such as backdoor elimination.
> - However, **BTI can also be applied in designing highly effective pre-processing-based defenses**, as shown in [1]. We never intended to claim that BTI is limited to pre-processing-based defenses.
> - We have also compared our method with transformation-based backdoor defenses besides BTI-based defenses to ensure a fair comparison.
> - We believe our experimental setup is fair.
>     - **The defenders' capabilities are the same**, as they are only allowed to modify the input samples and cannot alter the internal parameters of the model.
>     - **The defenders' objective is the same**: to map the backdoor images from the poisoned input domain to the benign input domain as effectively as possible. It would be unfair to compare our method with all paradigms based on BTI defenses.
>     - We also humbly mention that our method only requires unlabeled instead of labeled samples. In other words, our defense requires even fewer capacities than some of the baseline methods (e.g., FeatureRE).
>
> We provide further clarification on BTI-based defense methods in *Appendix I.2* to avoid any potential misunderstandings.
>
> **Reference**
>
> 1. Towards reliable and efficient backdoor trigger inversion via decoupling benign features.

---

> ### Author Response · Authors · 2024-11-20
> **Author Response (Part 4)**
>
> ---
>
> **Q3**: The experiments do not involve some SOTA works. e.g, [1, 2]. Moreover, the experimental results differ from the original paper a lot, e.g., BTI-DBF. Please explain why. [1] Neural Polarizer: A Lightweight and Effective Backdoor Defense via Purifying Poisoned Features [2] Black-box Backdoor Defense via Zero-shot Image Purification
>
> **R3**: Thank you for these insightful comments!
>
> - As mentioned in **R1-2**, **we believe that BTI-DBF (ICLR 2024 Spotlight) is a SOTA work, which is confirmed in [3]**.
> - As we explained in **R1-2**, the differences between the BTI-DBF experimental results and those presented in its original paper are largely **due to the diversity of triggers caused by data augmentation operations which are widely used in practice**.
> - We carefully review the Neural Polarizer [1], which achieves effective defense by training an additional linear transformation (e.g., neural polarizer), using only a small portion of clean data without modifying the model parameters.
>     - However, based on our classification of existing backdoor defense methods in Section 2.2, **Neural Polarizer falls under the category of backdoor elimination methods rather than pre-processing-based defenses** since it incorporates additional modules within the original model instead of outside of it. As such, we did not include it in our baseline methods for comparison. Even though, [1] is still an outstanding work towards effective backdoor defense. We will add a discussion of [1] in *Appendix I.2*.
>     - Nevertheless, we still consider Neural Polarizer to be an excellent defense method, and **we have now provided a brief introduction to this method in the related work section of the appendix** and highlighted it in blue. :)
> - We were aware of ZIP[2] defense in advance, but for the following reasons, we did not include it in our baseline methods.
>     - The execution logic of the ZIP open-source code is quite complex, making it difficult for us to integrate baseline attack methods beyond those originally supported, such as Adaptive-Patch.
>     - As mentioned earlier, BTI-DBF is a SOTA work, and we believe that comparing with it is sufficient to a large extent. Furthermore, as presented in the original ZIP paper, BDMAE outperforms ZIP in defending against certain attacks, such as the BadNets attack. This indicates that the defense performance of ZIP and BDMAE is comparable. Therefore, we chose BDMAE as our baseline method.
> - To further alleviate your concern, we conducted additional experiments using the attack methods supported by the original ZIP repository.
>     - Specifically, we trained ResNet-18 backdoor models on the CIFAR-10 dataset using four different attack methods, including BadNets, Blended, WaNet, and PhysicalBA.
>     - **As shown in Table 6, compared to ZIP, REFINE demonstrates better defense performance against all four attack methods.** This demonstrates the effectiveness of REFINE again.
>
> We provide more details in *Appendix D.2* of our revision.
>
> **Table 6**. The performance (\%) of REFINE and ZIP against four different attacks. **ND** means that no defense is applied. The best results are **boldfaced**.
> |            | BA (ND) | ASR (ND) | BA (ZIP) | ASR (ZIP) | BA (REFINE) | ASR (REFINE) |
> | ---------- | ------- | -------- | -------- | --------- | ----------- | ------------ |
> | BadNets    | 91.18   | 100      | 84.22    | 5.53      | **90.50**   | **1.05**     |
> | Blended    | 90.64   | 98.18    | 84.68    | 8.64      | **90.30**   | **1.00**     |
> | WaNet      | 91.29   | 99.91    | 85.19    | 15.46     | **90.64**   | **1.93**     |
> | PhysicalBA | 93.67   | 100      | 85.07    | 10.91     | **91.17**   | **0.78**     |
>
> **References**
>
> 1. Neural polarizer: a lightweight and effective backdoor defense via purifying poisoned features.
> 2. Black-box backdoor defense via zero-shot image purification.
> 3. Uncovering, Explaining, and Mitigating the Superficial Safety of Backdoor Defense.
>
> ---
>
> **Q4**: Please improve the soundness of the limitation study.
>
> **R4**: Thank you for this constructive suggestion! Please refer to **R1-1** and **R1-2** for detailed explanation :)
>
> ---
>
> **Q5**: Please discuss the reasonability of the categories of the mentioned defenses.
>
> **R5**: Thank you for this constructive suggestion! Please refer to **R2** for detailed explanation :)
>
> ---
>
> **Q6**: Please explain the different experimental results.
>
> **R6**: Thank you for this constructive suggestion! Please refer to **R3** for detailed explanation :)
>
> ---

---

> ### Author Response · Authors · 2024-11-23
> **Thanks to Reviewer Q8vQ**
>
> Please allow us to thank you again for reviewing our paper and the valuable feedback, and in particular for recognizing the strengths of our paper in terms of the efforts to revisit the limitations of the pre-processing defenses, simple but effective method, extensive experiments, and good presentation.
>
> Please let us know if our response and the new experiments have properly addressed your concerns. We are more than happy to answer any additional questions during the post-rebuttal period. Your feedback will be greatly appreciated.

---

> ### Author Response · Authors · 2024-11-24
> **A Gentle Reminder of the Final Feedback**
>
> Thank you very much again for your initial comments. They are extremely valuable for improving our work. We shall be grateful if you can have a look at our response and modifications, and please let us know if anything else that can be added to our next version.

---

> ### Author Response · Authors · 2024-11-25
> **A Second Reminder of the Post-rebuttal Feedback**
>
> Dear Reviewer Q8vQ,
>
> We greatly appreciate your initial comments. We totally understand that you may be extremely busy at this time. But we still hope that you could have a quick look at our responses to your concerns. We appreciate any feedback you could give to us. We also hope that you could kindly update the rating if your questions have been addressed. We are also happy to answer any additional questions before the rebuttal ends.
>
> Best Regards,
>
> Paper4208 Authors

---

> ### Author Response · Authors · 2024-11-28
> **A Third Reminder for Post-rebuttal Feedback**
>
> Dear Reviewer Q8vQ,
>
> We sincerely thank you for your time and valuable comments during the review process. As the paper revision stage has now ended, we kindly hope you could take a moment to review our responses and updates. If you have any remaining concerns, we would be happy to address them further.
>
> Thank you again for your understanding and support.
>
> Best Regards,
> Paper4208 Authors

---

> ### Author Response · Authors · 2024-12-02
> **Reminder of the Post-rebuttal Feedback and Summary of Our Response**
>
> Dear Reviewer Q8vQ,
>
> Thank you for your time and effort in evaluating our work. We greatly appreciate your initial comments. Your insights and suggestions are extremely valuable to us.
>
> Given that we have only one day left for discussion, we are hoping to receive any additional feedback or question you might have at your earliest convenience. Your expertise would be of great help to us in improving the quality and rigor of our work.
>
> To facilitate the discussion, we would like to summarize our response as follows.
>
> - **We further analyzed and clarified on the limitations of prior pre-processing-based defenses.** Additional experiments demonstrated that REFINE effectively overcomes these limitations and defends against backdoor attacks across different model architectures.
> - We explained the reasons for the performance difference of BTI-DBF compared to its original paper and improved Figure 3 in our revised paper.
> - **We explained why BTI-based defense can be utilized as a pre-processing defense** and justified the fairness of our comparison experimental setup from multiple perspectives.
> - **We addressed the questions regarding SOTA methods**, as follows:
>     - Regarding Neural Polarizer [1], we explained the reason for not including it in the comparison and cited it in our revised paper.
>     - Regarding ZIP [2], we clarified why its experimental results were initially omitted and conducted comparison experiments with REFINE under four different attacks.
>     - We also clarified that BDMAE and BTI-DBF are existing SOTA methods, as supported by [2, 3].
>
> If our responses address your concerns, we kindly request that you reconsider your evaluations. We would also be grateful for any additional comments or suggestions you might have to refine our work.
>
> Best regards,
>
> Paper4208 Authors
>
> **Reference**
>
> 1. Neural polarizer: a lightweight and effective backdoor defense via purifying poisoned features.
> 2. Black-box backdoor defense via zero-shot image purification.
> 3. Uncovering, Explaining, and Mitigating the Superficial Safety of Backdoor Defense.
>
> ---

---

### Official Review · Reviewer_p35s · 2024-11-04

**Soundness:** 3
**Presentation:** 2
**Contribution:** 2
**Rating:** 6
**Confidence:** 4

**Summary:**

This paper analyzes existing pre-processing-based backdoor defense methods and their limitations. Then, a simple yet effective method is proposed. This method utilizes model reprogramming techniques rather than model retraining, which not only eliminates backdoor attacks but also maintains the model's performance. The evaluation of different backdoor attacks also demonstrates the effectiveness of the proposed method.

**Strengths:**

1. A theoretical analysis demonstrates that the effect of backdoor defenses is bounded by the distance of the output features before and after the preprocessing. Therefore, existing methods can not break the trade-off between the model utility and the defense effectiveness.
2. The proposed method is novel and interesting. By integrating model reprogramming techniques, they only need to change the model input without changing the model parameters to achieve backdoor elimination, and it does not affect the original performance of the model.

**Weaknesses:**

1. The authors discuss the pre-processing defense methods, i.e., input-transformation defenses and BTI-based defenses, and analyze their limitations in details. However, the proposed methods actually belong to the input transformation-based method. This paper also spend a large amount of time to analyze and compare BTI methods with the proposed method, which makes it hard to read.
2. This paper assumes that they have access to an unlabeled dataset that is independent and identically distributed to the training dataset of the pre-trained model. The authors need to clarify how to acquire this dataset. In addition, since the performance of model reprogramming methods is related to the distribution of input data, if the author cannot obtain data with the same distribution, will it affect the performance of the model?
3. The experiment is pretty insufficient. REFINE is only evaluated with the ResNet-18 model. The evaluation under more complex models (such as ResNet-50) or other architecture models (such as InceptionV3 or VGG16) is lack.

**Questions:**

1. The two sentences (i.e., "by first obtaining ... outcomes" and "the internet ... poisoned samples)") in the Section 1 contradict each other. What is the difference between prior information and prior knowledge?
2. This paper focuses on the dirty-image backdoor attacks. But considering the recent clean-image backdoor attacks that do not have trigger features, can the proposed defense method work?
3. In Section 3.2, the "Pad Size" and "Mask Ratio" are not defined before. it is necessary to clarify.
4. In this paper, there are many long sentences, which makes it hard to understand. Such as "Additionally, we treat ... the transformation" on Page 4.
5. In Section 3.2, the authors did not show the dataset used for limitation analysis. Please clarify it.
6. In Section 4.2.1, the authors present that traditional model reprogramming methods are insufficient to remove triggers. I suggest that the authors explain why traditional methods are insufficient in detail.
7. The discussion about the feature distance changes in the proposed input transformation module is lacking. It is better to add it to highlight the function of this module.

---

> ### Author Response · Authors · 2024-11-20
> **Author Response (Part 1)**
>
> Dear Reviewer p35s, thank you very much for your careful review of our paper and thoughtful comments. We are encouraged by your positive comments on our **theoretical analysis**, **novel and interesting method**, and **good soundness**. We hope the following responses can help clarify potential misunderstandings and alleviate your concerns.
>
> ---
>
> **Q1**: The authors discuss the pre-processing defense methods, i.e., input-transformation defenses and BTI-based defenses, and analyze their limitations in details. However, the proposed methods actually belong to the input transformation-based method. This paper also spend a large amount of time to analyze and compare BTI methods with the proposed method, which makes it hard to read.
>
> **R1**: Thank you for this insightful comment! We sincerely apologize for failing to clearly express the connection between our motivation and the proposed method. We would like to provide additional clarifications as follows.
>
> - We acknowledge that our method is a novel transformation-based defenses, but **the BTI-based defense is also closely related to our motivation**.
> - In general, pre-processing-based backdoor defense can be categorized into two types: transformation-based defenses and backdoor trigger inversion (BTI)-based defenses. **Fundamentally, both types of defenses modify the samples in the image domain.**
>     - Transformation-based defenses distort trigger patterns by applying random transformations to the input or reconstructing the input using generative models. These approaches represent a paradigm that **maps backdoor inputs from the poisoned image domain to a transformed image domain**.
>     - **BTI-based defenses aim to leverage the inverted triggers to precisely map backdoor inputs from the poisoned image domain to the benign image domain**. Specifically, BTI-based defenses first train a trigger generator $T(\cdot)$, which can generate a trigger pattern $T(x)$ that mimics the attacker's trigger based on the input image $x$. During the defense process, subtracting the trigger pattern $T(x)$ from each input image $x$ yields a purified image $x - T(x)$. This process effectively maps backdoor inputs from the poisoned image domain to the benign image domain.
> - **Our Motivations from the Domain Aspect**:
>     - **For transformation-based defenses, fixing the output domain limits the transformations in the image domain.** Specifically, overly large transformations reduce the model's benign accuracy, while small transformations fail to effectively reduce the backdoor attack success rate. Mapping the poisoned image domain to an appropriate transformed domain presents a trade-off.
>     - **For BTI-based defenses, the inherent difficulty of trigger inversion leads to inaccurate image domain mapping.** Due to the lack of prior knowledge about the attack, existing BTI methods struggle to invert the ground-truth trigger. Therefore, using an incorrect trigger makes it challenging to purify the poisoned image domain to the benign image domain.
>     - Based on the above understanding, **to increase the flexibility of image domain transformation and avoid the need for accurate transformations, we map the output domain** and propose our method, REFINE. REFINE can significantly alter the input domain while preserving the model’s accuracy by allowing changes to the output domain.

---

> ### Author Response · Authors · 2024-11-20
> **Author Response (Part 2)**
>
> ---
>
> **Q2**: This paper assumes that they have access to an unlabeled dataset that is independent and identically distributed to the training dataset of the pre-trained model. The authors need to clarify how to acquire this dataset. In addition, since the performance of model reprogramming methods is related to the distribution of input data, if the author cannot obtain data with the same distribution, will it affect the performance of the model?
>
> **R2**: Thank you for this insightful comment! We are deeply sorry for not clearly explaining the scenario of obtaining the unlabeled benign dataset. We hereby provide further clarification on how the unlabeled dataset is acquired, and demonstrate the defense performance of REFINE under different data distributions through additional experiments.
>
> - Based on the general consensus in the community, **obtaining independent and identically distributed (i.i.d.) unlabeled datasets is relatively easy**.  For example, in real-world scenarios, large amounts of unlabeled data can be collected, with only a small subset being outsourced to third parties for labeling. We can assume that the unlabeled part and the labeled part are independent and identically distributed.
> - To address your concerns about REFINE's performance under different distribution scenarios, we conducted an additional experiment.
>     - We train a ResNet18 model on the CIFAR10 dataset using the BadNets attack. For defense, we trained the input transformation module of REFINE using CINIC10, **a dataset with the same categories as CIFAR10 but a different data distribution**.
>     - As shown in Table 1, **REFINE is still highly effective in reducing the attack success rate (ASR < 1.5\%) while maintaining the model's benign accuracy (BA drop < 3\%)**.
>     - This favorable result is due to the fact that REFINE first assigns pseudo-labels to the unlabeled benign samples using the original model, and then trains the input transformation module based on these pseudo-labels.
> - We will discuss how to further improve our method in non-i.i.d scenarios in our future work.
>
> We provide more details in *Appendix E.5* of our revision.
>
> **Table 1**. The performance (\%) of REFINE in scenarios with different data distributions. **ND** means that no defense is applied.
> |          | BA (ND) | ASR (ND) | BA (REFINE) | ASR (REFINE) |
> | -------- | ------- | -------- | ----------- | ------------ |
> | BadNets  | 91.18   | 100      | 88.39       | 1.40         |

---

> ### Author Response · Authors · 2024-11-20
> **Author Response (Part 3)**
>
> ---
>
> **Q3**: The experiment is pretty insufficient. REFINE is only evaluated with the ResNet-18 model. The evaluation under more complex models (such as ResNet-50) or other architecture models (such as InceptionV3 or VGG16) is lack.
>
> **R3**: Thank you for these constructive suggestions! We are deeply sorry for not demonstrating the effectiveness of our method across different network structures in our previous submission. We do agree with you that it is beneficial to verify the universality of our defense across different network structures.
> - To address your concerns, following your suggestions, **we conduct additional experiments on five different network structures**, including ResNet18, ResNet50, VGG16, DenseNet121 and InceptionV3. We select three representative types of backdoor attacks, including BadNets, WaNet and Adaptive-Patch (dubbed ‘Adaptive’). We conduct experiments on the CIFAR-10 dataset. We compare the defense performance of our REFINE with the most advanced transformation-based defense (i.e., BDMAE).
> - As shown in Tables 2 to 6, REFINE effectively defends against three representative attacks across five different network architectures, significantly outperforming BDMAE. Specifically, under the REFINE defense, the benign accuracy (BA) drop is less than 1.5%, with some cases showing an increase in BA. Meanwhile, the backdoor attack success rate (ASR) reduces to below 3%. **The additional experimental results verify the effectiveness of REFINE.**
>
> We provide more details in *Appendix E.1* of our revision.
>
> **Table 2**. The performance (\%) of REFINE and BDMAE on **ResNet18**. **ND** means that no defense is applied. The best results are **boldfaced**.
> |          | BA (ND) | ASR (ND) | BA (BDMAE) | ASR (BDMAE) | BA (REFINE) | ASR (REFINE) |
> | -------- | ------- | -------- | ---------- | ----------- | ----------- | ------------ |
> | BadNets  | 91.18   | 100      | 90.48      | 14.81       | **90.50**   | **1.05**     |
> | WaNet    | 91.29   | 99.91    | 89.87      | 99.93       | **90.64**   | **1.93**     |
> | Adaptive | 89.62   | 100      | 89.40      | 49.18       | **90.54**   | **1.23**     |
>
> **Table 3**. The performance (\%) of REFINE and BDMAE on **ResNet50**. **ND** means that no defense is applied. The best results are **boldfaced**.
> |          | BA (ND) | ASR (ND) | BA (BDMAE) | ASR (BDMAE) | BA (REFINE) | ASR (REFINE) |
> | -------- | ------- | -------- | ---------- | ----------- | ----------- | ------------ |
> | BadNets  | 91.91   | 100      | **91.04**  | 10.99       | 90.71       | **1.53**     |
> | WaNet    | 91.70   | 99.98    | 89.83      | 99.89       | **91.09**   | **0.35**     |
> | Adaptive | 89.59   | 85.11    | 89.06      | 35.91       | **90.05**   | **2.19**     |
>
> **Table 4**. The performance (\%) of REFINE and BDMAE on **VGG16**. **ND** means that no defense is applied. The best results are **boldfaced**.
> |          | BA (ND) | ASR (ND) | BA (BDMAE) | ASR (BDMAE) | BA (REFINE) | ASR (REFINE) |
> | -------- | ------- | -------- | ---------- | ----------- | ----------- | ------------ |
> | BadNets  | 84.44   | 99.36    | 84.25      | 18.32       | **86.86**   | **1.62**     |
> | WaNet    | 84.75   | 99.15    | 83.36      | 99.25       | **86.41**   | **2.39**     |
> | Adaptive | 84.98   | 99.99    | 84.69      | 40.09       | **86.63**   | **2.04**     |
>
> **Table 5**. The performance (\%) of REFINE and BDMAE on **DenseNet121**. **ND** means that no defense is applied. The best results are **boldfaced**.
> |          | BA (ND) | ASR (ND) | BA (BDMAE) | ASR (BDMAE) | BA (REFINE) | ASR (REFINE) |
> | -------- | ------- | -------- | ---------- | ----------- | ----------- | ------------ |
> | BadNets  | 86.40   | 99.99    | 86.05      | 11.85       | **89.44**   | **0.96**     |
> | WaNet    | 86.31   | 98.77    | 85.42      | 98.91       | **88.74**   | **0.88**     |
> | Adaptive | 85.16   | 100      | 84.45      | 45.36       | **88.74**   | **0.35**     |
>
> **Table 6**. The performance (\%) of REFINE and BDMAE on **InceptionV3**. **ND** means that no defense is applied. The best results are **boldfaced**.
> |          | BA (ND) | ASR (ND) | BA (BDMAE) | ASR (BDMAE) | BA (REFINE) | ASR (REFINE) |
> | -------- | ------- | -------- | ---------- | ----------- | ----------- | ------------ |
> | BadNets  | 90.46   | 99.97    | 90.61      | 80.51       | **91.03**   | **0.75**     |
> | WaNet    | 90.09   | 99.73    | 89.64      | 99.76       | **91.01**   | **0.54**     |
> | Adaptive | 88.58   | 13.53    | 88.54      | 13.52       | **90.36**   | **0.54**     |

---

> ### Author Response · Authors · 2024-11-20
> **Author Response (Part 4)**
>
> ---
>
> **Q4**: The two sentences (i.e., "by first obtaining ... outcomes" and "the internet ... poisoned samples)") in the Section 1 contradict each other. What is the difference between prior information and prior knowledge?
>
> **R4**: Thank you for pointing it out! We sincerely apologize for any potential confusion and misunderstanding our expression may have caused.
>
> - In the original paper,
>     - '**prior information**' refers to the trigger information obtained through trigger inversion in BTI-based defenses, including trigger patterns or backdoor features.
>     - '**prior knowledge**' refers to the knowledge of backdoor attacks, such as the implanted trigger pattern, trigger type, and target labels.
> - What we intended to express is that BTI-based methods perform pre-processing-based defenses by inverting trigger information. However, the lack of prior knowledge about the trigger prevents them from achieving the desired defense performance. In Section 3.3 of the paper, we provide experimental evidence to support this argument. Although our expression in the original paper may have caused some misunderstandings, the intention behind it is reasonable and well-founded.
> - Accordingly, to avoid any misunderstanding, we have changed '**prior information**' to '**information**' in the original paper:
>
> ```!
> On the other hand, BTI-based defenses can break the trade-off by first obtaining the information of backdoor triggers via trigger inversion but often fall short of achieving the desired defensive outcomes. This is mostly because their effectiveness heavily relies on the quality of inverted trigger patterns, whereas the inverted trigger patterns are often significantly different from the ones predefined by the adversary due to the inherent difficulties of BTI (e.g., lack of prior knowledge about the implanted backdoor and poisoned samples.
> ```
>
>
> We sincerely thank you again for pointing out this issue. We polish this sentence in *Section 1* of our revision.
>
>
> ---
>
> **Q5**: This paper focuses on the dirty-image backdoor attacks. But considering the recent clean-image backdoor attacks that do not have trigger features, can the proposed defense method work?
>
> **R5**: Thank you for this constructive suggestion!
>
> - Based on your suggestion, **we test the clean-image backdoor attack [1] (dubbed 'Clean-image')**.
>     - We train a ResNet18 backdoor model on CIFAR10 and applied defenses using BDMAE, BTI-DBF\(P\), and REFINE.
>     - For each defense method, we test the model's benign accuracy (BA), natural attack success rate (ASR-n), and manual attack success rate (ASR-m). ASR-m represents the ASR of poisoned samples that naturally contain the backdoor trigger, while ASR-m represents the ASR of manually generated poisoned samples.
> - As shown in Table 7, existing pre-processing-based defenses are unable to effectively reduce the ASR-n of the clean-image attack. Compared to BDMAE and BTI-DBF\(P\), **REFINE shows better defense performance**.
> - **However, we believe that the Clean-image attack, from certain perspectives, extends beyond the typical scope of backdoor attacks**. Specifically, the stealthiness of backdoor attacks usually requires the model to behave normally on benign samples, while the clean-image attack may cause misclassifications on some benign samples (i.e., natural poisoned samples), which significantly impacts the model's normal inference. We will discuss how to defend against this type of attack in our future work.
>
> We provide more details in *Appendix E.1* of our revision.
>
>
> **Table 7**. The performance (\%) of BDMAE, BTI-DBF\(P\) and REFINE against Clean-image. **ND** means that no defense is applied. The best results are **boldfaced**.
> |             | BA (ND) | ASR-n (ND) | ASR-m (ND) | BA (BDMAE) | ASR-n (BDMAE) | ASR-m (BDMAE) | BA (BTI-DBF\(P\)) | ASR-n (BTI-DBF\(P\)) | ASR-m (BTI-DBF\(P\)) | BA (REFINE) | ASR-n (REFINE) | ASR-m (REFINE) |
> | ----------- | ------- | ---------- | ---------- | ---------- | ------------- | ------------- | ----------------- | -------------------- | -------------------- | ------------- | -------------- | -------------- |
> | Clean-image | 87.78   | 82.93      | 4.73       | 86.73      | 58.54         | 4.73          | 78.95             | 86.59                | 9.53                 | **88.94**   | **53.57**   | **4.63**       |
>
>
> **References**
>
> 1. Clean-image backdoor attacks.

---

> ### Author Response · Authors · 2024-11-20
> **Author Response (Part 5)**
>
> ---
>
> **Q6**: In Section 3.2, the "Pad Size" and "Mask Ratio" are not defined before. it is necessary to clarify.
>
> **R6**: Thank you for pointing it out! We apologize for not clearly defining "Pad Size" and "Mask Ratio" in the original paper. **We have now provided accurate definitions and clarification in the revised paper**, as follows:
>
> ```!
> We use "Pad Size" (dubbed "S"), which refers to the padding size applied around the shrunk image, and "Mask Ratio" (dubbed "R"), which represents the masking rate applied to the image before reconstruction, to control the transformation intensity for ShrinkPad and BDMAE, respectively.
> ```
> ---
>
> **Q7**: In this paper, there are many long sentences, which makes it hard to understand. Such as "Additionally, we treat ... the transformation" on Page 4.
>
> **R7**: Thank you for these constructive suggestions! We agree that overusing long sentences may lead to confusion. We have proofread our paper and rewriten all the long sentences. Some examples are as follows.
>
> - In line 90-92, we rewrite 'Besides, we further ... after transformation' to:
> ```!
> Besides, we further improve our method by imposing constraints on the transformed samples using supervised contrastive loss. This ensures that samples of the same class remain more similar after transformation.
> ```
> - In line 210-213, we rewrite 'Additionally, we treat ... the transformation' to:
> ```!
> Additionally, we treat the original model as a feature extractor. We then visualize how transformation intensity affects the differences in feature distribution between benign and backdoor samples of the same class, both before and after the transformation.
> ```
> - In line 365-367, we rewrite 'Therefore, ... as follows' to:
> ```!
> Therefore, before calculating these loss functions, it is necessary to obtain the pseudo-labels $\bar{Y}$ for the current batch of unlabeled samples $X$. $\bar{Y}$ is predicted by the original model (without any additional modules), as follows.
> ```
>
>
> ---
>
> **Q8**: In Section 3.2, the authors did not show the dataset used for limitation analysis. Please clarify it.
>
> **R8**: Thank you for pointing it out! We apologize for not specifying the dataset used for the limitation analysis in Section 3.2. **We have now clarified this in the revised paper**, as follows:
> ```!
>  ShrinkPad (Li et al., 2021c) (dubbed "SP") and BD-MAE (Sun et al., 2023) (dubbed "BD"), to defend the BadNets attack (Gu et al., 2019) on CIFAR-10.
> ```
>
> ---
>
> **Q9**: In Section 4.2.1, the authors present that traditional model reprogramming methods are insufficient to remove triggers. I suggest that the authors explain why traditional methods are insufficient in detail.
>
> **R9**: Thank you for this constructive suggestion! We are sorry for not specifying that traditional model reprogramming methods are insufficient to remove trigger patterns in the original paper.
> - **We have provided a further explanation in the revised paper**, as follows:
> ```!
> Traditional model reprogramming methods (Elsayed et al., 2019; Tsai et al., 2020) add the optimized universal adversarial perturbation around the input, while the trigger pattern still remains intact on the backdoor image to some extent.
> ```
> - To further alleviate your concerns, we conduct additional defense experiments using traditional model reprogramming methods [1] (dubbed 'T-MR').
>     - We select three representative types of backdoor attacks, including BadNets, WaNet, and BATT.
>     - We train backdoor ResNet18 models on the CIFAR-10 dataset.
>     - We compare the defense performance of REFINE with T-MR.
> - As shown in Table 8, **the T-MR defense has a significant impact on the model's BA (BA drop > 15%) but fails to effectively reduce the ASR under the WaNet attack**. This is because traditional model reprogramming methods only add a universal adversarial perturbation around the image, while the trigger pattern remains unchanged on the backdoor image to some extent.
>
> We provide more details in *Appendix E.6* of our revision.
>
> **Table 8**. The performance (\%) of REFINE and T-MR. **ND** means that no defense is applied. The best results are **boldfaced**.
> |          | BA (ND) | ASR (ND) | BA (T-MR) | ASR (T-MR) | BA (REFINE) | ASR (REFINE) |
> | -------- | ------- | -------- | --------- | ---------- | ----------- | ------------ |
> | BadNets  | 91.18   | 100      | 75.51     | 3.36       | **90.50**   | **1.05**     |
> | WaNet    | 91.29   | 99.91    | 74.49     | 25.76      | **90.64**   | **1.93**     |
> | BATT     | 92.54   | 99.93    | 75.49     | 5.87       | **90.87**   | **1.76**     |
>
>
> **Reference**
> 1. Adversarial Reprogramming of Neural Networks.

---

> ### Author Response · Authors · 2024-11-20
> **Author Response (Part 6)**
>
> ---
>
> **Q10**: The discussion about the feature distance changes in the proposed input transformation module is lacking. It is better to add it to highlight the function of this module.
>
> **R10**: Thank you for this constructive suggestion! We are deeply sorry for not using figures to visually demonstrate the feature changes of the input samples before and after REFINE.
> - To visually reflect the changes of input sample distribution in the feature space before and after REFINE, **we used t-SNE plots for visualization**. We add two figures and a detailed explanation in *Appendix J* to clarify those details. The illustrations of added figures are as follows:
> ```!
> As shown in Figure 8, before applying REFINE, the feature distributions of benign and poisoned samples are clustered in two distinct locations. After applying REFINE, the feature distributions of benign and poisoned samples are interwoven and clustered in the same new location. This indicates that REFINE effectively removes the trigger patterns from the poisoned samples and maps samples of the same class to a new benign feature distribution.
> ```
> ```!
> As shown in Figure 9, before applying REFINE, the benign samples of each class form distinct clusters in the feature space. After applying REFINE, the benign samples, adjusted by the input transformation module and output mapping module, form new clusters in different positions. This empirically demonstrates that REFINE is able to maintain the model's benign accuracy.
> ```

---

> ### Author Response · Authors · 2024-11-23
> **Thanks to Reviewer p35s**
>
> Please allow us to thank you again for reviewing our paper and the valuable feedback, and in particular for recognizing the strengths of our paper in terms of the theoretical analysis, novel and interesting method, and good soundness.
>
> Please let us know if our response and the new experiments have properly addressed your concerns. We are more than happy to answer any additional questions during the post-rebuttal period. Your feedback will be greatly appreciated.

---

> ### Author Response · Authors · 2024-11-24
> **A Gentle Reminder of the Final Feedback**
>
> Thank you very much again for your initial comments. They are extremely valuable for improving our work. We shall be grateful if you can have a look at our response and modifications, and please let us know if anything else that can be added to our next version.

---

> ### Author Response · Authors · 2024-11-25
> **A Second Reminder of the Post-rebuttal Feedback**
>
> Dear Reviewer p35s,
>
> We greatly appreciate your initial comments. We totally understand that you may be extremely busy at this time. But we still hope that you could have a quick look at our responses to your concerns. We appreciate any feedback you could give to us. We also hope that you could kindly update the rating if your questions have been addressed. We are also happy to answer any additional questions before the rebuttal ends.
>
> Best Regards,
>
> Paper4208 Authors

---

> > ### Comment · Reviewer_GGnQ · 2024-11-26
> > **The Responsibility of Accepting a Reviewer Invitation?**
> >
> > **Despite repeated reminders from the authors**, Reviewer p35s (and also Q8vQ) has chosen not to provide any further responses, remaining unresponsive. **This is disappointing and uncharacteristic of ICLR, a conference renowned for its high level of interaction and engagement.** If a reviewer is unwilling to engage in any form of interaction with the authors, perhaps they should reconsider submitting their review comments or even accepting the invitation to review in the first place.

---

> ### Author Response · Authors · 2024-11-28
> **A Third Reminder for Post-rebuttal Feedback**
>
> Dear Reviewer p35s,
>
> We sincerely thank you for your time and valuable comments during the review process. As the paper revision stage has now ended, we kindly hope you could take a moment to review our responses and updates. If you have any remaining concerns, we would be happy to address them further.
>
> Thank you again for your understanding and support.
>
> Best Regards,
> Paper4208 Authors

---

> ### Author Response · Authors · 2024-12-02
> **Reminder of the Post-rebuttal Feedback and Summary of Our Response**
>
> Dear Reviewer p35s,
>
> Thank you for your time and effort in evaluating our work. We greatly appreciate your initial comments. Your insights and suggestions are extremely valuable to us.
>
> Given that we have only one day left for discussion, we are hoping to receive any additional feedback or question you might have at your earliest convenience. Your expertise would be of great help to us in improving the quality and rigor of our work.
>
> To facilitate the discussion, we would like to summarize our response as follows.
>
> - **We further clarified the relationship between our motivation and the proposed method**, primarily analyzing the limitations of existing pre-processing-based defenses from the domain aspect.
> - **We conducted extensive supplementary experiments**, including:
>     - Demonstrating the effectiveness of REFINE in defending against backdoor attacks using independently distributed datasets.
>     - Validating the effectiveness of REFINE in defending against backdoor attacks across different model architectures.
>     - Showing that REFINE is more effective than baseline methods in defending against clean-image attacks.
>     - Proving that traditional model reprogramming methods struggle to achieve satisfactory defense performance.
>     - Visualizing the distribution differences between benign and poison samples before and after applying REFINE using t-SNE plots.
> - **We revised the unclear parts of the original paper** based on your suggestions, including refining word choices, breaking long sentences into shorter ones, and adding clarifications.
>
> If our responses address your concerns, we kindly request that you reconsider your evaluations. We would also be grateful for any additional comments or suggestions you might have to refine our work.
>
> Best regards,
>
> Paper4208 Authors
>
> ---

---

### Official Review · Reviewer_gceM · 2024-11-07

**Soundness:** 3
**Presentation:** 2
**Contribution:** 2
**Rating:** 5
**Confidence:** 3

**Summary:**

This paper propose an inversion-free backdoor defense method based on model reprogramming, REFINE. REFINE consists of an input transformation module and an output mapping module. The key idea is to transform both input and output domains to break the trade-off between model utility and defense effectiveness faced by existing pre-processing based defenses.

**Strengths:**

- Extensive experiments demonstrating the effectiveness of REFINE across different datasets.

- Thorough ablation studies validating each component's contribution

**Weaknesses:**

- The connection between motivation and the proposed methods is not very close. However, the analysis of BTI-based defenses does not involve this domain-based perspective. The paper didn’t discuss whether and how the purification process in BTI-based defenses alter the image domain.

- The theoretical analysis through Theorem 1 effectively explains the limitations of transformation-based defenses by quantifying how domain transformations affect defense performance.

- Experiments mainly focus on ResNet. Conducting experiments under different network structures is beneficial for verifying the effectiveness of the proposed method, for example, VGG, DenseNet etc.

- Requiring 20% clean data may be impractical in many real-world scenarios.

- The proposed method requires training a U-Net, which involves a significant computational cost.

**Questions:**

In Figure 3, why did the color of the part of the image outside the trigger also change?

---

> ### Author Response · Authors · 2024-11-20
> **Author Response (Part 1)**
>
> Dear Reviewer gceM, thank you very much for your careful review of our paper and thoughtful comments. We are encouraged by your positive comments on our **extensive experiments**, **thorough ablation studies**, and **good soundness**. We hope the following responses can help clarify potential misunderstandings and alleviate your concerns.
>
> ---
>
> **Q1**: The connection between motivation and the proposed methods is not very close. However, the analysis of BTI-based defenses does not involve this domain-based perspective. The paper didn’t discuss whether and how the purification process in BTI-based defenses alter the image domain.
>
> **R1**: Thank you for this insightful comment! We sincerely apologize for failing to clearly express the connection between our motivation and the proposed method. We would like to provide additional clarifications as follows.
>
> - In general, pre-processing-based backdoor defenses can be categorized into two types: transformation-based defenses and backdoor trigger inversion (BTI)-based defenses. **Fundamentally, both types of defenses modify the samples in the image domain.**
>     - Transformation-based defenses distort trigger patterns by applying random transformations to the input or reconstructing the input using generative models. These approaches represent a paradigm that **maps backdoor inputs from the poisoned image domain to a transformed image domain**.
>     - **BTI-based defenses aim to leverage the inverted triggers to precisely map backdoor inputs from the poisoned image domain to the benign image domain**. Specifically, BTI-based defenses first train a trigger generator $T(\cdot)$, which can generate a trigger pattern $T(x)$ that mimics the attacker's trigger based on the input image $x$. During the defense process, subtracting the trigger pattern $T(x)$ from each input image $x$ yields a purified image $x - T(x)$. This process effectively maps backdoor inputs from the poisoned image domain to the benign image domain.
> - **Our Motivations from the Domain Aspect**:
>     - **For transformation-based defenses, fixing the output domain limits the transformations in the image domain.** Specifically, overly large transformations reduce the model's benign accuracy, while small transformations fail to reduce backdoor attack success rate effectively. Mapping the poisoned image domain to an appropriate transformed domain presents a trade-off.
>     - **For BTI-based defenses, the inherent difficulty of trigger inversion leads to inaccurate image domain mapping.** Due to the lack of prior knowledge about the attack, existing BTI methods struggle to invert the ground-truth trigger. Therefore, using an incorrect trigger makes it challenging to purify the poisoned image domain to the benign image domain.
>     - Based on the above understanding, **to increase the flexibility of image domain transformation and avoid the need for accurate transformations, we map the output domain** and propose our method, REFINE. REFINE can significantly alter the input domain while preserving the model’s accuracy by allowing changes to the output domain.
>
> ---
>
> **Q2**: The theoretical analysis through Theorem 1 effectively explains the limitations of transformation-based defenses by quantifying how domain transformations affect defense performance.
>
> **R2**: Thank you for this positive comment! We are delighted to have your recognition.

---

> ### Author Response · Authors · 2024-11-20
> **Author Response (Part 2)**
>
> ---
>
> **Q3**: Experiments mainly focus on ResNet. Conducting experiments under different network structures is beneficial for verifying the effectiveness of the proposed method, for example, VGG, DenseNet etc.
>
>
> **R3**: Thank you for these constructive suggestions! We are deeply sorry for not demonstrating the effectiveness of our method across different network structures in our previous submission. We do agree with you that it is beneficial to verify the universality of our defense across different network structures.
> - To address your concerns, following your suggestions, **we conduct additional experiments on five different network structures**, including ResNet18, ResNet50, VGG16, DenseNet121 and InceptionV3. We select three representative types of backdoor attacks, including BadNets, WaNet and Adaptive-Patch (dubbed ‘Adaptive’). We conduct experiments on the CIFAR-10 dataset. We compare the defense performance of our REFINE with the most advanced transformation-based defense (i.e., BDMAE).
> - As shown in Tables 1 to 5, REFINE effectively defends against three representative attacks across five different network architectures, significantly outperforming BDMAE. Specifically, under the REFINE defense, the benign accuracy (BA) drop is less than 1.5%, with some cases showing an increase in BA. Meanwhile, the backdoor attack success rate (ASR) reduces to below 3%. **The additional experimental results verify the effectiveness of REFINE.**
>
> We provide more details in *Appendix E.1* of our revision.
>
> **Table 1**. The performance (\%) of REFINE and BDMAE on **ResNet18**. **ND** means that no defense is applied. The best results are **boldfaced**.
> |          | BA (ND) | ASR (ND) | BA (BDMAE) | ASR (BDMAE) | BA (REFINE) | ASR (REFINE) |
> | -------- | ------- | -------- | ---------- | ----------- | ----------- | ------------ |
> | BadNets  | 91.18   | 100      | 90.48      | 14.81       | **90.50**   | **1.05**     |
> | WaNet    | 91.29   | 99.91    | 89.87      | 99.93       | **90.64**   | **1.93**     |
> | Adaptive | 89.62   | 100      | 89.40      | 49.18       | **90.54**   | **1.23**     |
>
> **Table 2**. The performance (\%) of REFINE and BDMAE on **ResNet50**. **ND** means that no defense is applied. The best results are **boldfaced**.
> |          | BA (ND) | ASR (ND) | BA (BDMAE) | ASR (BDMAE) | BA (REFINE) | ASR (REFINE) |
> | -------- | ------- | -------- | ---------- | ----------- | ----------- | ------------ |
> | BadNets  | 91.91   | 100      | **91.04**  | 10.99       | 90.71       | **1.53**     |
> | WaNet    | 91.70   | 99.98    | 89.83      | 99.89       | **91.09**   | **0.35**     |
> | Adaptive | 89.59   | 85.11    | 89.06      | 35.91       | **90.05**   | **2.19**     |
>
> **Table 3**. The performance (\%) of REFINE and BDMAE on **VGG16**. **ND** means that no defense is applied. The best results are **boldfaced**.
> |          | BA (ND) | ASR (ND) | BA (BDMAE) | ASR (BDMAE) | BA (REFINE) | ASR (REFINE) |
> | -------- | ------- | -------- | ---------- | ----------- | ----------- | ------------ |
> | BadNets  | 84.44   | 99.36    | 84.25      | 18.32       | **86.86**   | **1.62**     |
> | WaNet    | 84.75   | 99.15    | 83.36      | 99.25       | **86.41**   | **2.39**     |
> | Adaptive | 84.98   | 99.99    | 84.69      | 40.09       | **86.63**   | **2.04**     |
>
> **Table 4**. The performance (\%) of REFINE and BDMAE on **DenseNet121**. **ND** means that no defense is applied. The best results are **boldfaced**.
> |          | BA (ND) | ASR (ND) | BA (BDMAE) | ASR (BDMAE) | BA (REFINE) | ASR (REFINE) |
> | -------- | ------- | -------- | ---------- | ----------- | ----------- | ------------ |
> | BadNets  | 86.40   | 99.99    | 86.05      | 11.85       | **89.44**   | **0.96**     |
> | WaNet    | 86.31   | 98.77    | 85.42      | 98.91       | **88.74**   | **0.88**     |
> | Adaptive | 85.16   | 100      | 84.45      | 45.36       | **88.74**   | **0.35**     |
>
> **Table 5**. The performance (\%) of REFINE and BDMAE on **InceptionV3**. **ND** means that no defense is applied. The best results are **boldfaced**.
> |          | BA (ND) | ASR (ND) | BA (BDMAE) | ASR (BDMAE) | BA (REFINE) | ASR (REFINE) |
> | -------- | ------- | -------- | ---------- | ----------- | ----------- | ------------ |
> | BadNets  | 90.46   | 99.97    | 90.61      | 80.51       | **91.03**   | **0.75**     |
> | WaNet    | 90.09   | 99.73    | 89.64      | 99.76       | **91.01**   | **0.54**     |
> | Adaptive | 88.58   | 13.53    | 88.54      | 13.52       | **90.36**   | **0.54**     |

---

> ### Author Response · Authors · 2024-11-20
> **Author Response (Part 3)**
>
> ---
>
> **Q4**: Requiring 20% clean data may be impractical in many real-world scenarios.
>
> **R4**: Thank you for pointing it out and we do understand your concern. However, there may be some misunderstandings that we want to clarify here.
>
> - Our method only requires **unlabeled clean samples** instead of labeled ones.
> - Arguably, **obtaining unlabeled samples is relatively easy in practice**, although it is expensive to obtain labeled ones. For example, we usually can find these samples from the Internet or collect these data via sensors in the physical world.
> - To further alleviate your concern, we also conduct additional experiments under the scenario of using 10% unlabeled clean data. We apply the REFINE defense to a ResNet18 model trained on the CIFAR-10 dataset, which is subjected to the BadNets attack. In this case, the unlabeled training set for REFINE used only 10% of the CIFAR-10 training set.
>     - As shown in Table 6, even with only 10% unlabeled data, **REFINE is still effective to some extent**.
>     - We will discuss how to further reduce the need for unlabeled samples in our future work.
>
> We provide more details in *Appendix M* of our revision.
>
> **Table 6**. The performance (\%) of REFINE in the 10% unlabeled data scenario on ResNet18. **ND** means that no defense is applied.
> |          | BA (ND) | ASR (ND) | BA (REFINE) | ASR (REFINE) |
> | -------- | ------- | -------- | ----------- | ------------ |
> | BadNets  | 91.18   | 100      | 78.02       | 2.90         |
>
>
> ---
>
> **Q5**: The proposed method requires training a U-Net, which involves a significant computational cost.
>
> **R5**: Thank you for this insightful comment!
>
> - We acknowledge that training the input transformation module (U-Net) of REFINE incurs some computational cost. But we argue that **it is acceptable to a large extent**.
>     - For example, defending a ResNet18 model trained on CIFAR-10 takes only 40 minutes with a single RTX 3090 GPU. **This training time is lower than the time required to train a ResNet18 model from scratch on the CIFAR-10 dataset.**
>     - We can also further reduce the training cost by incorporating techniques like early stopping.
>     - We have discussed the overhead of REFINE in *Appendix G*.
> - Arguably, **the main objective of this paper is to clarify the design principles and processes of REFINE**, rather than to introduce specific improvements to the training process. We were concerned that presenting too much of this aspect would make it more difficult for readers to understand and would result in non-comprehensive writing of the methodology (due to space constraints). We will explore how to accelerate the process of REFINE in our future work.
> - Besides, **our method can be used in conjunction with existing (model reconstruction-based) defenses** to further enhance their effectiveness.
>     - To demonstrate this, we first applied model fine-tuning defense (dubbed 'FT') to a ResNet18 model subjected to the BadNets attack on CIFAR-10, followed by the REFINE defense.
>     - As shown in Table 7, the **FT+REFINE defense effectively reduces the backdoor ASR while maintaining the model's BA**.
>
> We provide more details in *Appendix H* of our revision.
>
>
> **Table 7**. The performance (\%) of FT and FT+REFINE on ResNet18. **ND** means that no defense is applied.
> |          | BA (ND) | ASR (ND) | BA (FT) | ASR (FT) | BA (FT+REFINE) | ASR (FT+REFINE) |
> | -------- | ------- | -------- | ------- | -------- | -------------- | --------------- |
> | BadNets  | 91.18   | 100      | 91.89   | 91.67    | 90.42          | 0.87            |
>
> ---
>
> **Q6**: In Figure 3, why did the color of the part of the image outside the trigger also change?
>
> **R6**: Thank you for pointing it out!
> - We have to admit that **this is due to a typo** and sincerely appreciate your reminder.
> - The reason for this typo is that during the plotting process, we used the `cv2.imread` function to read benign samples, then added the trigger pattern and saved the image using the `PIL.Image.save` function. This led to a swap of the RGB channel values.
> - To ensure the rigor of the paper, **we have redrawn Figure 3 and thoroughly checked the entire manuscript** to ensure that this issue does not occur elsewhere.
>
> ---

---

> ### Author Response · Authors · 2024-11-23
> **Thanks to Reviewer gceM**
>
> Please allow us to thank you again for reviewing our paper and the valuable feedback, and in particular for recognizing the strengths of our paper in terms of extensive experiments, thorough ablation studies, and good soundness.
>
> Please let us know if our response and the new experiments have properly addressed your concerns. We are more than happy to answer any additional questions during the post-rebuttal period. Your feedback will be greatly appreciated.

---

> ### Author Response · Authors · 2024-11-24
> **A Gentle Reminder of the Final Feedback**
>
> Thank you very much again for your initial comments. They are extremely valuable for improving our work. We shall be grateful if you can have a look at our response and modifications, and please let us know if anything else that can be added to our next version.

---

> ### Author Response · Authors · 2024-11-25
> **A Second Reminder of the Post-rebuttal Feedback**
>
> Dear Reviewer gceM,
>
> We greatly appreciate your initial comments. We totally understand that you may be extremely busy at this time. But we still hope that you could have a quick look at our responses to your concerns. We appreciate any feedback you could give to us. We also hope that you could kindly update the rating if your questions have been addressed. We are also happy to answer any additional questions before the rebuttal ends.
>
> Best Regards,
>
> Paper4208 Authors

---

> > ### Comment · Reviewer_gceM · 2024-11-25
> >
> > Thanks for the response. I will keep my original score for the following reasons:
> > 1. Experiments with different structures should be prepared in the original paper, rather than addressed during the rebuttal.
> > 2. I remain skeptical about the effectiveness of REFINE in the 10% clean data scenario since there are no more experiments on the other attacks.
> > 3. For Q5, the time consumption should be compared with other defense methods instead of training a model from scratch.
> > 4. The obvious typo in Figure 3 shows that this paper is not fully prepared.

---

> > > ### Author Response · Authors · 2024-11-26
> > > **Author Response Round 2**
> > >
> > > Dear Reviewer gceM, we sincerely thank you for taking the time to respond to our reply despite your busy schedule. We also found that there may still be some potential misunderstandings. We highly value your efforts and respect your opinions and therefore we want to further clarify them.
> > >
> > >
> > > ---
> > > **Q1**: Experiments with different structures should be prepared in the original paper, rather than addressed during the rebuttal.
> > >
> > > **R1**: Thank you for this constructive suggestion regarding experiments with REFINE on different architectures. We sincerely apologize for not testing REFINE on different model architectures in the original paper due to the limitation of time and space.
> > > - During the rebuttal stage, we conduct a large number of additional experiments. **The results have demonstrated that REFINE exhibits excellent defense performance across different model architectures**, and we have added them in *Appendix E.1* of our revision.
> > > - To be honest, we did not include this part of the experiment purely due to time and space constraints. As you can see in our upload history, we fought to deliver this paper almost until the last minute of submission DDL.
> > > - Of course, we fully acknowledge that this is our problem and we should have started relevant experiments earlier. However, many times plans really cannot keep up with changes. I believe that as a senior expert (for being invited as the reviewer), you can fully understand this.
> > >
> > >
> > > Thank you again for your constructive suggestions for adding these new experiments. We are very willing to provide more details and discussions if you still have any concerns. Again, we do highly value your efforts and respect your opinions :)
> > >
> > > ---
> > >
> > > **Q2**: I remain skeptical about the effectiveness of REFINE in the 10% clean data scenario since there are no more experiments on the other attacks.
> > >
> > > **R2**: Thank you for this constructive suggestion!
> > > - Our previous response focused solely on BadNets simply because our previous ablation experiments in the appendix only used BadNets attack as an example for illustration.
> > > - To further alleviate your concerns, we now evaluate our method under three additional attacks. As shown in Table 1, similar to its defense against BadNets, **REFINE is still effective to some extent**.
> > > - We also respectfully notice that **we mainly focus on scenarios where defenders can obtain some instead of highly limited unlabeled benign samples** since obtaining unlabeled samples is relatively easy in practice (as we have justified in our previous rebuttal).
> > >
> > > **Table 1**. The performance (\%) of REFINE in the 10% unlabeled data scenario on ResNet18. **ND** means that no defense is applied.
> > > |            | BA (ND) | ASR (ND) | BA (REFINE) | ASR (REFINE) |
> > > | ---------- | ------- | -------- | ----------- | ------------ |
> > > | BadNets    | 91.18   | 100      | 78.02       | 2.90         |
> > > | Blended    | 90.64   | 98.18    | 77.89       | 2.59         |
> > > | WaNet      | 91.29   | 99.91    | 78.79       | 1.83         |
> > > | PhysicalBA | 93.67   | 100      | 79.87       | 2.34         |
> > >
> > > ---
> > >
> > > **Q3**: For Q5, the time consumption should be compared with other defense methods instead of training a model from scratch.
> > >
> > > **R3**: Thank you for this constructive suggestion! We apologize for not comparing REFINE's runtime with other defenses.
> > > - To further alleviate your concern, we compare REFINE's time consumption with that of BDMAE and BTI-DBF\(P\), which are the representative of SOTA transformation-based and BTI-based defenses, respectively.
> > > - As shown in Table 2, **the overall time overhead of our defense is on par with SOTA baselines**. PS: Training the transformation module is a one-time process and can be done offline, although the pre-processing happens online (whose cost for an image is nearly 0 as we have discussed in *Appendix G*). As such, this overhead is acceptable.
> > >
> > > **Table 2**. The overhead (minutes) of REFINE compared with BDMAE and BTI-DBF\(P\).
> > > |           | BDMAE  | BTI-DBF\(P\) | REFINE |
> > > | --------- | ------ | ------------ | ------ |
> > > | Overhead  | 39.67  | 15.49        | 36.70  |
> > >
> > >
> > > ---
> > >
> > > **Q4**: The obvious typo in Figure 3 shows that this paper is not fully prepared.
> > >
> > > **R4**: Thank you for pointing out this typo!
> > > - We **have clarified the source of this typo** in our previous rebuttal, and **it will not affect the results of all other experiments**.
> > > - We **have also carefully double-checked all results** to ensure that they are accurate.
> > > - We are deeply sorry again for this negligence. However, **we respectfully argue that having a few typos is not a reasonable reason to maintain a low score for rejection**, as long as they have been fixed and do not influence the main results.
> > >
> > > However, we are very willing to provide more details and discussions if you still have any concerns. Again, we do highly value your efforts and respect your opinions :)
> > >
> > >
> > > ---

---

### Meta-Review · Area_Chair_XWCf · 2024-12-25

**Metareview:**

This paper proposes REFINE,  an inversion-free backdoor defense based on model reprogramming. This approach eliminates backdoor attacks while also maintaining the model's performance. The reviewers think the proposed method is simple but effective, and the approach is novel and interesting. The reviewers' concerns include that the original submission only includes the experiments on ResNet-18. The authors provided more experiments on a broader set of architectures. In my opinion, the authors addressed most of the major concerns raised by the reviewers. Given the novelty and the effectiveness of the proposed method, I recommend accept (poster).

**Additional Comments On Reviewer Discussion:**

The reviewers' concerns include that the original submission only includes the experiments on ResNet-18. The authors provided more experiments on a broader set of architectures. Reviewer gceM thinks experiments with different structures should be prepared in the original submission, rather than addressed during the rebuttal. however, I think this is more commonly accepted to provide new relevant results asked by the reviewers during the rebuttal. Some reviewers are rather unresponsive to the authors' responses, although the authors did tremendous work to answer the reviewers' questions. In my opinion, the authors addressed most of the major concerns raised by the reviewers.

---

### Decision · Program_Chairs · 2025-01-22

Accept (Poster)